# Finding Second-order Stationary Points for Generalized-Smooth Nonconvex Minimax Optimization via Gradient-based Algorithm

## Abstract

Nonconvex minimax problems have received intense interest in many machine learning applications such as generative adversarial network, robust optimization and adversarial training. Recently, a variety of minimax optimization algorithms based on Lipschitz smoothness for finding first-order or second-order stationary points have been proposed. However, the standard Lipschitz continuous gradient or Hessian assumption could fail to hold even in some classic minimax problems, rendering conventional minimax optimization algorithms fail to converge in practice. To address this challenge, we demonstrate a new gradient-based method for nonconvex-strongly-concave minimax optimization under a generalized smoothness assumption. Motivated by the important application of escaping saddle points, we propose a generalized Hessian smoothness condition, under which our gradient-based method can achieve the complexity of $\mathcal{O}(\epsilon^{-1.75} \log n)$ to find a second-order stationary point with only gradient calls involved, which improves the state-of-the-art complexity results for the nonconvex minimax optimization even under standard Lipschitz smoothness condition. To the best of our knowledge, this is the first work to show convergence for finding second-order stationary points on nonconvex minimax optimization with generalized smoothness. The experimental results on the application of domain adaptation confirm the superiority of our algorithm compared with existing methods.

## 1 Introduction

In recent years, minimax optimization problems, under various assumptions on the objective functions, has been a major focus of research in machine learning fields, with various applications including adversarial training (Madry et al., 2018), generative adversarial networks (GAN) (Goodfellow et al., 2014), and multi-agent reinforcement learning (Omidshafiei et al., 2017). A general formulation of Minimax optimization problem can be written as

$$\min_{\mathbf{x} \in \mathbb{R}^n} \max_{\mathbf{y} \in \mathbb{R}^d} f(\mathbf{x}, \mathbf{y}) \tag{1}$$

In this paper, we focus on the nonconvex-strongly-concave case where the objective function $f : \mathbb{R}^m \times \mathbb{R}^n \mapsto \mathbb{R}$ is nonconvex in $\mathbf{x}$ and strongly-concave in $\mathbf{y}$.

Historically, Nouiehed et al. (2019) was the first work providing non-asymptotic convergence rates for nonconvex-strongly-concave minimax problems without assuming special structure of the objective function. They use the notion of $\epsilon$-first-order stationary point to measure the rate of convergence of their algorithm. Using this notion, they showed that their algorithm finds an $\epsilon$-first-order stationary point in $\mathcal{O}(\epsilon^{-2})$ gradient evaluations.

Another way to measure the convergence rate of an algorithm for solving (1) is to define the primal function $\Phi(\mathbf{x}) = \max_{\mathbf{y} \in \mathcal{Y}} f(\mathbf{x}, \mathbf{y})$ and measure the first-order optimality in terms of the nonconvex problem $\min_{\mathbf{x} \in \mathcal{X}} \Phi(\mathbf{x})$. In this context, Thekumparampil et al. (2019) proposed the proximal dual implicit accelerated gradient (ProxDIAG) algorithm for smooth and nonconvex-strongly-concave minimax problems and proved that this algorithm finds an $\epsilon$-first-order stationary point of $\Phi$ with the rate of $\mathcal{O}(\epsilon^{-2})$.

Lin et al. (2020a) showed that a simple single-loop gradient descent ascent (GDA) method could obtain an $\epsilon$-first-order stationary point of $\Phi$ with $\mathcal{O}(\epsilon^{-2})$ gradients calls. Mahdavinia et al. (2022) also established the same iteration complexity by an extra-gradient method. Unfortunately, the first-order stationary points obtained by these algorithms cannot guarantee the local optimality since the objective function $f$ could be nonconvex on $\mathbf{x}$ and first-order stationarity includes suboptimal saddle points.

On the positive side, some recent literatures establish nonasymptotic convergence analysis for finding second-order stationary points. Luo et al. (2022) proposed a cubic Newton based method that can obtain an $(\epsilon, \sqrt{\epsilon})$-second-order stationary point in $\mathcal{O}(\epsilon^{-2})$ *Hessian-vector* oracle calls or $\mathcal{O}(\epsilon^{-1.5})$ *Hessian* oracle calls. Huang et al. (2022) obtained a gradient complexity of $\mathcal{O}(\epsilon^{-2})$ with a perturbed gradient descent-ascent algorithm. Yang et al. (2023) improved the complexity to $\mathcal{O}(\epsilon^{-1.75} \log^6 n)$ with a perturbed momentum-based method.

However, most of the existing analysis frameworks for minimax optimization are based on the requirement of Lipschitz smoothness. Though there are some works show the convergence for convex or weakly convex minimax problems without smoothness assumption (Rafique et al. (2022)), research on nonconvex minimax optimization with generalized smoothness is still limited. This drawback restricts the applications of minimax optimization algorithms because in some tasks the objective function does not satisfies Lipschitz smoothness such as distribution robust optimization (Yan et al., 2019; Levy et al., 2020; Jin et al., 2021) and phase retrieval (Drenth, 2007; Miao et al., 1999). Xian et al. (2024) conduct the convergence analysis of GDA and GDAmax under generalized smoothness and obtained a gradient complexity of $\mathcal{O}(\epsilon^{-2})$ for finding an $\epsilon$-first-order stationary point, but it is still open whether second-order stationary points could be obtained with generalized Lipschitz smoothness assumptions. This paper answers this question in the affirmative.

**Contributions.** In this paper, we propose a simple gradient-based accelerated methods, which have the following three advantages:

- We design a new algorithm named ANCGDA, which is the first algorithm to find a second-order stationary point in nonconvex-strongly-concave minimax optimization with generalized smoothness. We prove that it can obtain such points within $\mathcal{O}(\epsilon^{-1.75} \log n)$ number of gradient evaluations without Hessian-vector or Hessian oracle. Notably, this result is better than the state-of-the-art complexity results under Lipschitz smoothness assumption $\mathcal{O}(\epsilon^{-1.75} \log^6 n)$ in terms of the $\log n$ factor. The detailed comparison of existing nonconvex-strongly-concave minimax optimization algorithms is shown in Table 1.

- We proposed a second-order theory of generalized smoothness condition for minimax optimization and further conducted the new fundamental properties of the primal function $\Phi$ and $\mathbf{y}^*$ in Lemma 4.2 under the proposed second-order generalized smoothness condition, which is significantly important for controlling the hypergradient estimation error. Leveraging by this important properties, we develop a new convergence analysis framework for the second-order generalized smoothness minimax algorithm.

- We conduct a numerical experiment on domain adaptation task to validate the practical performance of our method. We show that ANCGDA consistently outperforms other minimax optimization algorithms.

## 2 RELATED WORK

**Nonconvex Minimax Optimization.** Recent years, many algorithms have been proposed for nonconvex minimax optimization under Lipschitz smoothness assumption. In Nonconvex-strongly-concave setting, Lin et al. (2020a) demonstrated the first non-asymptotic convergence of GDA to $\epsilon$ first-order stationary point of $\Phi(\boldsymbol{x})$, with the gradient complexity of $O(\kappa^2 \epsilon^{-2})$. Lin et al. (2020b) and Zhang et al. (2021) proposed triple loop algorithms achieving gradient complexity of $O(\sqrt{\kappa}\epsilon^{-2})$ by leveraging ideas from catalyst methods (adding $\alpha \|\boldsymbol{x} - \boldsymbol{x}_0\|^2$ to the objective function), and inexact proximal point methods, which nearly match the existing lower bound. (Li et al., 2021; Zhang et al., 2021; Ouyang & Xu, 2021) Approximating the inner loop optimization of catalyst idea by one step of GDA, Yang et al. (2022) developed a single loop algorithm called smoothed AGDA, which provably converges to $\epsilon$-stationary point, with gradient complexity of $O(\kappa\epsilon^{-2})$.

Table 1: Comparison of oracle complexity of nonconvex-strongly-concave minimax problems for finding first-order stationary points (FOSP) or second-order stationary points (SOSP). FO (First Order)-Generalized Smoothness and SO (Second Order)-Generalized Smoothness are defined in Definition 3.3 and 3.4. Note that the $\mathcal{O}(\epsilon^{-2})^*$ complexity of IMCN is computed with Hessian-vector oracles.

| Algorithm | Smoothness | FOSP | SOSP | Complexity |
|---|---|---|---|---|
| GDA (Lin et al., 2020a) | Lipschitz Smoothness | ✓ | ✕ | $\mathcal{O}(\epsilon^{-2})$ |
| Smoothed-GDA (Zhang et al., 2020b) | Lipschitz Smoothness | ✓ | ✕ | $\mathcal{O}(\epsilon^{-2})$ |
| GDmax (Jin et al., 2020) | Lipschitz Smoothness | ✓ | ✕ | $\mathcal{O}(\epsilon^{-2})$ |
| IMCN (Luo et al., 2022) | Lipschitz Hessian | ✓ | ✓ | $\mathcal{O}(\epsilon^{-2})^*$ |
| Perturbed GDmax (Huang et al., 2022) | Lipschitz Hessian | ✓ | ✓ | $\mathcal{O}(\epsilon^{-2})$ |
| PRAHGD (Yang et al., 2023) | Lipschitz Hessian | ✓ | ✓ | $\mathcal{O}(\epsilon^{-1.75} \log^6 n)$ |
| Generalized GDA (Xian et al., 2024) | FO-Generalized Smoothness | ✓ | ✕ | $\mathcal{O}(\epsilon^{-2})$ |
| **ANCGDA (This Work)** | **SO-Generalized Smoothness** | ✓ | ✓ | $\mathcal{O}(\epsilon^{-1.75} \log n)$ |

Compared to first-order methods, there has been significantly less research on the second-order methods for minimax optimization problems with global convergence rate estimation. However, a significant body of recent work shows that first-order stationary points cannot guarantee the local optimality in nonconvex-(strongly)concave settings and the global optimality in convex-concave settings. Lin et al. (2022) proposed newton-based methods and obtained global rates of convergence within $O(\epsilon^{-2/3})$ iterations using Hessian-vector information, matching the theoretically established lower bound in convex-concave settings. For nonconvex-strongly-concave settings, Luo et al. (2022) presented Minimax Cubic-Newton, obtaining a second-order stationary point of $\Phi$ with calling $O(\kappa^{1.5}\epsilon^{-1.5})$ times of Hessian oracles and $\tilde{O}(\kappa^2\epsilon^{-1.5})$ times of gradient oracles, while the inexact version obtaining a second-order stationary point with $\tilde{O}(\kappa^{1.5}\epsilon^{-2})$ Hessian-vector oracle calls and $\tilde{O}(\kappa^2\epsilon^{-1.5})$ gradient calls. Yang et al. (2023) proposed a Perturbed Restarted Accelerated HyperGradient Descent algorithm, improved the complexity bound to $\tilde{O}(\kappa^{1.75}\epsilon^{-1.75} \log^6 n)$ with only gradient iterations. But none of these algorithms are proved efficient under generalized smoothness assumption. To the best of our knowledge, we are the first work to study the convergence for finding second-order solutions in nonconvex-strongly-concave minimax optimization problems beyond bounded Lipschitz smoothness assumption.

**Generalized smoothness.** The convergence analysis of most existing minimax algorithms needs to assume the gradient or hessian is Lipschitz. However, such assumptions are fail to hold in an important class of neural networks such as recurrent neural networks (RNNs) (Elman, 1990), long-short-term memory networks (LSTMs) (Graves & Graves, 2012) and Transformers (Vaswani, 2017) which are shown to have unbounded smoothness (Pascanu, 2012; Zhang et al., 2019; Crawshaw et al., 2022). For minimization optimization, Zhang et al. (2019) proposed a relaxed smoothness assumption that bounds the Hessian by a linear function of the gradient norm, that is, a function $f$ is said to be $(l_0, l_1)$-smoothness if there exists some constants $l_0 > 0$ and $l_1 \geq 0$ such that

$$\|\nabla^2 f(\mathbf{x})\| \leq l_0 + l_1 \|\nabla f(\mathbf{x})\|, \quad \forall \mathbf{x} \in \mathbb{R}^n. \tag{2}$$

Under the same condition, Zhang et al. (2020a) considers momentum in the updates and improves the constant dependency of the convergence rate for SGD with clipping derived in Zhang et al. (2019). Qian et al. (2021) studies gradient clipping in incremental gradient methods, Zhao et al. (2021) studies stochastic normalized gradient descent, and Crawshaw et al. (2022) studies a generalized SignSGD method, under the $(l_0, l_1)$-smoothess condition. Reisizadeh et al. (2023) studies variance reduction for $(l_0, l_1)$-smooth functions. Wang et al. (2022) analyzes convergence of Adam and provides a lower bound which shows non-adaptive SGD may diverge. Li et al. (2024a) and Li et al. (2024b) further generalize the smoothness condition and analyze various methods under this condition through bounding the gradients along the trajectory:

$$\|\nabla f(\mathbf{x})\|^2 \leq 2(l_0 + 2l_1 \|\nabla f(\mathbf{x})\|) \cdot (f(\mathbf{x}) - f^*), \quad \forall \mathbf{x} \in \mathcal{X}, \tag{3}$$

if $f$ is $(l_0, l_1)$-smooth. Xie et al. (2024) show convergence beyond the first-order stationary condition for generalized smooth optimization. However, research on minimax optimization under gen-

eralized smoothness is few. Xian et al. (2024) prove that classic minimax optimization algorithms GDA, GDmax and their stochastic version can still converge to $\epsilon$-first-order stationary points under generalized smoothness condition and the complexity matches the Lipschitz smoothness counterparts. But it is still open whether second-order stationary points can be found in such conditions. We are thus led to ask the following question: *Is it possible to develop an effective method for finding **second-order** stationary points on nonconvex-strongly-concave minimax optimization under generalized smoothness and can such method matches the efficiency of accelerated algorithms for nonconvex minimization optimization?*

This paper answers this question in the affirmative. We further study the second-order generalized smoothness assumption for minimax optimization and present a gradient-based algorithm for finding second-order stationary points under generalized smoothness for nonconvex-strongly-concave minimax problem. We provide the convergence analysis and show that the proposed algorithm can find a second-order stationary point in $\mathcal{O}(\epsilon^{-1.75} \log n)$ iterations, which matches the state-of-the-art complexity results for nonconvex optimization under bounded Lipschitz smoothness assumption.

## 3 PRELIMINARIES

In this paper, we use $\langle \cdot, \cdot \rangle$ and $\| \cdot \|$ to denote the inner product and Euclidean norm. Aiming to solve minimax optimization problem 1, we introduce the following generalized smoothness assumptions. In Zhang et al. (2020a), the $(l_0, l_1)$-smooth assumption is defined as

**Definition 3.1** *A differentiable function $f : \mathbb{R}^n \to \mathbb{R}$ is $(l_0, l_1)$-smooth if $\|\nabla f(\mathbf{u}) - \nabla f(\mathbf{u}')\| \leq (l_0 + l_1 \|\nabla f(\mathbf{u})\|)\|\mathbf{u} - \mathbf{u}'\|$ for any $\|\mathbf{u} - \mathbf{u}'\| \leq R_l'$ with some constants $l_0 > 0$, $l_1 \geq 0$ and $R_l' > 0$.*

Definition 3.1 is a first-order smoothness condition relaxed from 2. When it comes to second-order condition, (Xie et al., 2024) proposed a second-order generalized smoothness assumption and interpret it from the perspective of the boundness of higher-order derivatives.

**Definition 3.2** *A twice-differentiable function $f : \mathbb{R}^n \to \mathbb{R}$ is $(\rho_0, \rho_1)$-Hessian continuous if $\|\nabla^2 f(\mathbf{u}) - \nabla^2 f(\mathbf{u}')\| \leq (\rho_0 + \rho_1 \|\nabla f(x)\|)\|\mathbf{u} - \mathbf{u}'\|$ for $\|\mathbf{u} - \mathbf{u}'\| \leq R_\rho'$ with some constants $\rho_0 > 0$, $\rho_1 \geq 0$ and $R_\rho' > 0$.*

Extending these assumptions to minimax optimization, we introduce the following first-order and second-order generalized smoothness conditions in Definition 3.3 and 3.4 respectively.

**Definition 3.3** *The function $f : \mathbb{R}^n \times \mathbb{R}^d \to \mathbb{R}$ is $(l_{\mathbf{x},0}, l_{\mathbf{x},1}, l_{\mathbf{y},0}, l_{\mathbf{y},1})$-smooth. i.e.*

$$\|\nabla_{\mathbf{x}} f(\mathbf{u}) - \nabla_{\mathbf{x}} f(\mathbf{u}')\| \leq (l_{\mathbf{x},0} + l_{\mathbf{x},1} \|\nabla_{\mathbf{x}} f(\mathbf{u})\|)\|\mathbf{u} - \mathbf{u}'\|$$
$$\|\nabla_{\mathbf{y}} f(\mathbf{u}) - \nabla_{\mathbf{y}} f(\mathbf{u}')\| \leq (l_{\mathbf{y},0} + l_{\mathbf{y},1} \|\nabla_{\mathbf{y}} f(\mathbf{u})\|)\|\mathbf{u} - \mathbf{u}'\|$$

*with $\mathbf{u} = (\mathbf{x}, \mathbf{y})$ and $\mathbf{u}' = (\mathbf{x}', \mathbf{y}')$ satisfy $\|\mathbf{u} - \mathbf{u}'\| \leq R_l$ for some constant $R_l > 0$.*

Hao et al. (2024) proved that (3.3) is equivalent to Definition 3.1 by letting $l_{\mathbf{x},0} = l_{\mathbf{y},0} = l_0/2$, $l_{\mathbf{x},1} = l_{\mathbf{y},1} = l_1/2$, $R_l = 1/\sqrt{2(l_{\mathbf{x},1}^2 + l_{\mathbf{y},1}^2)}$ and $R_l' = 1/l_1$. Inspired by the Hessian lipschitz condition for minimax optimization, we extend the concept of first-order generalized smoothness to second-order condition and propose the following generalized Hessian continuous condition.

**Definition 3.4** *The function $f : \mathbb{R}^n \times \mathbb{R}^d \to \mathbb{R}$ is $(\rho_{\mathbf{x},0}, \rho_{\mathbf{x},1}, \rho_{\mathbf{y},0}, \rho_{\mathbf{y},1}, \rho_{\mathbf{xy},0}, \rho_{\mathbf{xy},1})$-Hessian continuous. i.e.*

$$\|\nabla_{\mathbf{xx}}^2 f(\mathbf{u}) - \nabla_{\mathbf{xx}}^2 f(\mathbf{u}')\| \leq (\rho_{\mathbf{x},0} + \rho_{\mathbf{x},1} \|\nabla_{\mathbf{x}} f(\mathbf{u})\|)\|\mathbf{u} - \mathbf{u}'\|$$
$$\|\nabla_{\mathbf{yy}}^2 f(\mathbf{u}) - \nabla_{\mathbf{yy}}^2 f(\mathbf{u}')\| \leq (\rho_{\mathbf{y},0} + \rho_{\mathbf{y},1} \|\nabla_{\mathbf{y}} f(\mathbf{u})\|)\|\mathbf{u} - \mathbf{u}'\|$$
$$\|\nabla_{\mathbf{xy}}^2 f(\mathbf{u}) - \nabla_{\mathbf{xy}}^2 f(\mathbf{u}')\| \leq (\rho_{\mathbf{xy},0} + \rho_{\mathbf{xy},1} \min\{\|\nabla_{\mathbf{x}} f(\mathbf{u})\|, \|\nabla_{\mathbf{y}} f(\mathbf{u})\|\})\|\mathbf{u} - \mathbf{u}'\|$$

*with $\mathbf{u} = (\mathbf{x}, \mathbf{y})$ and $\mathbf{u}' = (\mathbf{x}', \mathbf{y}')$ satisfy $\|\mathbf{u} - \mathbf{u}'\| \leq R_\rho$ for some constant $R_\rho > 0$.*

*Remark:* Here, we assume that the objective function $f$ of minimax optimization is twice differentiable and has continuous second-order derivative, therefore we have $\|\nabla_{\mathbf{xy}}^2 f(\cdot)\| = \|\nabla_{\mathbf{yx}}^2 f(\cdot)\|$.

Also, with Eq.(2) it is easy to verify that

$$\|\nabla_{\mathbf{xy}}^2 f(\mathbf{u})\| \le \min\{M_0 + M_1\|\nabla_{\mathbf{x}} f(\mathbf{u})\|, M_0' + M_1'\|\nabla_{\mathbf{y}} f(\mathbf{u})\|\}$$

with some constants $M_0, M_1, M_0', M_1'$. Therefore, for simplicity we assume that

$$\|\nabla_{\mathbf{xy}}^2 f(\mathbf{u}) - \nabla_{\mathbf{xy}}^2 f(\mathbf{u}')\| \le (\rho_{\mathbf{xy},0} + \rho_{\mathbf{xy},1} \min\{\|\nabla_{\mathbf{x}} f(\mathbf{u})\|, \|\nabla_{\mathbf{y}} f(\mathbf{u})\|\})\|\mathbf{u} - \mathbf{u}'\|$$

Also, we proved that Definition 3.4 can be recovered to second-order generalized smoothness condition for minimization optimization (Definition 3.2) when $\rho_{\mathbf{x},0} = \rho_{\mathbf{y},0} = \rho_{\mathbf{xy},0} = \frac{\rho_0}{2\sqrt{2}}$ and $\rho_{\mathbf{x},1} = \rho_{\mathbf{y},1} = \rho_{\mathbf{xy},1} = \frac{\rho_1}{2\sqrt{2}}$. The details can be found in Lemma A.4.

Recall that the nonconvex-strongly-concave minimax problem in (1) is equivalent to minimizing a function $\Phi(\cdot) = \max_{\mathbf{y} \in \mathcal{Y}} f(\cdot, \mathbf{y})$. Huang et al. (2022) proved that in this context suppose $\Phi(\cdot)$ has a strict local minimum, then a strict local minimax point of (1) always exists and is equivalent to a strict local minimum of $\Phi$. A common notion of the stationarity of $\Phi$ is as follows.

**Definition 3.5** *A point $\mathbf{x} \in \mathbb{R}^n$ is said to be an $\epsilon$-first-order stationary point of function $\Phi(\cdot)$ if we have*

$$\|\nabla \Phi(x)\| \le c_1 \cdot \epsilon$$

*A point $\mathbf{x} \in \mathbb{R}^n$ is said to be an $(\epsilon, \sqrt{\epsilon})$-second-order stationary point of function $\Phi(\cdot)$ if we have*

$$\|\nabla \Phi(x)\| \le c_1 \cdot \epsilon, \quad \lambda_{\min}\left(\nabla^2 \Phi(x)\right) \ge -c_2 \cdot \sqrt{\epsilon}$$

*for some positive constants $c_1, c_2 > 0$.*

Most existing convergence theory for minimax problems focuses on finding $\epsilon$-first-order stationary point of $\Phi$ under Lipschitz smoothness or generalized smoothness assumptions. However, such results can be highly suboptimal saddle points because $\Phi$ can be nonconvex for nonconvex-strongly-concave minimax optimization. Therefore, in this paper, our goal is to find second-order stationary points of $\Phi$, with generalized smoothness assumptions.

## 4 THEORETICAL ANALYSIS

### 4.1 MAIN CHALLENGES

The main idea of the convergence analyses of the existing nonconvex minimax optimization algorithms is controlling the estimation error of maximizer $\delta_{\mathbf{y}_t} = \|\mathbf{y}_t - \mathbf{y}^*(\mathbf{x}_t)\|$ or approximating hypergradient $\nabla \Phi(\mathbf{x}) = \nabla_{\mathbf{x}} f(\mathbf{x}, \mathbf{y}^*(\mathbf{x}))$ and controlling the hypergradient estimation error $\delta_{\widehat{\Phi}} = \|\widehat{\nabla}\Phi(\mathbf{x}_t) - \nabla \Phi(\mathbf{x}_t)\| = \|\nabla_{\mathbf{x}} f(\mathbf{x}_t, \mathbf{y}_t) - \nabla_{\mathbf{x}} f(\mathbf{x}_t, \mathbf{y}^*(\mathbf{x}_t))\|$. With the classical Lipschitz smoothness assumption, both the two estimation error cannot blow up and can be easily controlled.

However, when the function has an unbounded smoothness (i.e. generalized smoothness) as illustrated in Section 3, the upper bound of estimation errors depend on the norm of the gradient of both the minimizer $\mathbf{x}$ and maximizer $\mathbf{y}$, with the term of $l_{\mathbf{x},1}\|\mathbf{y}_t - \mathbf{y}_t^*\|\|\nabla \Phi(\mathbf{x}_t)\|$, and can be arbitrarily large. This quantity is difficult to handle because $\|\nabla \Phi(\mathbf{x}_t)\|$ can be large, and it is difficult to decouple the two measurable term $\|\mathbf{y}_t - \mathbf{y}_t^*\|$ and $\|\nabla \Phi(\mathbf{x}_t)\|$. To address these challenges, some generalized version of GDA (Xian et al., 2024) have been proposed for nonconvex minimax optimization under generalized smoothness, with the idea to bound the gradient norm by the non-increasing function value for the convergence analyses. Unfortunately, when it comes to accelerated algorithm, both the gradient norm and the function value are no longer monotonically non-increasing. Therefore, existing minimax optimization algorithms are not guarantee to converge as long as to find a second-order stationary point in such problem settings that the objection function exhibits with unbounded smoothness.

---

**Algorithm 1:** Accelerated Negative Curvature Gradient Descent Ascent (ANCGDA)

1 **Input:** $\mathbf{x}_0, \mathbf{y}_{-1}, \mathbf{z}_0 = \mathbf{x}_0, \theta_{\mathbf{x}}, \theta_{\mathbf{y}}, B, r, K, \mathscr{T}$
2 **Initialize:** $k = 0, \zeta = 0$
3 **for** $t = 0, 1, 2, \ldots, T$ **do**
4    $\mathbf{y}_t = AGD(\mathbf{y}_{t-1}, -f(\mathbf{z}_t, \cdot), \eta_{\mathbf{y}}, \theta_{\mathbf{y}})$;
5    $\mathbf{x}_{t+1} = \mathbf{z}_t - \eta_{\mathbf{x}} \cdot (\nabla_{\mathbf{x}} f(\mathbf{z}_t, \mathbf{y}_t) - \zeta)$;
6    $\mathbf{z}_{t+1} = \mathbf{x}_{t+1} + (1 - \theta_x)(\mathbf{x}_{t+1} - \mathbf{x}_t)$;
7    $k = k + 1$;
8    **if** $\zeta = 0$ **then**
9      **if** $k \sum_{j=t-k+1}^{t} \|\mathbf{x}_{j+1} - \mathbf{x}_j\|^2 > B^2$ **then**
10       $\mathbf{z}_{t+1} = \mathbf{x}_{t+1}, k = 0$;            *# Reset k and Restart*
11      **else if** $k = K$ **then**
12       $\hat{t} = \operatorname{argmin}_{t-\lfloor \frac{K}{2} \rfloor + 1 \leq j \leq t} \|\mathbf{x}_{j+1} - \mathbf{x}_j\|^2$;
13       $\hat{\mathbf{z}} = \frac{1}{\hat{t}-t+K} \sum_{j=t-K+1}^{\hat{t}} \mathbf{z}_t$;
14       $\hat{\mathbf{y}} = AGD(\mathbf{y}_t, -f(\hat{\mathbf{z}}, \cdot), \eta_{\mathbf{y}}, \theta_{\mathbf{y}})$;
15       $\zeta = \nabla_{\mathbf{x}} f(\hat{\mathbf{z}}, \hat{\mathbf{y}})$;
16       $\mathbf{z}_{t+1} = \mathbf{x}_{t+1} = \hat{\mathbf{z}} + \xi$, where $\xi = Unif(\mathbb{B}_0(r))$;     *# Uniform Perturbation*
17       $k = 0$;
18    **else**
19      $\mathbf{z}_{t+1} = \hat{\mathbf{z}} + r \cdot \frac{\mathbf{z}_{t+1} - \hat{\mathbf{z}}}{\|\mathbf{z}_{t+1} - \hat{\mathbf{z}}\|}, \mathbf{x}_{t+1} = \hat{\mathbf{z}} + r \cdot \frac{\mathbf{x}_{t+1} - \hat{\mathbf{z}}}{\|\mathbf{z}_{t+1} - \hat{\mathbf{z}}\|}$;
20      **if** $k = \mathscr{T}$ **then**
21       $\hat{\mathbf{e}} = \frac{\mathbf{x}_{t+1} - \hat{\mathbf{z}}}{\|\mathbf{x}_{t+1} - \hat{\mathbf{z}}\|}$;
22       $\mathbf{x}_{t+1} = \hat{\mathbf{z}} - \frac{1}{4}\sqrt{\frac{\epsilon}{\rho}} \cdot \hat{\mathbf{e}}$;        *# One-step Descent along NC Direction*
23       $\mathbf{z}_{t+1} = \mathbf{x}_{t+1}, \zeta = 0, k = 0$;

---

**Algorithm 2:** AGD

1 **Input:** $\mathbf{y}_{t-1}, h(\cdot), \theta_{\mathbf{y}}, \eta_{\mathbf{y}}$
2 **Initialize:** $\hat{\mathbf{y}}_t^0 = \mathbf{y}_t^0 = \mathbf{y}_{t-1}$
3 **for** $d = 0, 1, 2, \ldots, D - 1$ **do**
4    $\mathbf{y}_t^{d+1} = \hat{\mathbf{y}}_t^d - \eta_{\mathbf{y}} \nabla h(\hat{\mathbf{y}}_t^d)$;
5    $\hat{\mathbf{y}}_t^{d+1} = \mathbf{y}_t^{d+1} + (1 - \theta_y)(\mathbf{y}_t^{d+1} - \mathbf{y}_t^d)$;
6 **Output:** $\mathbf{y}_t^D$

---

### 4.2 ALGORITHM DESIGN

We now introduce our algorithm for nonconvex-strongly-concave minimax optimization under generalized smoothness. Let $\mathbf{x}_0$ and $\mathbf{y}_{-1}$ be the initial values in Algorithm 1. First, in each iteration, the algorithm runs a Nesterov's classical Accelerated Gradient Descent (AGD) algorithm subroutine, as shown in Algorithm 2, to solve the strongly-convex generalized smoothness subproblem $\mathbf{y}^\star(\cdot) = \operatorname{argmax}_{\mathbf{y} \in \mathbb{R}^d} f(\cdot, \mathbf{y})$ and obtain the estimation of maximizer with the output $\mathbf{y}_t = \mathbf{y}_t^D \approx \mathbf{y}^*(\mathbf{z}_t)$ after $D = \mathcal{O}(\log(1/\epsilon))$ iterations in Algorithm 2, therefore control the hypergradient estimation error shown in Lemma 4.4. Then, the algorithm runs following iterations to update $\mathbf{x}_t$ with $\mathbf{y}_t$:

$$\mathbf{x}_{t+1} = \mathbf{z}_t - \eta_{\mathbf{x}} \cdot (\nabla_{\mathbf{x}} f(\mathbf{z}_t, \mathbf{y}_t) - \zeta), \quad \mathbf{z}_{t+1} = \mathbf{x}_{t+1} + (1 - \theta_x)(\mathbf{x}_{t+1} - \mathbf{x}_t), \quad (4)$$

where the variable $\zeta$ is initialized to be $\mathbf{0}$, which will be introduced later, so that these iterations become Nesterov's classical AGD procedure. Specifically, inspired by Li & Lin (2022), we use a counter variable $k$ to denote the iteration number in a round before the conditions on Line 9, Line 11 or Line 20 (after the uniform perturbation is added) triggers. To simplify the description, we define

an *epoch* to be a round from $k = 0$ to the iteration that triggers one of these conditions and resets $k$ to 0.

As the condition on Line 9 triggers, we simply set $\mathbf{z}_{t+1}$ equal to $\mathbf{x}_{t+1}$ and reset $k$. In such epoch the algorithm makes progress in decreasing the function value of $\Phi$ for at least $\mathscr{F} = \mathcal{O}(\sqrt{\epsilon^3/\rho})$, described in Lemma 4.5. If not, Line 11 triggers when $k = K = \mathcal{O}(\epsilon^{-1/4})$ as the algorithm achieve enough decrease. In that case, the gradient $\|\widehat{\nabla}\Phi(\hat{\mathbf{z}})\|$ is small, as shown in Lemma 4.6, then we denote $\zeta = \widehat{\nabla}\Phi(\hat{\mathbf{z}}) = \nabla_{\mathbf{x}}f(\hat{\mathbf{z}}, \hat{\mathbf{y}})$ to be the estimation of hypergradient $\nabla\Phi(\hat{\mathbf{z}})$ and add a uniform perturbation on that $\hat{\mathbf{z}}$. After that, with the negative curvature (NC) finding technique, the algorithm start finding a negative curvature direction in the following $\mathscr{T} = \mathcal{O}(\epsilon^{-1/4}\log n)$ iterations, then take a one-step descent along the found NC direction $\hat{\mathbf{e}}$. With possibility the point $\mathbf{x}_{t+1}$ in that iteration will be a second-order approximate stationary point, as shown in Lemma 4.7 and 4.8. After the one-step descent we reset $\zeta$, $k$ and set $\mathbf{z}_{t+1}$ equal to $\mathbf{x}_{t+1}$ then continue to the next epoch. Finally at least one of the iterations $\mathbf{x}_t$ will be a second-order stationary point with possibility at least $1 - \delta$ with some constant $\delta \in (0, 1]$. The main result is shown in Theorem 4.3.

### 4.3 MAIN RESULTS

In this section, we present our main results on complexity bounds for Algorithm 1 in terms of gradient evaluations. First, we proposed the following assumptions for the nonconvex-strongly-concave minimax optimization (1).

**Assumption 4.1** *The objective function $f(\mathbf{x}, \mathbf{y})$ satisfies the following assumptions*

1. *$f(\mathbf{x}, \mathbf{y})$ is $(l_{\mathbf{x},0}, l_{\mathbf{x},1}, l_{\mathbf{y},0}, l_{\mathbf{y},1})$-smooth with $(\rho_{\mathbf{x},0}, \rho_{\mathbf{x},1}, \rho_{\mathbf{y},0}, \rho_{\mathbf{y},1}, \rho_{\mathbf{xy},0}, \rho_{\mathbf{xy},1})$-Hessian.*

2. *$f(\mathbf{x}, \cdot)$ is $\mu$-strongly concave while $f(\cdot, \mathbf{y})$ is not necessary convex.*

3. *The function $\Phi(\mathbf{x}) \triangleq \max_{\mathbf{y} \in \mathbb{R}^m} f(\mathbf{x}, \mathbf{y})$ is lower bounded.*

These assumptions are standard prerequisites for the convergence analysis of nonconvex-strongly-concave minimax optimization. Then, we present a key technical lemma on the structure of the function $\Phi(\cdot)$ and $\mathbf{y}^\star(\cdot)$ and their generalized smoothness properties. Define

$$\Phi(\cdot) = \max_{\mathbf{y} \in \mathbb{R}^d} f(\cdot, \mathbf{y}), \quad \mathbf{y}^\star(\cdot) = \mathrm{argmax}_{\mathbf{y} \in \mathbb{R}^d} f(\cdot, \mathbf{y})$$

We proposed the following lemma:

**Lemma 4.2** *Under Assumption 4.1, denote*

$$G = \max\left\{ \sqrt{2\mathcal{L} \cdot (\Phi(\mathbf{x}_0) - \Phi^*)}, 2\|\nabla\Phi(\mathbf{x}_0)\| \right\}, \tag{5}$$

*where $\Phi^*$ denotes $\min_{\mathbf{x}} \Phi(\mathbf{x})$ and $\mathcal{L} = l_{\Phi,0} + 2l_{\Phi,1}G$ is the effective smoothness constant of $\Phi$. Denote the Euclidean ball with radius $R$ centered at $x$ as $\mathcal{B}(x, R)$, then for any $\mathbf{x}$, $\mathbf{x}'$ such that*

$$\mathbf{x}, \mathbf{x}' \in \mathcal{B}\left(\mathbf{x}_0, \frac{G\mu}{\mathcal{L}(\mu + l_{\mathbf{y},0})}\right) \tag{6}$$

*the function $\Phi : \mathbb{R}^n \mapsto \mathbb{R}$ and $\mathbf{y}^\star(\cdot) : \mathbb{R}^n \mapsto \mathbb{R}^d$ satisfies*

1. *$\mathbf{y}^*(\mathbf{x})$ is well-defined and $\frac{l_{\mathbf{y},0}}{\mu}$-Lipschitz continous.*

2. *The derivative $\left\|\nabla^2_{\mathbf{xy}}f(\mathbf{x}, \mathbf{y})\right\|$ is bounded. i.e. $\|\nabla^2_{\mathbf{xy}}f(\mathbf{x}, \mathbf{y})\| \leq M$.*

3. *$\Phi(x)$ is $(l_{\phi,0}, l_{\phi,1})$-smooth, i.e.*

$$\|\nabla\Phi(\mathbf{x}) - \nabla\Phi(\mathbf{x}')\| \leq (l_{\Phi,0} + l_{\Phi,1}\|\nabla\Phi(\mathbf{x}')\|) \|\mathbf{x} - \mathbf{x}'\|$$

*where $l_{\Phi,0}, l_{\Phi,1}$ are defined as*

$$l_{\Phi,0} = \left(1 + \frac{l_{\mathbf{y},\mathbf{0}}}{\mu}\right) l_{\mathbf{x},0}, \quad l_{\Phi,1} = \left(1 + \frac{l_{\mathbf{y},\mathbf{0}}}{\mu}\right) l_{\mathbf{x},1}$$

4. $\Phi(x)$ has $(\rho_{\phi,0}, \rho_{\phi,1})$-continuous Hessian, i.e.

$$\|\nabla^2\Phi(\mathbf{x}') - \nabla^2\Phi(\mathbf{x})\| \leq (\rho_{\Phi,0} + \rho_{\Phi,1}\|\nabla\Phi(\mathbf{x}')\|)\|\mathbf{x} - \mathbf{x}'\|$$

where

$$\rho_{\phi,0} = \left(1 + \frac{l_{\mathbf{y},0}}{\mu}\right)(\rho_{\mathbf{x},0} + (\mu^{-1}M\sqrt{\rho_{\mathbf{y},0}} + \frac{\rho_{\mathbf{xy},0}}{\sqrt{\rho_{\mathbf{y},0}}})^2),$$

$$\rho_{\phi,1} = \left(1 + \frac{l_{\mathbf{y},0}}{\mu}\right)\rho_{\mathbf{x},1}$$

Lemma 4.2 proposed the generalized smoothness properties of function $\Phi$ and $\mathbf{y}^*$ in terms of the smoothness constants of the objective function $f$, under which we can bound the hypergradient estimation error, which will be mentioned in Lemma 4.4.

Denoting $\Delta_\Phi = \Phi(\mathbf{x}_0) - \min_\mathbf{x}\Phi(\mathbf{x})$, we summarize our results for Algorithm 1 in the following theorem.

**Theorem 4.3** *Under Assumption 4.1, Denote $G$, $\mathcal{L}$ as (5), $G_\mathbf{y}$, $\mathcal{L}_\mathbf{y}$ as (23), run Algorithm 1 with $\delta \in (0, 1]$ and $\epsilon \leq \min\left\{\frac{\mathcal{L}_\mathbf{x}^2}{16\rho}, \frac{4G_\mathbf{x}^2\rho}{\mathcal{L}_\mathbf{x}^2}, \frac{G_\mathbf{y}^2\rho}{\mathcal{L}_\mathbf{y}^2}\right\}$, where $\rho = \rho_{\Phi,0} + 2\rho_{\Phi,1}G$ is the effective hessian smoothness constant of $\Phi$. If we choose $B = \sqrt{\frac{\epsilon}{\rho}}$, $\eta_\mathbf{x} \leq \frac{1}{4\mathcal{L}}$, $\theta = (\eta_\mathbf{x}^2\rho\epsilon)^{1/4} < 1$, $K = \frac{1}{\theta}$, $D = \mathcal{O}\left(\sqrt{\frac{\mathcal{L}_\mathbf{y}}{\mu}}\log(1/\epsilon)\right)$, $\eta_\mathbf{y}$, $\theta_\mathbf{y}$ as (38), $r$, $\mathcal{T}$, $\delta_0$ as (79), Algorithm 1 satisfies that at least one of the iterations $\mathbf{x}_t$ will be an $(\epsilon, \sqrt{\epsilon})$-second-order approximate stationary point in*

$$T = D \cdot \mathcal{O}\left(\frac{\Delta_\Phi}{\epsilon^{1.75}} \cdot \log n\right) = \mathcal{O}\left(\frac{\Delta_\Phi}{\epsilon^{1.75}} \cdot \log n\right)$$

*iterations, with probability at least $1 - \delta$.*

Theorem 4.3 says that after designated number of iterations, which is polylogarithmic in dimension of $\mathbf{x}$, at least one of the iterates is an $(\epsilon, \sqrt{\epsilon})$-second-order approximate stationary point. The complexity results $\mathcal{O}\left(\frac{\Delta_\Phi \log n}{\epsilon^{1.75}}\right)$, which improves the state-of-the-art complexity results by a polynomial factor of $\mathcal{O}(\log^5 n)$ in nonconvex-strongly-concave minimax optimization even under Lipschitz smoothness condition. The detailed proof is deferred to Appendix E.

### 4.4 PROOF SKETCH

In this subsection, we present an overview of the proof of Theorem 4.3. Lemma 4.4 presents the hypergradient estimation error for every maximizer estimation subproblem conduct by Algorithm 2. Lemma 4.5 is the key property of monotonic decrease for the function value of $\Phi$ in each round and Lemma 4.6 shows that when the condition on Line 11 of Algorithm 1 triggers, a first-order approximate stationary point can be found, which leads to the negative curvature direction finding process on Lemma 4.7. Lemma 4.8 demonstrates that with a one-step descent along the found negative curvature direction the function value guarantee to decrease. Complete details can be found in the appendix.

#### 4.4.1 CONTROL FOR HYPERGRADIENT ESTIMATION

**Lemma 4.4** *Denote $\widehat{\nabla}\Phi(\mathbf{x}_t) = \nabla_\mathbf{x}f(\mathbf{x}_t, \mathbf{y}_t)$. Let $\iota$ be a constant with $\iota = c \cdot \log(\frac{1}{\delta_0}\sqrt{\frac{n}{\pi\rho_\Phi}}) > 1$. Running Algorithm 1 with the parameters setting in Theorem 4.3, after each AGD subroutine of Algorithm 2 with parameter $\eta_\mathbf{y}$, $\theta_\mathbf{y}$ in (38), the estimation error $\delta_{\widehat{\Phi}} = \|\nabla\Phi(\mathbf{x}_t) - \widehat{\nabla}\Phi(\mathbf{x}_t)\|$ can be bounded as*

$$\|\nabla\Phi(\mathbf{x}_t) - \widehat{\nabla}\Phi(\mathbf{x}_t)\| \leq \min\left\{\frac{1}{4}, \frac{1}{\iota^2 2^{6-\iota}}\right\} \cdot \epsilon \tag{7}$$

Lemma 4.4 controls the error in the hypergradient estimator by estimate the maximizer $\mathbf{y}^*(\mathbf{x})$ with the AGD subroutine in Algorithm 2. With the bounded hypergradient estimation error, we can show

the function value of $\Phi$ decrease for the iterations between two successive triggers of the condition on Line 9 of Algorithm 1. Then we introduce the following lemmas to show the algorithm make progress for decreasing the function value of $\Phi$ in every epoch until the gradient is small enough.

### 4.4.2 MONOTONIC DESCENT

**Lemma 4.5** *Running Algorithm 1 with parameters setting in Theorem 4.3. When the condition on Line 9 triggers, denote $t_{\mathcal{K}}$ to be the iteration number, $\mathcal{K}$ to be the value of k on that iteration and $t_0 = t_{\mathcal{K}} - \mathcal{K} + 1$. In each epoch of Algorithm 1 where the Line 9 triggers, we have*

$$\Phi(\mathbf{x}_{t_{\mathcal{K}}+1}) - \Phi(\mathbf{x}_{t_0}) \leq -\frac{51}{64}\sqrt{\frac{\epsilon^3}{\rho}}$$

**Lemma 4.6** *Running Algorithm 1 with parameters setting in Theorem 4.3. In the epoch that the condition on Line 11 triggers, the point $\hat{\mathbf{z}}$ in Line 13 satisfies $\|\nabla\Phi(\hat{\mathbf{z}})\| \leq \mathcal{O}(\epsilon)$.*

See Appendix C for more details. We see that if the function value of $\Phi$ does not decrease much (when the condition on Line 11 triggers), the gradient is guaranteed to be small. Then as shown in Lemma 4.7 and 4.8 after the following $\mathscr{T}$ iterations a negative curvature direction will be found.

### 4.4.3 ESCAPE SADDLE POINT

**Lemma 4.7** *Running Algorithm 1 with parameters setting in Theorem 4.3. For the point $\hat{\mathbf{z}}$ satisfying $\lambda_{\min}\left(\nabla^2\Phi(\hat{\mathbf{z}})\right) \leq -\sqrt{\rho\epsilon}$, adding an uniform perturbation in Line 16, the unit vector $\hat{\mathbf{e}}$ in Line 21 obtained after $\mathscr{T}$ iterations satisfies*

$$\mathbb{P}\left(\hat{\mathbf{e}}^T\mathcal{H}(\hat{\mathbf{z}})\hat{\mathbf{e}} \leq -\sqrt{\rho\epsilon}/4\right) \geq 1 - \delta_0,$$

*where $\rho = \rho_{\Phi,0} + 2\rho_{\Phi,1}G$ denotes the effective Hessian smoothness constant of $\Phi$.*

Here, we take the definition of negative curvature direction from Xu et al. (2018), which implies that for a non-degenerate saddle point $\mathbf{x}$ of a function $f(\mathbf{x})$ with $\|\nabla f(\mathbf{x})\| \leq \epsilon$ and $\lambda_{\min}\left(\nabla^2 f(\mathbf{x})\right) \leq -\gamma$, the negative curvature direction $\mathbf{v}$ satisfies $\|\mathbf{v}\| = 1$ and $\mathbf{v}^\top\nabla^2 f(\mathbf{x})\mathbf{v} \leq -c\gamma$. Taking $c = \frac{1}{4}$ and $\gamma = \sqrt{\rho\epsilon}$ yields that the obtained $\hat{\mathbf{e}}$ is a NC direction.

**Lemma 4.8** *Running Algorithm 1 with parameters setting in Theorem 4.3. For each $\hat{\mathbf{z}}$ if there exists a unit vector $\hat{\mathbf{e}}$ satisfying $\hat{\mathbf{e}}^T\mathcal{H}(\hat{\mathbf{z}})\hat{\mathbf{e}} \leq -\frac{\sqrt{\rho\epsilon}}{4}$ where $\mathcal{H}$ stands for the Hessian matrix of function $\Phi$, the following inequality holds*

$$\Phi\left(\hat{\mathbf{z}} - \frac{1}{4}\sqrt{\frac{\epsilon}{\rho}} \cdot \hat{\mathbf{e}}\right) \leq \Phi(\hat{\mathbf{z}}) - \frac{1}{384}\sqrt{\frac{\epsilon^3}{\rho}}$$

Lemma 4.7 and 4.8 demonstrate that Algorithm 1 can compute the negative curvature direction, discribed by a unit vector $\hat{\mathbf{e}}$, via the $\mathscr{T}$ iterations after a unit perturbation is added on Line 11, as the negative curvature finding subroutine. Then after a one-step descent along the found direction, the function value of $\Phi$ is guaranteed to decrease. We give the full details in Appendix D.

## 5 EXPERIMENTS

**Domain adaptation.** We follow Luo et al. (2022) and optimize Domain-Adversarial Neural Network (Ajakan et al., 2014) with two different source datasets, SVHN (Netzer et al., 2011) and MNIST-M (Goodfellow et al., 2014), and test on target domain dataset MNIST (LeCun et al., 1998). The DANN aims to solve the following nonconvex-concave minimax problem

$$\min_{[\mathbf{x}_1;\mathbf{x}_2]\in\mathbb{R}^{d_x}} \max_{\mathbf{y}\in\mathbb{R}^{d_y}} L_1(\mathbf{x}_1,\mathbf{x}_2) - \alpha \cdot L_2(\mathbf{x}_1,\mathbf{y}), \tag{8}$$

where $L_1(\mathbf{x}_1,\mathbf{x}_2) = \frac{1}{N_{\mathcal{S}}}\sum_{i=1}^{N_{\mathcal{S}}} l\left(\mathbf{x}_2; \Phi\left(\mathbf{x}_1; \mathbf{a}_i^{\mathcal{S}}\right), b_i^{\mathcal{S}}\right)$ is the loss of supervised learning and

$$L_2(\mathbf{x}_1,\mathbf{y}) = \frac{1}{N_{\mathcal{S}}}\sum_{i=1}^{N_{\mathcal{S}}} D_{\mathcal{S}}\left(h\left(\mathbf{y}; \Phi\left(\mathbf{x}_1; \mathbf{a}_i^{\mathcal{S}}\right)\right)\right) - \frac{1}{N_{\mathcal{T}}}\sum_{i=1}^{N_{\mathcal{T}}} D_{\mathcal{T}}\left(h\left(\mathbf{y}; \Phi\left(\mathbf{x}_1; \mathbf{a}_i^{\mathcal{T}}\right)\right)\right) + \lambda\|\mathbf{y}\|^2 \tag{9}$$

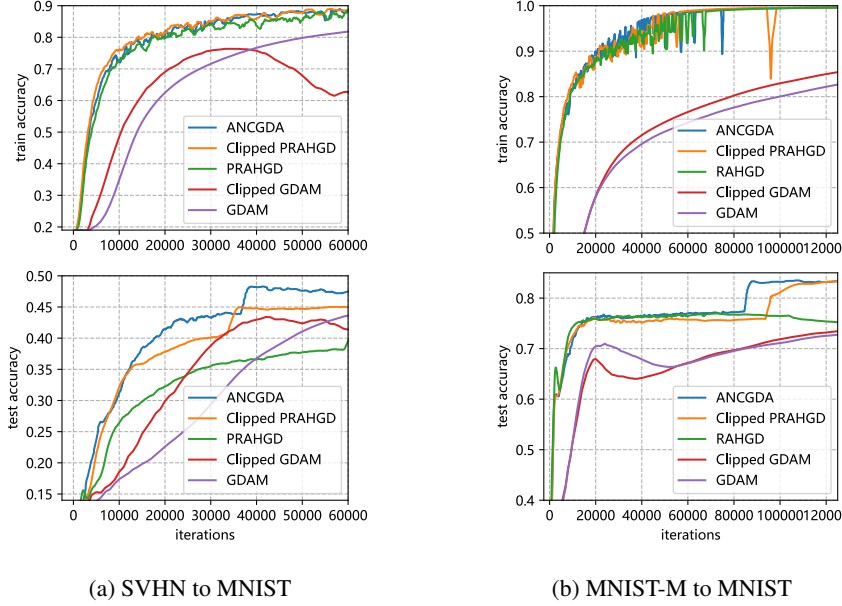

(a) SVHN to MNIST          (b) MNIST-M to MNIST

Figure 1: Comparison of various minimax optimization algorithms with train accuracy and test accuracy on two different domain adaptation tasks: (a) SVHN as source datasets to MNIST as target datasets and (b) MNIST-M as source datasets to MNIST as target datasets.

is the domain classification loss, where the source domain dataset is $\mathcal{S} = \{(\mathbf{a}_i^S, b_i^S)\}_{i=1}^{N_S}$ where $\mathbf{a}_i^S$ is the feature vector of the $i$-th sample and $b_i^S$ is the corresponding label. The target domain dataset $\mathcal{T} = \{\mathbf{a}_i^\mathcal{T}\}_{i=1}^{N_\mathcal{T}}$ only contains features. Here $\Phi$ is a single-layer neural network as the feature extractor with the size of $(28 \times 28) \times 200$ with parameter $\mathbf{x}_1$ and $l$ is a two-layer neural network as the domain classifier with the size of $200 \times 20 \times 10$ with parameter $\mathbf{x}_2$, followed by a cross entropy loss. For the logistic loss functions for $L_2$, we let $h(\mathbf{y}; \mathbf{z}) = 1/(1 + \exp(-\mathbf{y}^\top \mathbf{z}))$, $D_S(z) = 1 - \log(z)$ and $D_\mathcal{T}(z) = \log(1 - z)$. Note that $\lambda$ makes the function $L_2$ strongly-concave/concave in terms of discriminator parameters.

Performance on the value of train accuracy and test accuracy is depicted in Figure 1a and 1b, in comparison to GDAM, Clipped GDAM, PRAHGD Yang et al. (2023) and Clipped PRAHGD via oracle calls. For each algorithm, we choose the best learning rates $\eta_\mathbf{x}$, $\eta_\mathbf{y}$ in $[0.001, 1]$ and momentum $\theta_\mathbf{x}$, $\theta_\mathbf{y}$ in $[0.01, 0.5]$ that make it converge by grid search. For the other hyperparameters for ANCGDA, PRAHGD and Clipped PRAHGD, we choose $r = 0.04$, $K = 30$, $\mathscr{T} = 10$ for both the source domain dataset while setting $B = 10$ for SVHN as source dataset and $B = 7$ for MNIST-M.

It can be seen that ANCGDA outperforms standard GDAM and PRAHGD as a representative of non-Clipped algorithm family. Furthermore, it is clear that ANCGDA performs the best in convergence speed and overall performance among all the five algorithms.

## 6   CONCLUSION

In this paper, we proposed a new algorithm named ANCGDA for nonconvex-strongly-concave minimax optimization under generalized smoothness. We investigate the convergence analysis of the propose algorithm and proved that ANCGDA requires $\mathcal{O}(\epsilon^{-1.75} \log n)$ gradient oracles to obtain a $(\epsilon, \sqrt{\epsilon})$-second-order approximate stationary point, which matches the state-of-art single-level non-convex minimization conplexity results under the Lipschitz smoothness assumption and is better than all the existing complexity results for nonconvex-strongly-concave minimax optimization with Lipschitz smoothness or generalized smoothness. We conduct a numerical experiment of domain adaptation task to validate the practical performance of our method.

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

# A    TECHNICAL LEMMAS

**Lemma A.1** *(Li et al., 2024a) If $f$ is $(l_0, l_1)$-smooth, for any $\mathbf{x}_1, \mathbf{x}_2$ that satisfy $\|\nabla f(\mathbf{x}_1)\| \leq G$, $\|\nabla f(\mathbf{x}_2)\| \leq G$ and $\|\mathbf{x}_1 - \mathbf{x}_2\| \leq \frac{G}{\mathcal{L}}$ we have*

$$\|\nabla f(\mathbf{x}_1) - \nabla f(\mathbf{x}_2)\| \leq \mathcal{L}\|\mathbf{x}_1 - \mathbf{x}_2\|, \quad f(\mathbf{x}_1) \leq f(\mathbf{x}_2) + \langle \nabla f(\mathbf{x}_2), \mathbf{x}_1 - \mathbf{x}_2 \rangle + \frac{\mathcal{L}}{2}\|\mathbf{x}_1 - \mathbf{x}_2\|^2 \quad (10)$$

*where $\mathcal{L} := l_0 + 2l_1 G$ denotes the effective smoothness constant.*

**Lemma A.2** *(Li et al., 2024a) Suppose $f$ is $(l_0, l_1)$-smooth. If $f(\mathbf{x}) - f^* \leq F$ for some $\mathbf{x}$ and $F \geq 0$, denoting $G := \sup\{u \geq 0 \mid u^2 \leq 2(l_0 + 2l_1 u) \cdot F\}$, then they satisfy $G^2 = 2(l_0 + 2l_1 G) \cdot F$ and we have $\|\nabla f(\mathbf{x})\| \leq G < \infty$.*

**Lemma A.3** *If $f$ is $(\rho_0, \rho_1)$-Hessian continuous, for any $\mathbf{x}_1, \mathbf{x}_2$ that satisfy $\|\nabla f(\mathbf{x}_1)\| \leq G$, $\|\nabla f(\mathbf{x}_2)\| \leq G$ and $\|\mathbf{x}_1 - \mathbf{x}_2\| \leq \frac{G}{\rho}$ we have*

$$f(\mathbf{x}_1) \leq f(\mathbf{x}_2) + \langle \nabla f(\mathbf{x}_2), \mathbf{x}_1 - \mathbf{x}_2 \rangle + \frac{1}{2}(\mathbf{x}_1 - \mathbf{x}_2)^T \nabla^2 f(\mathbf{x}_2)(\mathbf{x}_1 - \mathbf{x}_2) + \frac{\rho}{6}\|\mathbf{x}_1 - \mathbf{x}_2\|^3 \quad (11)$$

*where $\rho := \rho_0 + 2\rho_1 G$ denotes the effective Hessian smoothness constant.*

*Proof.* With Definition 3.2 and the definition of $G$ and $\rho$ we have

$$\|\nabla^2 f(\mathbf{x}_1) - \nabla^2 f(\mathbf{x}_2)\| \leq \rho\|\mathbf{x}_1 - \mathbf{x}_2\|$$

Indeed, we have

$$\left\|\nabla f(\mathbf{x}_2) - \nabla f(\mathbf{x}_1) - \nabla^2 f(\mathbf{x}_1)(\mathbf{x}_2 - \mathbf{x}_1)\right\|$$
$$= \left\|\int_0^1 \left[\nabla^2 f(\mathbf{x}_1 + \tau(\mathbf{x}_2 - \mathbf{x}_1)) - \nabla^2 f(\mathbf{x}_1)\right](\mathbf{x}_2 - \mathbf{x}_1)d\tau\right\|$$
$$\leq \rho\|\mathbf{x}_2 - \mathbf{x}_1\|^2 \int_0^1 \tau d\tau = \frac{\rho}{2}\|\mathbf{x}_2 - \mathbf{x}_1\|^2$$

Therefore,

$$\left|f(\mathbf{x}_2) - f(\mathbf{x}_1) - \langle \nabla f(\mathbf{x}_1), \mathbf{x}_2 - \mathbf{x}_1 \rangle - \frac{1}{2}\left\langle \nabla^2 f(\mathbf{x}_1)(\mathbf{x}_2 - \mathbf{x}_1), \mathbf{x}_2 - \mathbf{x}_1 \right\rangle\right|$$
$$= \left|\int_0^1 \int_0^\tau \left\langle \left(\nabla^2 f(\mathbf{x}_1 + \alpha(\mathbf{x}_2 - \mathbf{x}_1))\right)(\mathbf{x}_2 - \mathbf{x}_1), \mathbf{x}_2 - \mathbf{x}_1 \right\rangle d\alpha d\tau - \int_0^1 \int_0^\tau \left\langle \nabla^2 f(\mathbf{x}_1)(\mathbf{x}_2 - \mathbf{x}_1), \mathbf{x}_2 - \mathbf{x}_1 \right\rangle d\alpha d\tau\right|$$
$$= \left|\int_0^1 \int_0^\tau \left\langle \left(\nabla^2 f(\mathbf{x}_1 + \alpha(\mathbf{x}_2 - \mathbf{x}_1)) - \nabla^2 f(\mathbf{x}_1)\right)(\mathbf{x}_2 - \mathbf{x}_1), \mathbf{x}_2 - \mathbf{x}_1 \right\rangle d\alpha d\tau\right|$$
$$\leq \int_0^1 \int_0^\tau \left\|\nabla^2 f(\mathbf{x}_1 + \alpha(\mathbf{x}_2 - \mathbf{x}_1)) - \nabla^2 f(\mathbf{x}_1)\right\| d\alpha d\tau \cdot \|\mathbf{x}_2 - \mathbf{x}_1\|^2$$
$$\leq \int_0^1 \int_0^\tau \frac{\rho}{2}\alpha\|\mathbf{x}_2 - \mathbf{x}_1\| d\alpha d\tau \cdot \|\mathbf{x}_2 - \mathbf{x}_1\|^2$$
$$= \frac{\rho}{6}\|\mathbf{x}_2 - \mathbf{x}_1\|^3,$$

which complete the proof.

**Lemma A.4** *When $\rho_{\mathbf{x},0} = \rho_{\mathbf{y},0} = \rho_{\mathbf{xy},0} = \frac{\rho_0}{2\sqrt{2}}$ and $\rho_{\mathbf{x},1} = \rho_{\mathbf{y},1} = \rho_{\mathbf{xy},1} = \frac{\rho_1}{2\sqrt{2}}$, Definition 3.4 implies that for any $\mathbf{u}, \mathbf{u}'$ such that $\|\mathbf{u} - \mathbf{u}'\| \leq R_\rho$, we have*

$$\|\nabla^2_{\mathbf{u}} f(\mathbf{u}) - \nabla^2_{\mathbf{u}} f(\mathbf{u}')\| \leq (\rho_0 + \rho_1\|\nabla_{\mathbf{u}} f(\mathbf{u})\|)\|\mathbf{u} - \mathbf{u}'\| \quad (12)$$

*In other words, $(\rho_{\mathbf{x},0}, \rho_{\mathbf{x},1}, \rho_{\mathbf{y},0}, \rho_{\mathbf{y},1}, \rho_{\mathbf{xy},0}, \rho_{\mathbf{xy},1})$-Hessian smoothness can recover to the second-order generalized smoothness assumption for single-level optimization (Assumption 3.2).*

*Proof.* Let $R_\rho = 1/\sqrt{2(\rho_{\mathbf{x},1}^2 + \rho_{\mathbf{y},1}^2 + 2\rho_{\mathbf{xy},1}^2)}$, with $\rho_{\mathbf{x},0} = \rho_{\mathbf{y},0} = \rho_{\mathbf{xy},0} = \frac{\rho_0}{2\sqrt{2}}$ and $\rho_{\mathbf{x},1} = \rho_{\mathbf{y},1} = \rho_{\mathbf{xy},1} = \frac{\rho_1}{2\sqrt{2}}$. Definition 3.4 implies that

$$\|\mathbf{u} - \mathbf{u}'\| \leq \frac{1}{\sqrt{2(\rho_{\mathbf{x},1}^2 + \rho_{\mathbf{y},1}^2 + 2\rho_{\mathbf{xy},1}^2)}} \leq \frac{1}{\rho_1}$$

Moreover we have

$$\|\nabla_{\mathbf{u}}^2 f(\mathbf{u}) - \nabla_{\mathbf{u}}^2 f(\mathbf{u}')\|$$

$$= \sqrt{\|\nabla_{\mathbf{xx}}^2 f(\mathbf{u}) - \nabla_{\mathbf{xx}}^2 f(\mathbf{u}')\|^2 + \|\nabla_{\mathbf{yy}}^2 f(\mathbf{u}) - \nabla_{\mathbf{yy}}^2 f(\mathbf{u}')\|^2 + 2\|\nabla_{\mathbf{xy}}^2 f(\mathbf{u}) - \nabla_{\mathbf{xy}}^2 f(\mathbf{u}')\|^2}$$

$$\leq \sqrt{\frac{1}{4}(\rho_0 + \rho_1 \|\nabla_{\mathbf{x}} f(\mathbf{u})\|)^2 \|\mathbf{u} - \mathbf{u}'\|^2 + \frac{1}{4}(\rho_0 + \rho_1 \|\nabla_{\mathbf{y}} f(\mathbf{u})\|)^2 \|\mathbf{u} - \mathbf{u}'\|^2}$$

$$\leq \sqrt{(\rho_0^2 + \rho_1^2 \|\nabla_{\mathbf{u}} f(\mathbf{u})\|^2) \|\mathbf{u} - \mathbf{u}'\|^2}$$

$$\leq (\rho_0 + \rho_1 \|\nabla_{\mathbf{u}} f(\mathbf{u})\|) \|\mathbf{u} - \mathbf{u}'\|,$$

where the first inequality holds by using

$$\|\nabla_{\mathbf{xy}}^2 f(\mathbf{u}) - \nabla_{\mathbf{xy}}^2 f(\mathbf{u}')\|^2 \leq \frac{1}{8}(\rho_0 + \rho_1 \min\{\|\nabla_{\mathbf{x}} f(\mathbf{u})\|, \|\nabla_{\mathbf{y}} f(\mathbf{u})\|\})^2 \|\mathbf{u} - \mathbf{u}'\|^2$$

Then we finish the proof.

**Lemma A.5** *Under Assumption 4.1, running Algorithm 1 with parameters setting in Theorem 4.3. For iterations in the epochs that the if condition on Line 9 of Algorithm 1 triggers, we have*

$$\|\mathbf{x}_t - \mathbf{x}_{t-k+1}\| \leq B, \quad \|\mathbf{z}_t - \mathbf{x}_{t-k+1}\| \leq 2B. \tag{13}$$

*Otherwise we have*

$$\|\mathbf{x}_{t+1} - \mathbf{x}_{t-k+1}\| \leq B, \quad \|\mathbf{z}_{t+1} - \mathbf{x}_{t-k+1}\| \leq 2B. \tag{14}$$

*Proof.* Denote $t_{\mathcal{K}}$ to be the iteration number when Line 9 triggers and $\mathcal{K}$ to be the value of $k$ in that iteration with $\mathcal{K} \leq K$. Then we have

$$\mathcal{K} = \min_k \left\{ k \mid k \sum_{i=t-k+1}^{t} \|\mathbf{x}_{i+1} - \mathbf{x}_i\|^2 > B^2 \right\}. \tag{15}$$

Then for any iteration with $t_{\mathcal{K}} - \mathcal{K} + 1 \leq t' \leq t_{\mathcal{K}}$ and $0 \leq k' < \mathcal{K}$, we have

$$\|\mathbf{x}_{t'} - \mathbf{x}_{t'-k'+1}\|^2 = \left\| \sum_{i=t'-k'+1}^{t'-1} \mathbf{x}_{i+1} - \mathbf{x}_i \right\|^2 \leq k' \sum_{i=t'-k'+1}^{t'-1} \|\mathbf{x}_{i+1} - \mathbf{x}_i\|^2 \leq B^2 \tag{16}$$

Also, from the update of $\mathbf{z}$ we have

$$\|\mathbf{z}_{t'} - \mathbf{x}_{t'-k'+1}\| \leq \|\mathbf{x}_{t'} - \mathbf{x}_{t'-k+1}\| + \|\mathbf{x}_{t'} - \mathbf{x}_{t'-1}\| \leq 2B \tag{17}$$

On the other hand, in the epochs that the condition $k = K$ on Line 11 triggers, for any iteration with $t_K - K + 1 \leq t' \leq t_K$ and $0 \leq k' \leq K$, we have

$$\|\mathbf{x}_{t'+1} - \mathbf{x}_{t'-k'+1}\|^2 \leq k' \sum_{i=t'-k'+1}^{t'} \|\mathbf{x}_{i+1} - \mathbf{x}_i\|^2 \leq B^2$$

$$\|\mathbf{z}_{t'+1} - \mathbf{x}_{t'-k'+1}\| \leq 2B \tag{18}$$

For all the other iterations, if the condition on Line 20 triggers, where we have

$$\|\mathbf{z}_{t'+1} - \mathbf{x}_{t'-k'+1}\| = \|\mathbf{x}_{t'+1} - \mathbf{x}_{t'-k'+1}\| \leq \frac{1}{4}B \tag{19}$$

Otherwise from the setting of $r$ in Theorem 4.3 we have

$$\|\mathbf{z}_{t'+1} - \mathbf{x}_{t'-k'+1}\| = \|\mathbf{x}_{t'+1} - \mathbf{x}_{t'-k'+1}\| \leq r \leq B \tag{20}$$

which complete the proof.

**Lemma A.6** *Under Assumption 4.1, running Algorithm 1 with parameters setting in Theorem 4.3. Denote $\Delta_\Phi := \Phi(\mathbf{x}_0) - \Phi^*$, there must exist a constant $G$ such that*

$$G = \max\left\{2\|\nabla\Phi(\mathbf{x}_0)\|, \max\left\{u \geq 0 \mid u^2 \leq 2\mathcal{L} \cdot (\Phi(\mathbf{x}_0) - \Phi^*)\right\}\right\}$$

*Proof.* Consider the first epoch before Line 9, 11, or 20 trigger. By Lemma A.2 and the choice of $G$, it is easy to verify that $\|\nabla\Phi(\mathbf{x}_0)\| \leq G$. By Lemma A.5 we have $\|\mathbf{x}_t - \mathbf{x}_0\| \leq B \leq r(G)$ and $\|\mathbf{z}_t - \mathbf{x}_0\| \leq 2B \leq r(G)$ for any $t$. Therefore, by Lemma A.1 we have

$$\|\nabla\Phi(\mathbf{x}_t)\| \leq \|\nabla\Phi(\mathbf{x}_0)\| + \mathcal{L}\|\mathbf{x}_t - \mathbf{x}_0\| \leq \frac{1}{2}G + \mathcal{L} \cdot B \leq G \tag{21}$$

Similarally, we have $\|\nabla\Phi(\mathbf{z}_t)\| \leq G$. Without loss of generality, we first consider that only the if condition on Line 9 triggers in all epochs. From C.1 we directly obtain that $\Phi(\mathbf{z}_{\mathcal{K}-1}) \leq \Phi(\mathbf{x}_0)$. Then by Lemma A.2, we have $\|\nabla\Phi(\mathbf{z}_{\mathcal{K}-1})\| \leq \|\nabla\Phi(\mathbf{x}_0)\| \leq \frac{1}{2}G$. By the restart operation we have $\mathbf{x}_{\mathcal{K}} = \mathbf{z}_{\mathcal{K}} = \mathbf{z}_{\mathcal{K}-1}$. Telescoping to all epochs, we have $\|\nabla\Phi(\mathbf{x}_t)\| \leq G$ and $\|\nabla\Phi(\mathbf{z}_t)\| \leq G$.

For the epoch that Line 11 triggers, according to A.5, from the updates of $\hat{\mathbf{z}}$ we have $\|\hat{\mathbf{z}} - \mathbf{x}_{t-K+1}\| \leq 2B \leq G$. Then by the settings of $G$ and $r$ we have

$$\|\mathbf{x}_{t+1} - \mathbf{x}_{t-K+1}\| = \|\mathbf{z}_{t+1} - \mathbf{x}_{t-K+1}\| \leq \|\hat{\mathbf{z}} - \mathbf{x}_{t-K+1}\| + r \leq \frac{G}{\mathcal{L}}, \tag{22}$$

which yields that $\|\nabla\Phi(\mathbf{z}_{t+1})\| = \|\nabla\Phi(\mathbf{x}_{t+1})\| \leq G$.

For the other epochs, before the condition on Line 20 triggers (i.e. $k < \mathscr{T}$), we have $\|\mathbf{x}_{t+1} - \hat{\mathbf{z}}\| = \|\mathbf{z}_{t+1} - \hat{\mathbf{z}}\| = r$. When Line 20 triggers, we have $\|\mathbf{x}_{t+1} - \hat{\mathbf{z}}\| = \|\mathbf{z}_{t+1} - \hat{\mathbf{z}}\| = \frac{1}{4}B \leq r(G)$. Therefore for iterations in these epochs we have $\|\nabla\Phi(\mathbf{z}_{t+1})\| = \|\nabla\Phi(\mathbf{x}_{t+1})\| \leq G$, which finish the proof.

**Lemma A.7** *(Li et al., 2024a) Consider the first AGD routine. Denote $\Delta_f = f(\mathbf{x}_0, \mathbf{y}_0^0) - f(\mathbf{x}_0, \mathbf{y}^*(\mathbf{x}_0))$, there must exist a constant $G_\mathbf{y}$ such that for $\mathcal{L}_\mathbf{y} = l_{\mathbf{y},0} + 2l_{\mathbf{y},1}G_\mathbf{y}$ we have*

$$G_\mathbf{y} \geq \max\left\{2\|\nabla f(\mathbf{x}_0, \mathbf{y}_0^0)\|, 8\max\left\{\sqrt{\mathcal{L}_\mathbf{y}}, 1\right\}\sqrt{\mathcal{L}_\mathbf{y}\left(\Delta_f + \mu\|\mathbf{y}_0 - \mathbf{y}^*(\mathbf{x}_0)\|^2\right)/\min\{\mu, 1\}}\right\} \tag{23}$$

*Also, for any $d \leq D$, we have $\|\nabla_\mathbf{y} f(\mathbf{x}_0, \mathbf{y}_0^d)\| \leq G_\mathbf{y}$.*

**Lemma A.8** *For any $t \leq T$ in Algorithm 1 and $d \leq D$ in all the AGD routine of Algorithm 2 we have $\|\nabla_\mathbf{y} f(\mathbf{x}_t, \mathbf{y}_t^d)\| \leq G_\mathbf{y}$.*

*Proof.* By Lemma A.5 and the setting of $G_\mathbf{y}$ we have for any $t \leq T$, $\|\mathbf{x}_t - \mathbf{x}_{t-1}\| \leq B \leq \frac{G_\mathbf{y}}{\mathcal{L}_\mathbf{y}}$. Also by warm start strategy on $\mathbf{y}$ we have $\mathbf{y}_{t+1}^0 = \mathbf{y}_t^d$. Together with Lemma A.1 we complete the proof.

**Lemma A.9** *(Chen et al., 2021) Under Assumption 4.1, $\|\nabla_{\mathbf{yy}} f(\mathbf{x}, \mathbf{y})\|^{-1}$ is bounded. i.e. $\|\nabla_{\mathbf{yy}} f(\mathbf{x}, \mathbf{y})\|^{-1} \leq \mu^{-1}$*

## A.1 PROOF OF LEMMA 4.2

Under Assumption 4.1, indeed, a function $\mathbf{y}^*(\cdot)$ is well-defined since $f(\mathbf{x}, \cdot)$ is strongly concave for each $\mathbf{x} \in \mathbb{R}^m$. Then, let $\mathbf{x}_1, \mathbf{x}_2 \in \mathbb{R}^m$, the definition of $\mathbf{y}^*(\mathbf{x}_1)$ and the definition of $\mathbf{y}^*(\mathbf{x}_2)$ imply that

$$(\mathbf{y} - \mathbf{y}^*(\mathbf{x}_1))^\top \nabla_\mathbf{y} f(\mathbf{x}_1, \mathbf{y}^*(\mathbf{x}_1)) \leq 0, \text{ for all } \mathbf{y} \in \mathcal{Y} \tag{24}$$

$$(\mathbf{y} - \mathbf{y}^*(\mathbf{x}_2))^\top \nabla_\mathbf{y} f(\mathbf{x}_2, \mathbf{y}^*(\mathbf{x}_2)) \leq 0, \text{ for all } \mathbf{y} \in \mathcal{Y} \tag{25}$$

Letting $\mathbf{y} = \mathbf{y}^*(\mathbf{x}_2)$ in Eq. (A.6) and $\mathbf{y} = \mathbf{y}^*(\mathbf{x}_1)$ in Eq. (A.7) and adding them yields

$$(\mathbf{y}^*(\mathbf{x}_2) - \mathbf{y}^*(\mathbf{x}_1))^\top (\nabla_\mathbf{y} f(\mathbf{x}_1, \mathbf{y}^*(\mathbf{x}_1)) - \nabla_\mathbf{y} f(\mathbf{x}_2, \mathbf{y}^*(\mathbf{x}_2))) \leq 0 \tag{26}$$

Recall that $f(\mathbf{x}_1, \cdot)$ is $\mu$-strongly concave, we have

$$(\mathbf{y}^*(\mathbf{x}_2) - \mathbf{y}^*(\mathbf{x}_1))^\top (\nabla_\mathbf{y} f(\mathbf{x}_1, \mathbf{y}^*(\mathbf{x}_2)) - \nabla_\mathbf{y} f(\mathbf{x}_1, \mathbf{y}^*(\mathbf{x}_1))) + \mu\|\mathbf{y}^*(\mathbf{x}_2) - \mathbf{y}^*(\mathbf{x}_1)\|^2 \leq 0 \tag{27}$$

By combining Eq. (A.8) and Eq. (A.9) with the $(l_{\mathbf{x},0}, l_{\mathbf{x},1}, l_{\mathbf{y},0}, l_{\mathbf{y},1})$-smoothness of $f$, we have

$$\mu \|\mathbf{y}^*(\mathbf{x}_2) - \mathbf{y}^*(\mathbf{x}_1)\|^2 \leq (l_{\mathbf{y},0} + l_{\mathbf{y},1}\|\nabla_{\mathbf{y}}f(\mathbf{x}_2, \mathbf{y}^*(\mathbf{x}_2))\|)\|\mathbf{y}^*(\mathbf{x}_2) - \mathbf{y}^*(\mathbf{x}_1)\|\|\mathbf{x}_2 - \mathbf{x}_1\| \quad (28)$$

Combine with the definition of $\mathbf{y}^*(\cdot)$, we obtain that $\mathbf{y}^*(\cdot)$ is $\frac{l_{\mathbf{y},0}}{\mu}$-Lipschitz.
Then, we prove the smoothness of $\Phi(\cdot)$. Let $\mathbf{u} = (\mathbf{x}, \mathbf{y}^*(\mathbf{x}))$ and $\mathbf{u}' = (\mathbf{x}', \mathbf{y}^*(\mathbf{x}'))$, by (6) and the $\frac{l_{\mathbf{y},0}}{\mu}$-Lipschitz of $\mathbf{y}'(\mathbf{x})$ we have

$$\|\mathbf{u} - \mathbf{u}'\| = \sqrt{\|\mathbf{x} - \mathbf{x}'\|^2 + \|\mathbf{y}^*(\mathbf{x}) - \mathbf{y}^*(\mathbf{x}')\|^2} \leq \left(1 + \frac{l_{\mathbf{y},\mathbf{0}}}{\mu}\right)\|\mathbf{x} - \mathbf{x}'\| \leq \frac{G}{\mathcal{L}} \quad (29)$$

Then we have

$$\begin{aligned}
&\|\nabla\Phi(\mathbf{x}) - \nabla\Phi(\mathbf{x}')\| \\
&\leq \|\nabla_{\mathbf{x}}f(\mathbf{x}, \mathbf{y}(\mathbf{x})) - \nabla_{\mathbf{x}}f(\mathbf{x}', \mathbf{y}'(\mathbf{x}))\| \\
&\leq (l_{\mathbf{x},0} + l_{\mathbf{x},1}\|\nabla_{\mathbf{x}}f(\mathbf{x}', \mathbf{y}^*(\mathbf{x}'))\|)(\|\mathbf{x} - \mathbf{x}'\| + \|\mathbf{y}^*(\mathbf{x}) - \mathbf{y}^*(\mathbf{x}')\|) \\
&\leq (l_{\mathbf{x},0} + l_{\mathbf{x},1}\|\nabla\Phi(\mathbf{x}')\|)\left(1 + \frac{l_{\mathbf{y},\mathbf{0}}}{\mu}\right)\|\mathbf{x} - \mathbf{x}'\|
\end{aligned} \quad (30)$$

Therefore, the function $\Phi(\mathbf{x})$ is $(l_{\Phi,0}, l_{\Phi,1})$-smooth, where we denote

$$l_{\Phi,0} = \left(1 + \frac{l_{\mathbf{y},\mathbf{0}}}{\mu}\right)l_{\mathbf{x},0}, \quad l_{\Phi,1} = \left(1 + \frac{l_{\mathbf{y},\mathbf{0}}}{\mu}\right)l_{\mathbf{x},1} \quad (31)$$

For minimax optimization (1), we know that $\nabla^2_{\mathbf{x}\mathbf{y}}f(\mathbf{x}, \mathbf{y}) = \nabla^2_{\mathbf{y}\mathbf{x}}f(\mathbf{x}, \mathbf{y})$. According to (2), with the setting of $G_{\mathbf{y}}$ and Lemma A.7, A.8 we can easily verify that

$$\|\nabla^2_{\mathbf{x}\mathbf{y}}f(\mathbf{x}, \mathbf{y})\| \leq l_{\mathbf{x},\mathbf{0}} + l_{\mathbf{x},\mathbf{1}}\|\nabla_{\mathbf{y}}f(\mathbf{x}, \mathbf{y})\| \leq l_{\mathbf{x},\mathbf{0}} + l_{\mathbf{x},\mathbf{1}}G_{\mathbf{y}} = M,$$

Next, we prove the Hessian Lipschitz continuity of $\Phi(\mathbf{x})$. Define mapping $\mathcal{H}(\mathbf{x}, \mathbf{y}) = [\nabla_{\mathbf{x}\mathbf{x}}f - \nabla_{\mathbf{x}\mathbf{y}}f(\nabla_{\mathbf{y}\mathbf{y}}f)^{-1}\nabla_{\mathbf{y}\mathbf{x}}f](\mathbf{x}, \mathbf{y})$. Also, denote that $\mathbf{u} = (\mathbf{x}, \mathbf{y})$ and $\mathbf{u}' = (\mathbf{x}', \mathbf{y}')$, by the assumptions we have

$$\begin{aligned}
&\|\mathcal{H}(\mathbf{x}', \mathbf{y}') - \mathcal{H}(\mathbf{x}, \mathbf{y})\| \\
&\leq \|\nabla_{\mathbf{x}\mathbf{x}}f(\mathbf{x}', \mathbf{y}') - \nabla_{\mathbf{x}\mathbf{x}}f(\mathbf{x}, \mathbf{y})\| + \|\nabla_{\mathbf{x}\mathbf{y}}f(\mathbf{x}, \mathbf{y})\|\|\left(\nabla_{\mathbf{y}\mathbf{y}}f(\mathbf{x}', \mathbf{y}')\right)^{-1} - \left(\nabla_{\mathbf{y}\mathbf{y}}f(\mathbf{x}, \mathbf{y})\right)^{-1}\|\|\nabla_{\mathbf{y}\mathbf{x}}f(\mathbf{x}', \mathbf{y}')\| \\
&\quad + \|\nabla_{\mathbf{x}\mathbf{y}}f(\mathbf{x}', \mathbf{y}') - \nabla_{\mathbf{x}\mathbf{y}}f(\mathbf{x}, \mathbf{y})\|\|\left(\nabla_{\mathbf{y}\mathbf{y}}f(\mathbf{x}', \mathbf{y}')\right)^{-1}\|\|\nabla_{\mathbf{y}\mathbf{x}}f(\mathbf{x}', \mathbf{y}')\| \\
&\quad + \|\nabla_{\mathbf{x}\mathbf{y}}f(\mathbf{x}, \mathbf{y})\|\|\left(\nabla_{\mathbf{y}\mathbf{y}}f(\mathbf{x}, \mathbf{y})^{-1}\right)\|\|\nabla_{\mathbf{y}\mathbf{x}}f(\mathbf{x}', \mathbf{y}') - \nabla_{\mathbf{y}\mathbf{x}}f(\mathbf{x}, \mathbf{y})\| \\
&\leq (\rho_{\mathbf{x},0} + \rho_{\mathbf{x},1}\|\nabla_{\mathbf{x}}f(\mathbf{u})\|)\|\mathbf{u}' - \mathbf{u}\| + (\rho_{\mathbf{x}\mathbf{y},0} + \rho_{\mathbf{x}\mathbf{y},1}\|\nabla_{\mathbf{y}}f(\mathbf{u})\|)\|\mathbf{u}' - \mathbf{u}\|\mu^{-1}M \\
&\quad + M\mu^{-1}(\rho_{\mathbf{x}\mathbf{y},0} + \rho_{\mathbf{x}\mathbf{y},1}\|\nabla_{\mathbf{y}}f(\mathbf{u})\|)\|\mathbf{u}' - \mathbf{u}\| \\
&\quad + M^2\|(\nabla_{\mathbf{y}\mathbf{y}}f(\mathbf{x}', \mathbf{y}'))^{-1}\|\|\nabla_{\mathbf{y}\mathbf{y}}f(\mathbf{x}, \mathbf{y}) - \nabla_{\mathbf{y}\mathbf{y}}f(\mathbf{x}', \mathbf{y}')\|\|(\nabla_{\mathbf{y}\mathbf{y}}f(\mathbf{x}, \mathbf{y}))^{-1}\| \\
&\leq (\rho_{\mathbf{x},0} + \rho_{\mathbf{x},1}\|\nabla_{\mathbf{x}}f(\mathbf{u})\|)\|\mathbf{u}' - \mathbf{u}\| + 2(\rho_{\mathbf{x}\mathbf{y},0} + \rho_{\mathbf{x}\mathbf{y},1}\|\nabla_{\mathbf{y}}f(\mathbf{u})\|)\|\mathbf{u}' - \mathbf{u}\|\mu^{-1}M \\
&\quad + (\rho_{\mathbf{y},0} + \rho_{\mathbf{y},1}\|\nabla_{\mathbf{y}}f(\mathbf{u})\|)\|\mathbf{u}' - \mathbf{u}\|\mu^{-2}M^2 \\
&\leq (\rho_{\mathbf{x},0} + \rho_{\mathbf{x},1}\|\nabla_{\mathbf{x}}f(\mathbf{u})\|)\|\mathbf{u}' - \mathbf{u}\| + (\mu^{-1}M\sqrt{\rho_{\mathbf{y},\mathbf{0}}} + \frac{\rho_{\mathbf{x}\mathbf{y},\mathbf{0}}}{\sqrt{\rho_{\mathbf{y},\mathbf{0}}}})^2\|\mathbf{u}' - \mathbf{u}\| \\
&\quad + (\mu^{-1}M\sqrt{\rho_{\mathbf{y},\mathbf{1}}} + \frac{\rho_{\mathbf{x}\mathbf{y},\mathbf{1}}}{\sqrt{\rho_{\mathbf{y},\mathbf{1}}}})^2\|\nabla_{\mathbf{y}}f(\mathbf{u})\|\|\mathbf{u}' - \mathbf{u}\|
\end{aligned}$$

$$(32)$$

From the definition of $\mathbf{y}^*(\mathbf{x})$, we know that $\nabla_{\mathbf{y}}f(\mathbf{x}, \mathbf{y}^*(\mathbf{x})) = 0$ for all $\mathbf{x} \in \mathbb{R}^{d_x}$, Thus we can obtain that

$$\mathbf{0} = \nabla_{\mathbf{x}}\nabla_{\mathbf{y}}f(\mathbf{x}, \mathbf{y}^*(\mathbf{x})) = \nabla_{\mathbf{y}\mathbf{x}}f(\mathbf{x}, \mathbf{y}^*(\mathbf{x})) + \nabla_{\mathbf{y}\mathbf{y}}f(\mathbf{x}, \mathbf{y}^*(\mathbf{x}))\nabla\mathbf{y}^*(\mathbf{x}) \quad (33)$$

which implies that

$$\nabla\mathbf{y}^*(\mathbf{x}) = -\left[\nabla_{\mathbf{y}\mathbf{y}}f(\mathbf{x}, \mathbf{y}^*(\mathbf{x}))\right]^{-1}\nabla_{\mathbf{y}\mathbf{x}}f(\mathbf{x}, \mathbf{y}^*(\mathbf{x})) \quad (34)$$

Substitute all above, with $\nabla\Phi(\mathbf{x}) = \nabla_{\mathbf{x}}f(\mathbf{x}, \mathbf{y}^*(\mathbf{x}))$, we have

$$
\begin{aligned}
\nabla^2\Phi(\mathbf{x}) &= \nabla_{\mathbf{xx}}f(\mathbf{x}, \mathbf{y}^*(\mathbf{x})) + \nabla_{\mathbf{xy}}f(\mathbf{x}, \mathbf{y}^*(\mathbf{x}))\nabla\mathbf{y}^*(\mathbf{x}) \\
&= \nabla_{\mathbf{xx}}f(\mathbf{x}, \mathbf{y}^*(\mathbf{x})) - \nabla_{\mathbf{xy}}f(\mathbf{x}, \mathbf{y}^*(\mathbf{x}))[\nabla_{\mathbf{yy}}f(\mathbf{x}, \mathbf{y}^*(\mathbf{x}))]^{-1}\nabla_{\mathbf{yx}}f(\mathbf{x}, \mathbf{y}^*(\mathbf{x})) \qquad (35) \\
&= \mathcal{H}(\mathbf{x}, \mathbf{y}^*(\mathbf{x}))
\end{aligned}
$$

Then,

$$
\begin{aligned}
\|\nabla^2\Phi(\mathbf{x}') - \nabla^2\Phi(\mathbf{x})\| &= \|\mathcal{H}(\mathbf{x}', \mathbf{y}^*(\mathbf{x}')) - \mathcal{H}(\mathbf{x}, \mathbf{y}^*(\mathbf{x}))\| \\
&\le \left( \rho_{\mathbf{x},0} + \rho_{\mathbf{x},1}\|\nabla_{\mathbf{x}}f(\mathbf{x}, \mathbf{y}^*(\mathbf{x}))\| + \left( (\mu^{-1}M\sqrt{\rho_{\mathbf{y},0}} + \frac{\rho_{\mathbf{xy},0}}{\sqrt{\rho_{\mathbf{y},0}}})^2 + (\mu^{-1}M\sqrt{\rho_{\mathbf{y},1}} + \frac{\rho_{\mathbf{xy},1}}{\sqrt{\rho_{\mathbf{y},1}}})^2\|\nabla_{\mathbf{y}}f(\mathbf{x}, \mathbf{y}^*(\mathbf{x}))\| \right) \right)\|\mathbf{u}' - \mathbf{u}\| \\
&\le \left( (\rho_{\mathbf{x},0} + \rho_{\mathbf{x},1}\|\nabla_{\mathbf{x}}f(\mathbf{x}, \mathbf{y}^*(\mathbf{x}))\|) + (\mu^{-1}M\sqrt{\rho_{\mathbf{y},0}} + \frac{\rho_{\mathbf{xy},0}}{\sqrt{\rho_{\mathbf{y},0}}})^2 \right)\|\mathbf{u}' - \mathbf{u}\| \\
&\le \left( \rho_{\mathbf{x},0} + (\mu^{-1}M\sqrt{\rho_{\mathbf{y},0}} + \frac{\rho_{\mathbf{xy},0}}{\sqrt{\rho_{\mathbf{y},0}}})^2 + \rho_{\mathbf{x},1}\|\nabla_{\mathbf{x}}f(\mathbf{x}, \mathbf{y}^*(\mathbf{x}))\| \right)(\|\mathbf{x}' - \mathbf{x}\| + \|\mathbf{y}^*(\mathbf{x}') - \mathbf{y}^*(\mathbf{x})\|) \\
&\le \left( \rho_{\mathbf{x},0} + (\mu^{-1}M\sqrt{\rho_{\mathbf{y},0}} + \frac{\rho_{\mathbf{xy},0}}{\sqrt{\rho_{\mathbf{y},0}}})^2 + \rho_{\mathbf{x},1}\|\nabla\Phi(\mathbf{x})\| \right)\left( 1 + \frac{l_{\mathbf{y},0}}{\mu} \right)\|\mathbf{x}' - \mathbf{x}\|
\end{aligned}
$$

$$(36)$$

Therefore, the function $\Phi(\mathbf{x})$ is $(\rho_{\phi,0}, \rho_{\phi,1})$-Hessian Lipschitz continuous, where

$$
\begin{aligned}
\rho_{\phi,0} &= \left( 1 + \frac{l_{\mathbf{y},0}}{\mu} \right)(\rho_{\mathbf{x},0} + (\mu^{-1}M\sqrt{\rho_{\mathbf{y},0}} + \frac{\rho_{\mathbf{xy},0}}{\sqrt{\rho_{\mathbf{y},0}}})^2), \\
\rho_{\phi,1} &= \left( 1 + \frac{l_{\mathbf{y},0}}{\mu} \right)\rho_{\mathbf{x},1}
\end{aligned}
$$

$$(37)$$

**Lemma A.10** *Running Algorithm 1 with the parameters on 4.3. Denote $\mathcal{L}_{\mathbf{y}} = l_{\mathbf{y},0} + 2l_{\mathbf{y},0}G_{\mathbf{y}}$ as the efficient smoothness constant of $f(\mathbf{x}, \cdot)$ and $\kappa = \frac{\mathcal{L}_{\mathbf{y}}}{\mu}$. For the AGD procedure of Algorithm 2, set $\eta_{\mathbf{y}}, \theta_{\mathbf{y}}$ to be*

$$
\eta_{\mathbf{y}} = \frac{1}{\mathcal{L}_{\mathbf{y}}}, \quad \theta_{\mathbf{y}} = \frac{\sqrt{\kappa} - 1}{\sqrt{\kappa} + 1} \tag{38}
$$

*the output $\mathbf{y}_t$ satisfying $\|\mathbf{y}_D - \mathbf{y}^*\|_2^2 \le (\kappa+1)\left( 1 - \frac{1}{\sqrt{\kappa}} \right)^D\|\mathbf{y}_0 - \mathbf{y}^*\|_2^2$, where $\mathbf{y}^* = \arg\min_{\mathbf{y}} h(\mathbf{y})$.*

*Proof.* For Algorithm 2 with function $h'(\cdot)$ that is $l_h$-Lipschitz smooth and $\mu_h$ strongly-convex, from the analysis of Wang & Li (2020) it yields that $\|\mathbf{y}_D - \mathbf{y}^*\|_2^2 \le (\kappa_h + 1)(1 - \frac{1}{\sqrt{\kappa_h}})^D\|\mathbf{y}_0 - \mathbf{y}^*\|_2^2$, where $\kappa_h = \frac{l_h}{\mu_h}$.

For a $(l_{\mathbf{y},0}, l_{\mathbf{y},1})$-smooth and $\mu$-strongly-convex function $-f(\mathbf{x}, \cdot)$, it is easy to verify that by the setting of $G_{\mathbf{y}}$ and Lemma A.7 the condition still holds, which complete the proof.

# B  PROOF OF SECTION 4.4.1

**Lemma B.1** *Denote $\widehat{\nabla}\Phi(\mathbf{x_t}) = \nabla_{\mathbf{x}}f(\mathbf{x_t}, \mathbf{y_t})$. Let $\iota$ be a constant with $\iota = c \cdot \log(\frac{1}{\delta_0}\sqrt{\frac{n}{\pi\rho}}) > 1$ and $\kappa = \frac{\mathcal{L}_{\mathbf{y}}}{\mu}$. Running Algorithm 1 with the parameters setting in Theorem 4.3, Denote $\delta_{\mathbf{y}_0} = \|\mathbf{y}^0 - \mathbf{y}^*(\mathbf{x}_0)\|$, then the estimation error $\delta_{\widehat{\Phi}} = \|\nabla\Phi(\mathbf{x}_t) - \widehat{\nabla}\Phi(\mathbf{x}_t)\|$ can be bounded as*

$$
\|\nabla\Phi(\mathbf{x}_t) - \widehat{\nabla}\Phi(\mathbf{x}_t)\| \le \min\left\{ \frac{1}{4}, \frac{1}{\iota^2 2^{6-\iota}} \right\} \cdot \epsilon \tag{39}
$$

*Proof.* Denote $\kappa = \frac{\mathcal{L}_{\mathbf{y}}}{\mu}$, The gradient estimation error can be bounded by

$$
\begin{aligned}
\left\| \widehat{\nabla}\Phi(\mathbf{x}_t) - \nabla\Phi(\mathbf{x}_t) \right\| &= \left\| \nabla_{\mathbf{x}}f(\mathbf{x}_t, \mathbf{y}_t^D) - \nabla_{\mathbf{x}}f(\mathbf{x}_t, \mathbf{y}^*(\mathbf{x}_t)) \right\| \le \mathcal{L}\left\| \mathbf{y}_t^D - \mathbf{y}^*(\mathbf{x}_t) \right\| \\
&\le \mathcal{L}(\kappa + 1)\left( 1 - \frac{1}{\sqrt{\kappa}} \right)^{D/2}\left\| \mathbf{y}_t^0 - \mathbf{y}^*(\mathbf{x}_t) \right\|
\end{aligned}
$$

$$(40)$$

where the last inequality follows Lemma A.10. By the warm start strategy $\mathbf{y}_t^0 = \mathbf{y}_{t-1}^D$, we have

$$\left\| \mathbf{y}_t^0 - \mathbf{y}^*(\mathbf{x}_t) \right\| \leq \left\| \mathbf{y}_{t-1}^D - \mathbf{y}^*(\mathbf{x}_{t-1}) \right\| + \left\| \mathbf{y}^*(\mathbf{x}_{t-1}) - \mathbf{y}^*(\mathbf{x}_t) \right\|$$

$$\leq \left( 1 - \frac{1}{\sqrt{\kappa}} \right)^{\frac{D}{2}} \left\| \mathbf{y}_{t-1}^0 - \mathbf{y}^*(\mathbf{x}_{t-1}) \right\| + \frac{l_{\mathbf{y},0}}{\mu} \left\| \mathbf{x}_t - \mathbf{x}_{t-1} \right\| \tag{41}$$

$$\leq \left( 1 - \frac{1}{\sqrt{\kappa}} \right)^{\frac{D}{2}} \left\| \mathbf{y}_{t-1}^0 - \mathbf{y}^*(\mathbf{x}_{t-1}) \right\| + \frac{l_{\mathbf{y},0}}{\mu} B.$$

By setting

$$D > 2 \log 2 / \log \left( \frac{1}{1 - \kappa^{-1/2}} \right) = \mathcal{O}(\kappa) \tag{42}$$

we have

$$\left\| \mathbf{y}_t^0 - \mathbf{y}^*(\mathbf{x}_t) \right\| \leq \frac{1}{2} \left\| \mathbf{y}_{t-1}^0 - \mathbf{y}^*(\mathbf{x}_{t-1}) \right\| + \frac{l_{\mathbf{y},0} B}{\mu}$$

$$\leq \left( \frac{1}{2} \right)^t \left\| \mathbf{y}^0 - \mathbf{y}^*(\mathbf{x}_0) \right\| + \sum_{j=0}^{t-1} \left( \frac{1}{2} \right)^{t-1-j} \frac{l_{\mathbf{y},0}}{\mu} B \tag{43}$$

$$\leq \delta_{\mathbf{y}_0} + 2 \frac{l_{\mathbf{y},0} B}{\mu},$$

which yields that

$$\left\| \widehat{\nabla} \Phi(\mathbf{x}_t) - \nabla \Phi(\mathbf{x}_t) \right\| \leq \mathcal{L} (\kappa + 1) (\delta_{\mathbf{y}_0} + 2\kappa B) \left( 1 - \frac{1}{\sqrt{\kappa}} \right)^{D/2} \tag{44}$$

Then, it is easy to verify that let

$$D = 2 \log \left( \frac{\mathcal{L} (\kappa + 1) \left( \delta_{\mathbf{y}_0} + 2 \frac{l_{\mathbf{y},0} B}{\mu} \right)}{\min \left\{ \frac{1}{4}, \frac{1}{\iota^2 2^{6-\iota}} \right\} \cdot \epsilon} \right) / \log \left( \frac{1}{1 - \kappa^{-1/2}} \right) = \mathcal{O} \left( \sqrt{\kappa} \log \left( \frac{1}{\epsilon} \right) \right) \tag{45}$$

finish the proofs.

## C   PROOF OF SECTION 4.4.2

**Lemma C.1** *Running Algorithm 1 with parameters setting in Theorem 4.3. When the condition on Line 9 triggers, denote $t_{\mathcal{K}}$ to be the iteration number, $\mathcal{K}$ to be the value of $k$ on that iteration and $t_0 = t_{\mathcal{K}} - \mathcal{K} + 1$. If $\| \widehat{\nabla} \Phi(\mathbf{z}_{t_{\mathcal{K}}}) \| > \frac{B}{\eta_{\mathbf{x}}}$, we have*

$$\Phi(\mathbf{x}_{t_{\mathcal{K}}+1}) - \Phi(\mathbf{x}_{t_0}) \leq -\frac{5B^2}{128\eta_{\mathbf{x}}} \tag{46}$$

Proof. Denote $\delta_{\widehat{\Phi}} = \nabla \Phi(\mathbf{z}_t) - \widehat{\nabla} \Phi(\mathbf{z}_t)$. From the $\mathcal{L}$-smoothness condition and Lemma A.1, we have for $t_0 \leq t \leq t_{\mathcal{K}}$

$$\Phi(\mathbf{x}_{t+1}) \leq \Phi(\mathbf{z}_t) + \langle \nabla \Phi(\mathbf{z}_t), \mathbf{x}_{t+1} - \mathbf{z}_t \rangle + \frac{\mathcal{L}}{2} \| \mathbf{x}_{t+1} - \mathbf{z}_t \|^2$$

$$\leq \Phi(\mathbf{z}_t) + \frac{\mathcal{L}}{2} \| \mathbf{x}_{t+1} - \mathbf{z}_t \|_2^2 + \langle \nabla \Phi(\mathbf{z}_t) - \widehat{\nabla} \Phi(\mathbf{z}_t), \mathbf{x}_{t+1} - \mathbf{z}_t \rangle + \langle \widehat{\nabla} \Phi(\mathbf{z}_t), \mathbf{x}_{t+1} - \mathbf{z}_t \rangle$$

$$\leq \Phi(\mathbf{z}_t) + \frac{\mathcal{L}}{2} \| \mathbf{x}_{t+1} - \mathbf{z}_t \|_2^2 + 8\eta_{\mathbf{x}} \| \delta_{\widehat{\Phi}} \|^2 + \frac{1}{16\eta_{\mathbf{x}}} \| \mathbf{x}_{t+1} - \mathbf{z}_t \|^2 - \eta_{\mathbf{x}} \| \widehat{\nabla} \Phi(\mathbf{z}_t) \|^2$$

$$\leq \Phi(\mathbf{z}_t) - \frac{13}{16} \eta_{\mathbf{x}} \| \widehat{\nabla} \Phi(\mathbf{z}_t) \|^2 + 8\eta_{\mathbf{x}} \| \delta_{\widehat{\Phi}} \|^2$$

$$\tag{47}$$

where we use the AGD iteration and $\eta_{\mathbf{x}} \leq \frac{1}{4\mathcal{L}}$. We also have

$$\Phi(\mathbf{x}_t) \geq \Phi(\mathbf{z}_t) + \langle \nabla \Phi(\mathbf{z}_t), \mathbf{x}_t - \mathbf{z}_t \rangle - \frac{\mathcal{L}}{2} \| \mathbf{x}_t - \mathbf{z}_t \|^2 \tag{48}$$

So we have
$$\Phi(\mathbf{x}_{t+1}) - \Phi(\mathbf{x}_t)$$

$$\leq -\langle \nabla\Phi(\mathbf{z}_t), \mathbf{x}_t - \mathbf{z}_t \rangle + \frac{\mathcal{L}}{2}\|\mathbf{x}_t - \mathbf{z}_t\|^2 - \frac{13}{16}\eta_{\mathbf{x}}\|\widehat{\nabla}\Phi(\mathbf{z}_t)\|^2 + 8\eta_{\mathbf{x}}\|\delta_{\widehat{\Phi}}\|^2$$

$$= -\langle \widehat{\nabla}\Phi(\mathbf{z}_t), \mathbf{x}_t - \mathbf{z}_t \rangle + \langle \widehat{\nabla}\Phi(\mathbf{z}_t) - \nabla\Phi(\mathbf{z}_t), \mathbf{x}_t - \mathbf{z}_t \rangle + \frac{\mathcal{L}}{2}\|\mathbf{x}_t - \mathbf{z}_t\|^2 - \frac{13}{16}\eta_{\mathbf{x}}\|\widehat{\nabla}\Phi(\mathbf{z}_t)\|^2 + 8\eta_{\mathbf{x}}\|\delta_{\widehat{\Phi}}\|^2$$

$$\leq \frac{1}{\eta_{\mathbf{x}}}\langle \mathbf{x}_{t+1} - \mathbf{z}_t, \mathbf{x}_t - \mathbf{z}_t \rangle + 4\eta_{\mathbf{x}}\|\delta_{\widehat{\Phi}}\|^2 + \frac{1}{8\eta_{\mathbf{x}}}\|\mathbf{x}_t - \mathbf{z}_t\|^2 + \frac{\mathcal{L}}{2}\|\mathbf{x}_t - \mathbf{z}_t\|^2 - \frac{13}{16}\eta_{\mathbf{x}}\|\widehat{\nabla}\Phi(\mathbf{z}_t)\|^2 + 8\eta_{\mathbf{x}}\|\delta_{\widehat{\Phi}}\|^2$$

$$= \frac{1}{2\eta_{\mathbf{x}}}(\|\mathbf{x}_{t+1} - \mathbf{z}_t\|^2 + \|\mathbf{x}_t - \mathbf{z}_t\|^2 - \|\mathbf{x}_{t+1} - \mathbf{x}_t\|^2) + \frac{1}{8\eta_{\mathbf{x}}}\|\mathbf{x}_t - \mathbf{z}_t\|^2 + \frac{\mathcal{L}}{2}\|\mathbf{x}_t - \mathbf{z}_t\|^2 - \frac{13}{16}\eta_{\mathbf{x}}\|\widehat{\nabla}\Phi(\mathbf{z}_t)\|^2 + 12\eta_{\mathbf{x}}\|\delta_{\widehat{\Phi}}\|^2$$

$$\leq \frac{3}{4\eta_{\mathbf{x}}}\|\mathbf{x}_t - \mathbf{z}_t\|^2 - \frac{1}{2\eta_{\mathbf{x}}}\|\mathbf{x}_{t+1} - \mathbf{x}_t\|^2 - \frac{5}{16}\eta_{\mathbf{x}}\|\widehat{\nabla}\Phi(\mathbf{z}_t)\|^2 + 12\eta_{\mathbf{x}}\|\delta_{\widehat{\Phi}}\|^2$$

$$\leq \frac{3}{4\eta_{\mathbf{x}}}\|\mathbf{x}_t - \mathbf{x}_{t-1}\|^2 - \frac{1}{2\eta_{\mathbf{x}}}\|\mathbf{x}_{t+1} - \mathbf{x}_t\|^2 - \frac{5}{16}\eta_{\mathbf{x}}\|\widehat{\nabla}\Phi(\mathbf{z}_t)\|^2 + 12\eta_{\mathbf{x}}\|\delta_{\widehat{\Phi}}\|^2$$

$$\tag{49}$$

where we use $\mathcal{L} \leq \frac{1}{4\eta_{\mathbf{x}}}$ and $\|\mathbf{x}_t - \mathbf{z}_t\| = (1 - \theta_{\mathbf{x}})\|\mathbf{x}_t - \mathbf{x}_{t-1}\| \leq \|\mathbf{x}_t - \mathbf{x}_{t-1}\|$. Summing over $t = t_0, ..., t_{\mathcal{K}}$ and using $\mathbf{x}_{t_0} = \mathbf{x}_{t_0-1}$, we have

$$\Phi(\mathbf{x}_{t_{\mathcal{K}}+1}) - \Phi(\mathbf{x}_{t_0}) \leq \frac{1}{4\eta_{\mathbf{x}}}\sum_{k=t_0}^{t_{\mathcal{K}}-1}\|\mathbf{x}_{k+1} - \mathbf{x}_k\|^2 - \frac{5\eta_{\mathbf{x}}}{16}\sum_{k=t_0}^{t_{\mathcal{K}}}\|\widehat{\nabla}\Phi(\mathbf{z}_k)\|^2 + 12\eta_{\mathbf{x}}\sum_{k=t_0}^{t_{\mathcal{K}}}\|\delta_{\widehat{\Phi}}\|^2$$

$$\leq \frac{B^2}{4\eta_{\mathbf{x}}} - \frac{5}{16}\eta_{\mathbf{x}}\|\widehat{\nabla}\Phi(\mathbf{z}_{t_{\mathcal{K}}})\|^2 + 12\eta_{\mathbf{x}}K\|\delta_{\widehat{\Phi}}\|^2$$

$$\leq \frac{B^2}{4\eta_{\mathbf{x}}} - \frac{5B^2}{16\eta_{\mathbf{x}}} + 12\eta_{\mathbf{x}}K\|\delta_{\widehat{\Phi}}\|^2$$

$$\tag{50}$$

$$\leq -\frac{5B^2}{128\eta_{\mathbf{x}}}$$

**Lemma C.2** *Running Algorithm 1 with parameters setting in Theorem 4.3. When the condition on Line 9 triggers, denote $t_{\mathcal{K}}$ to be the iteration number, $\mathcal{K}$ to be the value of $k$ on that iteration and $t_0 = t_{\mathcal{K}} - \mathcal{K} + 1$. If $\|\widehat{\nabla}\Phi(\mathbf{z}_{t_{\mathcal{K}}})\| \leq \frac{B}{\eta_{\mathbf{x}}}$, denote $\mathbf{H} = \nabla^2\Phi(\mathbf{x}_{t_0})$ and $\mathbf{H} = \mathbf{U}\boldsymbol{\Lambda}\mathbf{U}^T$ to be its eigenvalue decomposition with $\mathbf{U}, \boldsymbol{\Lambda} \in \mathbb{R}^{d\times d}$. Define the quadratic approximation function $g$ as*

$$g(\mathbf{x}) = \left\langle \widetilde{\nabla}\Phi(\mathbf{x}_{t_0}), \mathbf{x} - \tilde{\mathbf{x}}_{t_0} \right\rangle + \frac{1}{2}(\mathbf{x} - \tilde{\mathbf{x}}_{t_0})^T\boldsymbol{\Lambda}(\mathbf{x} - \tilde{\mathbf{x}}_{t_0})$$

*where we denote $\tilde{\mathbf{x}} = \mathbf{U}^T\mathbf{x}$, $\tilde{\mathbf{z}} = \mathbf{U}^T\mathbf{z}$, $\widetilde{\nabla}\Phi(\mathbf{z}) = \mathbf{U}^T\nabla\Phi(\mathbf{z})$ and $\widetilde{\widehat{\nabla}}\Phi(\mathbf{z}) = \mathbf{U}^T\widehat{\nabla}\Phi(\mathbf{z})$. Then, the approximation error $\widetilde{\delta}_t = \widetilde{\nabla}\Phi(\mathbf{z}_t) - \nabla g(\tilde{\mathbf{z}}_t)$ at iteration $t$ can be bounded as $\|\widetilde{\delta}_t\| \leq \frac{9}{4}\rho B^2$, where $\rho = \rho_{\boldsymbol{\Phi},0} + 2\rho_{\boldsymbol{\Phi},1}G$ denotes the efficient hessian smoothness constant.*

*Proof.* If $\|\widehat{\nabla}\Phi(\mathbf{z}_{t_{\mathcal{K}}})\| \leq \frac{B}{\eta_{\mathbf{x}}}$, from the AGD iteration we have

$$\|\mathbf{x}_{t_{\mathcal{K}}+1} - \mathbf{x}_{t_0}\| \leq \|\mathbf{z}_{t_{\mathcal{K}}} - \mathbf{x}_{t_0}\| + \eta_{\mathbf{x}}\|\widehat{\nabla}\Phi(\mathbf{z}_{t_{\mathcal{K}}})\| \leq 3B \tag{51}$$

From the generalized Hessian smoothness condition and Lemma A.3 we have

$$\Phi(\mathbf{x}_{t_{\mathcal{K}}+1}) - \Phi(\mathbf{x}_{t_0})$$

$$\leq \langle \nabla\Phi(\mathbf{x}_{t_0}), \mathbf{x}_{t_{\mathcal{K}}+1} - \mathbf{x}_{t_0} \rangle + \frac{1}{2}(\mathbf{x}_{t_{\mathcal{K}}+1} - \mathbf{x}_{t_0})^T\mathbf{H}(\mathbf{x}_{t_{\mathcal{K}}+1} - \mathbf{x}_{t_0}) + \frac{\rho}{6}\|\mathbf{x}_{t_{\mathcal{K}}+1} - \mathbf{x}_{t_0}\|^3$$

$$\leq \left\langle \widetilde{\nabla}\Phi(\mathbf{x}_{t_0}), \widetilde{\mathbf{x}}_{t_{\mathcal{K}}+1} - \widetilde{\mathbf{x}}_{t_0} \right\rangle + \frac{1}{2}(\widetilde{\mathbf{x}}_{t_{\mathcal{K}}+1} - \widetilde{\mathbf{x}}_{t_0})^T\boldsymbol{\Lambda}(\widetilde{\mathbf{x}}_{t_{\mathcal{K}}+1} - \widetilde{\mathbf{x}}_{t_0}) + \frac{\rho}{6}\|\mathbf{x}_{t_{\mathcal{K}}+1} - \mathbf{x}_{t_0}\|^3$$

$$\tag{52}$$

$$\leq g(\widetilde{\mathbf{x}}_{t_{\mathcal{K}}+1}) - g(\widetilde{\mathbf{x}}_{t_0}) + 4.5\rho B^3$$

where $\rho$ is the effective Hessian smoothness constant. Let $\lambda_j$ be the $j$th eigenvalue. Denote

$$g^{(j)}(x) = \left\langle \widetilde{\nabla}^{(j)}\Phi(\mathbf{x}_{t_0}), x - \widetilde{\mathbf{x}}_{t_0}^{(j)} \right\rangle + \frac{1}{2}\lambda^{(j)}(x - \widetilde{\mathbf{x}}_{t_0}^{(j)})^2$$

$$\widetilde{\delta}_t^{(j)} = \widetilde{\nabla}^{(j)}\Phi(\mathbf{z}_t) - \nabla g_j(\widetilde{\mathbf{z}}_t^{(j)})$$

Then the AGD iterations can be rewritten as

$$\widetilde{\mathbf{z}}_t^{(j)} = \widetilde{\mathbf{x}}_t^{(j)} + (1 - \theta_{\mathbf{x}})(\widetilde{\mathbf{x}}_t^{(j)} - \widetilde{\mathbf{x}}_{t-1}^{(j)}),$$

$$\widetilde{\mathbf{x}}_{t+1}^{(j)} = \widetilde{\mathbf{z}}_t^{(j)} - \eta_{\mathbf{x}}\widehat{\widetilde{\nabla}}_j\Phi(\mathbf{z}_t) = \widetilde{\mathbf{z}}_t^{(j)} - \eta_{\mathbf{x}}\nabla g_j(\widetilde{\mathbf{z}}_t^{(j)}) - \eta_{\mathbf{x}}\widetilde{\delta}_t^{(j)}$$

and $\|\widetilde{\delta}_t\|$ can be bounded as

$$
\begin{aligned}
\|\widetilde{\delta}_t\| &= \|\widehat{\widetilde{\nabla}}\Phi(\mathbf{z}_t) - \widetilde{\nabla}\Phi(\mathbf{x}_{t_0}) - \mathbf{\Lambda}(\widetilde{\mathbf{z}}_t - \widetilde{\mathbf{x}}_{t_0})\| \\
&= \|\widehat{\nabla}\Phi(\mathbf{z}_t) - \nabla\Phi(\mathbf{x}_{t_0}) - \mathbf{H}(\mathbf{z}_t - \mathbf{x}_{t_0})\| \\
&= \|\nabla\Phi(\mathbf{z}_t) - \nabla\Phi(\mathbf{x}_{t_0}) + \widehat{\nabla}\Phi(\mathbf{z}_t) - \nabla\Phi(\mathbf{z}_t) + \mathbf{H}(\mathbf{z}_t - \mathbf{x}_{t_0})\| \\
&\leq \|(\int_0^1 \nabla^2\Phi(\mathbf{x}_{t_0} + t(\mathbf{z}_t - \mathbf{x}_{t_0})) - \mathbf{H})(\mathbf{z}_t - \mathbf{x}_{t_0})dt\| + \|\widehat{\nabla}\Phi(\mathbf{z}_t) - \nabla\Phi(\mathbf{z}_t)\| \\
&\leq \frac{\rho}{2}\|\mathbf{z}_t - \mathbf{x}_{t_0}\|^2 + \|\widehat{\nabla}\Phi(\mathbf{z}_t) - \nabla\Phi(\mathbf{z}_t)\| \leq \frac{9}{4}\rho B^2
\end{aligned}
\tag{53}
$$

To prove the decrease from $\Phi(\mathbf{x}_{t_0})$ to $\Phi(\mathbf{x}_{t_{\mathcal{K}}+1})$, we only need to study the decrease of the quadratic approximation function $g(\mathbf{x})$. The quadratic function $g(\mathbf{x})$ equals to the sum of $d$ scalar functions $g^{(j)}(\mathbf{x}^{(j)})$. We decompose $g(\mathbf{x})$ into $\sum_{j\in\mathcal{S}_1} g^{(j)}(\mathbf{x}^{(j)})$ and $\sum_{j\in\mathcal{S}_2} g^{(j)}(\mathbf{x}^{(j)})$, where $\mathcal{S}_1 = \left\{j : \lambda_j \geq -\frac{\theta_{\mathbf{x}}}{\eta_{\mathbf{x}}}\right\}$ and $\mathcal{S}_2 = \left\{j : \lambda_j < -\frac{\theta_{\mathbf{x}}}{\eta_{\mathbf{x}}}\right\}$. We see that $g^{(j)}(x)$ is approximate convex when $j \in \mathcal{S}_1$, and strongly concave when $j \in \mathcal{S}_2$. We will prove the approximate decrease of $g^{(j)}(\mathbf{x}^{(j)})$ in the two cases. We first consider $\sum_{j\in\mathcal{S}_1} g^{(j)}(\mathbf{x}^{(j)})$.

**Lemma C.3** *Running Algorithm 1 with parameters setting in Theorem 4.3. When the condition on Line 9 triggers, denote $t_{\mathcal{K}}$ to be the iteration number, $\mathcal{K}$ to be the value of $k$ on that iteration and $t_0 = t_{\mathcal{K}} - \mathcal{K} + 1$. If $\|\widehat{\nabla}\Phi(\mathbf{z}_{t_{\mathcal{K}}})\| \leq \frac{B}{\eta_{\mathbf{x}}}$, we have*

$$\sum_{j\in\mathcal{S}_1} g^{(j)}(\widetilde{\mathbf{x}}_{t_{\mathcal{K}}+1}^{(j)}) \leq \sum_{j\in\mathcal{S}_1} g^{(j)}(\widetilde{\mathbf{x}}_{t_0}^{(j)}) - \sum_{j\in\mathcal{S}_1} \frac{3\theta_{\mathbf{x}}}{8\eta_{\mathbf{x}}}\sum_{k=t_0}^{t_{\mathcal{K}}}\|\widetilde{\mathbf{x}}_{k+1}^{(j)} - \widetilde{\mathbf{x}}_k^{(j)}\|^2 + \frac{9\eta_{\mathbf{x}}\rho^2 B^4 \mathcal{K}}{\theta_{\mathbf{x}}} \tag{54}$$

*Proof.* Since $g^{(j)}(x)$ is quadratic, we have

$$
\begin{aligned}
g^{(j)}(\tilde{\mathbf{x}}_{t+1}^{(j)}) &= g^{(j)}(\tilde{\mathbf{x}}_t^{(j)}) + \left\langle \nabla^{(j)}g(\tilde{\mathbf{x}}_t^{(j)}), \tilde{\mathbf{x}}_{t+1}^{(j)} - \tilde{\mathbf{x}}_t^{(j)} \right\rangle + \frac{\lambda_j}{2}\|\tilde{\mathbf{x}}_{t+1}^{(j)} - \tilde{\mathbf{x}}_t^{(j)}\|^2 \\
&\stackrel{a}{=} g^{(j)}(\tilde{\mathbf{x}}_t^{(j)}) - \frac{1}{\eta_{\mathbf{x}}}\left\langle \tilde{\mathbf{x}}_{t+1}^{(j)} - \tilde{\mathbf{z}}_t^{(j)} + \eta_{\mathbf{x}}\delta_t^{(j)}, \tilde{\mathbf{x}}_{t+1}^{(j)} - \tilde{\mathbf{x}}_t^{(j)} \right\rangle \\
&\quad + \left\langle \nabla^{(j)}g(\tilde{\mathbf{x}}_t^{(j)}) - \nabla^{(j)}g(\tilde{\mathbf{z}}_t^{(j)}), \tilde{\mathbf{x}}_{t+1}^{(j)} - \tilde{\mathbf{x}}_t^{(j)} \right\rangle + \frac{\lambda_j}{2}\|\tilde{\mathbf{x}}_{t+1}^{(j)} - \tilde{\mathbf{x}}_t^{(j)}\|^2 \\
&= g^{(j)}(\tilde{\mathbf{x}}_t^{(j)}) - \frac{1}{\eta_{\mathbf{x}}}\left\langle \tilde{\mathbf{x}}_{t+1}^{(j)} - \tilde{\mathbf{z}}_t^{(j)}, \tilde{\mathbf{x}}_{t+1}^{(j)} - \tilde{\mathbf{x}}_t^{(j)} \right\rangle - \left\langle \delta_t^{(j)}, \tilde{\mathbf{x}}_{t+1}^{(j)} - \tilde{\mathbf{x}}_t^{(j)} \right\rangle \\
&\quad + \lambda_j\left\langle \tilde{\mathbf{x}}_t^{(j)} - \tilde{\mathbf{z}}_t^{(j)}, \tilde{\mathbf{x}}_{t+1}^{(j)} - \tilde{\mathbf{x}}_t^{(j)} \right\rangle + \frac{\lambda_j}{2}\|\tilde{\mathbf{x}}_{t+1}^{(j)} - \tilde{\mathbf{x}}_t^{(j)}\|^2 \\
&\leq g^{(j)}(\tilde{\mathbf{x}}_t^{(j)}) + \frac{1}{2\eta_{\mathbf{x}}}(\|\tilde{\mathbf{x}}_t^{(j)} - \tilde{\mathbf{z}}_t^{(j)}\|^2 - \|\tilde{\mathbf{x}}_{t+1}^{(j)} - \tilde{\mathbf{z}}_t^{(j)}\|^2 - \|\tilde{\mathbf{x}}_{t+1}^{(j)} - \tilde{\mathbf{x}}_t^{(j)}\|^2) \\
&\quad + \frac{1}{2\alpha}\|\delta_t^{(j)}\|^2 + \frac{\alpha}{2}\|\tilde{\mathbf{x}}_{t+1}^{(j)} - \tilde{\mathbf{x}}_t^{(j)}\|^2 + \frac{\lambda_j}{2}(\|\tilde{\mathbf{x}}_{t+1}^{(j)} - \tilde{\mathbf{z}}_t^{(j)}\|^2 - \|\tilde{\mathbf{x}}_t^{(j)} - \tilde{\mathbf{z}}_t^{(j)}\|^2)
\end{aligned}
\tag{55}
$$

Using $\mathcal{L} \geq \lambda_j \geq -\frac{\theta_{\mathbf{x}}}{\eta_{\mathbf{x}}}$ when $j \in \mathcal{S}_1 = \left\{ j : \lambda_j \geq -\frac{\theta_{\mathbf{x}}}{\eta_{\mathbf{x}}} \right\}$ and $(-\frac{1}{2\eta_{\mathbf{x}}} + \frac{\lambda_j}{2})\|\tilde{\mathbf{x}}_{t+1}^{(j)} - \tilde{\mathbf{z}}_t^{(j)}\|^2 \leq$ $(-2\mathcal{L} + \frac{\mathcal{L}}{2})\|\tilde{\mathbf{x}}_{t+1}^{(j)} - \tilde{\mathbf{z}}_t^{(j)}\|^2 \leq 0$, we have for each $j \in \mathcal{S}_1$,

$$g^{(j)}(\tilde{\mathbf{x}}_{t+1}^{(j)}) \leq g^{(j)}(\tilde{\mathbf{x}}_t^{(j)}) + \frac{1}{2\eta_{\mathbf{x}}}(\|\tilde{\mathbf{x}}_t^{(j)} - \tilde{\mathbf{z}}_t^{(j)}\|^2 - \|\tilde{\mathbf{x}}_{t+1}^{(j)} - \tilde{\mathbf{x}}_t^{(j)}\|^2)$$

$$+ \frac{1}{2\alpha}\|\delta_t^{(j)}\|^2 + \frac{\alpha}{2}\|\tilde{\mathbf{x}}_{t+1}^{(j)} - \tilde{\mathbf{x}}_t^{(j)}\|^2 + \frac{\theta_{\mathbf{x}}}{2\eta_{\mathbf{x}}}\|\tilde{\mathbf{x}}_t^{(j)} - \tilde{\mathbf{z}}_t^{(j)}\|^2$$

$$\overset{b}{=} g^{(j)}(\tilde{\mathbf{x}}_t^{(j)}) + \frac{(1-\theta_{\mathbf{x}})^2(1+\theta_{\mathbf{x}})}{2\eta_{\mathbf{x}}}\|\tilde{\mathbf{x}}_t^{(j)} - \tilde{\mathbf{x}}_{t-1}^{(j)}\|^2$$

$$- (\frac{1}{2\eta_{\mathbf{x}}} - \frac{\alpha}{2})\|\tilde{\mathbf{x}}_{t+1}^{(j)} - \tilde{\mathbf{x}}_t^{(j)}\|^2 + \frac{1}{2\alpha}\|\delta_t^{(j)}\|^2 \tag{56}$$

Defining the potential function

$$p_{t+1}^{(j)} = g^{(j)}(\tilde{\mathbf{x}}_{t+1}^{(j)}) + \frac{(1-\theta_{\mathbf{x}})^2(1+\theta_{\mathbf{x}})}{2\eta_{\mathbf{x}}}\|\tilde{\mathbf{x}}_{t+1}^{(j)} - \tilde{\mathbf{x}}_t^{(j)}\|^2 \tag{57}$$

we have

$$p_{t+1}^{(j)} \leq p_t^{(j)} - (\frac{1}{2\eta_{\mathbf{x}}} - \frac{\alpha}{2} - \frac{(1-\theta_{\mathbf{x}})^2(1+\theta_{\mathbf{x}})}{2\eta_{\mathbf{x}}})\|\tilde{\mathbf{x}}_{t+1}^{(j)} - \tilde{\mathbf{x}}_t^{(j)}\|^2 + \|\frac{1}{2\alpha}\delta_t^{(j)}\|^2$$

$$\overset{c}{\leq} p_t^{(j)} - \frac{3\theta_{\mathbf{x}}}{8\eta_{\mathbf{x}}}\|\tilde{\mathbf{x}}_{t+1}^{(j)} - \tilde{\mathbf{x}}_t^{(j)}\|^2 + \frac{2\eta_{\mathbf{x}}}{\theta_{\mathbf{x}}}\|\delta_t^{(j)}\|^2, \tag{58}$$

where we let $\alpha = \frac{\theta_{\mathbf{x}}}{4\eta_{\mathbf{x}}}$ in $\overset{c}{\leq}$ such that $\frac{1}{2\eta_{\mathbf{x}}} - \frac{\theta_{\mathbf{x}}}{8\eta_{\mathbf{x}}} - \frac{(1-\theta_{\mathbf{x}})^2(1+\theta_{\mathbf{x}})}{2\eta_{\mathbf{x}}} = \frac{3\theta_{\mathbf{x}}}{8\eta_{\mathbf{x}}} + \frac{\theta_{\mathbf{x}}^2}{2\eta_{\mathbf{x}}} - \frac{\theta_{\mathbf{x}}^3}{2\eta_{\mathbf{x}}} \geq \frac{3\theta_{\mathbf{x}}}{8\eta_{\mathbf{x}}}$. Summing over $t = t_0, \cdots, t_{\mathcal{K}}$ and $j \in \mathcal{S}_1$, using $\mathbf{x}_{t_0} - \mathbf{x}_{t_0-1} = 0$, we have

$$\sum_{j \in \mathcal{S}_1} g^{(j)}(\tilde{\mathbf{x}}_{t_{\mathcal{K}}+1}^{(j)})) \leq \sum_{j \in \mathcal{S}_1} p_j^{\mathcal{K}} \leq \sum_{j \in \mathcal{S}_1} g^{(j)}(\tilde{\mathbf{x}}_{t_0}^{(j)})) - \sum_{j \in \mathcal{S}_1} \frac{3\theta_{\mathbf{x}}}{8\eta_{\mathbf{x}}} \sum_{k=t_0}^{t_{\mathcal{K}}} \|\tilde{\mathbf{x}}_{k+1}^{(j)} - \tilde{\mathbf{x}}_k^{(j)}\|^2 + \frac{2\eta_{\mathbf{x}}}{\theta_{\mathbf{x}}} \sum_{k=t_0}^{\mathcal{K}-1} \|\delta_t\|^2$$

$$\leq \sum_{j \in \mathcal{S}_1} g^{(j)}(\tilde{\mathbf{x}}_{t_0}^{(j)})) - \sum_{j \in \mathcal{S}_1} \frac{3\theta_{\mathbf{x}}}{8\eta_{\mathbf{x}}} \sum_{k=t_0}^{t_{\mathcal{K}}} \|\tilde{\mathbf{x}}_{k+1}^{(j)} - \tilde{\mathbf{x}}_k^{(j)}\|^2 + \frac{9\eta_{\mathbf{x}}\rho^2 B^4 \mathcal{K}}{\theta_{\mathbf{x}}} \tag{59}$$

Next, we consider $\sum_{j \in \mathcal{S}_2} g^{(j)}(\mathbf{x}^{(j)})$.

**Lemma C.4** *Running Algorithm 1 with parameters setting in Theorem 4.3. When the condition on Line 9 triggers, denote $t_{\mathcal{K}}$ to be the iteration number, $\mathcal{K}$ to be the value of $k$ on that iteration and $t_0 = t_{\mathcal{K}} - \mathcal{K} + 1$. If $\|\widehat{\nabla}\Phi(\mathbf{z}_{t_{\mathcal{K}}})\| \leq \frac{B}{\eta_{\mathbf{x}}}$, we have*

$$\sum_{j \in \mathcal{S}_2} g_j(\tilde{\mathbf{x}}_{t_{\mathcal{K}}+1}^{(j)}) - \sum_{j \in \mathcal{S}_2} g_j(\tilde{\mathbf{x}}_{t_0}^{(j)}) \leq -\sum_{j \in \mathcal{S}_2} \frac{\theta_{\mathbf{x}}}{2\eta} \sum_{k=t_0}^{t_{\mathcal{K}}} |\tilde{\mathbf{x}}_{k+1}^{(j)} - \tilde{\mathbf{x}}_k^{(j)}|^2 + \frac{9\eta\rho^2 B^4 \mathcal{K}}{4\theta_{\mathbf{x}}} \tag{60}$$

*Proof.* Denoting $\mathbf{v}_j = \tilde{\mathbf{x}}_{t_0}^{(j)} - \frac{1}{\lambda_j}\widetilde{\nabla}^{(j)}\Phi(\mathbf{x}_{t_0})$, $g^{(j)}(x)$ can be rewritten as

$$g^{(j)}(x) = \frac{\lambda_j}{2}(x - \tilde{\mathbf{x}}_{t_0}^{(j)} + \frac{1}{\lambda_j}\widetilde{\nabla}^{(j)}\Phi(\mathbf{x}_{t_0}))^2 - \frac{1}{2\lambda_j}\|\widetilde{\nabla}_j\Phi(\mathbf{x}_{t_0})\|^2$$

$$= \frac{\lambda_j}{2}\|x - \mathbf{v}_j\|^2 - \frac{1}{2\lambda_j}\|\widetilde{\nabla}^{(j)}\Phi(\mathbf{x}_{t_0})\|^2 \tag{61}$$

For each $j \in \mathcal{S}_2 = \left\{ j : \lambda_j < -\frac{\theta_{\mathbf{x}}}{\eta_{\mathbf{x}}} \right\}$, we have

$$g^{(j)}(\tilde{\mathbf{x}}_{t+1}^{(j)}) - g^{(j)}(\tilde{\mathbf{x}}_t^{(j)}) = \frac{\lambda_j}{2}\|\tilde{\mathbf{x}}_{t+1}^{(j)} - \mathbf{v}_j\|^2 - \frac{\lambda_j}{2}\|\tilde{\mathbf{x}}_t^{(j)} - \mathbf{v}_j\|^2$$

$$= \frac{\lambda_j}{2}\|\tilde{\mathbf{x}}_{t+1}^{(j)} - \tilde{\mathbf{x}}_t^{(j)}\|^2 + \lambda_j \left\langle \tilde{\mathbf{x}}_{t+1}^{(j)} - \tilde{\mathbf{x}}_t^{(j)}, \tilde{\mathbf{x}}_t^{(j)} - \mathbf{v}_j \right\rangle \tag{62}$$

$$\leq -\frac{\theta_{\mathbf{x}}}{2\eta_{\mathbf{x}}}\|\tilde{\mathbf{x}}_{t+1}^{(j)} - \tilde{\mathbf{x}}_t^{(j)}\|^2 + \lambda_j \left\langle \tilde{\mathbf{x}}_{t+1}^{(j)} - \tilde{\mathbf{x}}_t^{(j)}, \tilde{\mathbf{x}}_t^{(j)} - \mathbf{v}_j \right\rangle.$$

So we only need to bound the second term, where

$$
\begin{aligned}
\tilde{\mathbf{x}}_{t+1}^{(j)} - \tilde{\mathbf{x}}_t^{(j)} &= \tilde{\mathbf{y}}_t^{(j)} - \tilde{\mathbf{x}}_t^{(j)} - \eta_{\mathbf{x}} \nabla^{(j)} g(\tilde{\mathbf{y}}_t^{(j)}) - \eta_{\mathbf{x}} \delta_t^{(j)} \\
&= (1 - \theta_{\mathbf{x}})(\tilde{\mathbf{x}}_t^{(j)} - \tilde{\mathbf{x}}_{t-1}^{(j)}) - \eta_{\mathbf{x}} \nabla^{(j)} g(\tilde{\mathbf{y}}_t^{(j)}) - \eta_{\mathbf{x}} \delta_t^{(j)} \\
&= (1 - \theta_{\mathbf{x}})(\tilde{\mathbf{x}}_t^{(j)} - \tilde{\mathbf{x}}_{t-1}^{(j)}) - \eta_{\mathbf{x}} \lambda_j (\tilde{\mathbf{y}}_t^{(j)} - \mathbf{v}_j) - \eta_{\mathbf{x}} \delta_t^{(j)} \\
&= (1 - \theta_{\mathbf{x}})(\tilde{\mathbf{x}}_t^{(j)} - \tilde{\mathbf{x}}_{t-1}^{(j)}) - \eta_{\mathbf{x}} \lambda_j (\tilde{\mathbf{x}}_t^{(j)} - \mathbf{v}_j + (1 - \theta_{\mathbf{x}})(\tilde{\mathbf{x}}_t^{(j)} - \tilde{\mathbf{x}}_{t-1}^{(j)})) - \eta_{\mathbf{x}} \delta_t^{(j)}
\end{aligned}
\tag{63}
$$

So for each $j \in \mathcal{S}_2$, we have

$$
\begin{aligned}
&\lambda_j \left\langle \tilde{\mathbf{x}}_{t+1}^{(j)} - \tilde{\mathbf{x}}_t^{(j)}, \tilde{\mathbf{x}}_t^{(j)} - \mathbf{v}_j \right\rangle \\
&= (1 - \theta_{\mathbf{x}}) \lambda_j \left\langle \tilde{\mathbf{x}}_t^{(j)} - \tilde{\mathbf{x}}_{t-1}^{(j)}, \tilde{\mathbf{x}}_t^{(j)} - \mathbf{v}_j \right\rangle - \eta_{\mathbf{x}} \lambda_j^2 \| \tilde{\mathbf{x}}_t^{(j)} - \mathbf{v}_j \|^2 \\
&\quad - \eta_{\mathbf{x}} \lambda_j^2 (1 - \theta_{\mathbf{x}}) \left\langle \tilde{\mathbf{x}}_t^{(j)} - \tilde{\mathbf{x}}_{t-1}^{(j)}, \tilde{\mathbf{x}}_t^{(j)} - \mathbf{v}_j \right\rangle - \eta_{\mathbf{x}} \lambda_j \left\langle \delta_t^{(j)}, \tilde{\mathbf{x}}_t^{(j)} - \mathbf{v}_j \right\rangle \\
&\leq (1 - \theta_{\mathbf{x}}) \lambda_j \left\langle \tilde{\mathbf{x}}_t^{(j)} - \tilde{\mathbf{x}}_{t-1}^{(j)}, \tilde{\mathbf{x}}_t^{(j)} - \mathbf{v}_j \right\rangle - \eta_{\mathbf{x}} \lambda_j^2 \| \tilde{\mathbf{x}}_t^{(j)} - \mathbf{v}_j \|^2 \\
&\quad + \frac{\eta_{\mathbf{x}} \lambda_j^2 (1 - \theta_{\mathbf{x}})}{2} (\| \tilde{\mathbf{x}}_t^{(j)} - \tilde{\mathbf{x}}_{t-1}^{(j)} \|^2 + \| \tilde{\mathbf{x}}_t^{(j)} - \mathbf{v}_j \|^2) \\
&\quad + \frac{\eta_{\mathbf{x}}}{2(1 + \theta_{\mathbf{x}})} \| \delta_t^{(j)} \|^2 + \frac{\eta_{\mathbf{x}} \lambda_j^2 (1 + \theta_{\mathbf{x}})}{2} \| \tilde{\mathbf{x}}_t^{(j)} - \mathbf{v}_j \|^2 \\
&= (1 - \theta_{\mathbf{x}}) \lambda_j \left\langle \tilde{\mathbf{x}}_t^{(j)} - \tilde{\mathbf{x}}_{t-1}^{(j)}, \tilde{\mathbf{x}}_t^{(j)} - \mathbf{v}_j \right\rangle \\
&\quad + \frac{\eta_{\mathbf{x}} \lambda_j^2 (1 - \theta_{\mathbf{x}})}{2} \| \tilde{\mathbf{x}}_t^{(j)} - \tilde{\mathbf{x}}_{t-1}^{(j)} \|^2 + \frac{\eta_{\mathbf{x}}}{2(1 + \theta_{\mathbf{x}})} \| \delta_t^{(j)} \|^2 \\
&= (1 - \theta_{\mathbf{x}}) \lambda_j \left\langle \tilde{\mathbf{x}}_t^{(j)} - \tilde{\mathbf{x}}_{t-1}^{(j)}, \tilde{\mathbf{x}}_{t-1}^{(j)} - \mathbf{v}_j \right\rangle + (1 - \theta_{\mathbf{x}}) \lambda_j \| \tilde{\mathbf{x}}_t^{(j)} - \tilde{\mathbf{x}}_{t-1}^{(j)} \|^2 \\
&\quad + \frac{\eta_{\mathbf{x}} \lambda_j^2 (1 - \theta_{\mathbf{x}})}{2} \| \tilde{\mathbf{x}}_t^{(j)} - \tilde{\mathbf{x}}_{t-1}^{(j)} \|^2 + \frac{\eta_{\mathbf{x}}}{2(1 + \theta_{\mathbf{x}})} \| \delta_t^{(j)} \|^2 \\
&\leq (1 - \theta_{\mathbf{x}}) \lambda_j \left\langle \tilde{\mathbf{x}}_t^{(j)} - \tilde{\mathbf{x}}_{t-1}^{(j)}, \tilde{\mathbf{x}}_{t-1}^{(j)} - \mathbf{v}_j \right\rangle + \frac{\eta_{\mathbf{x}}}{2} \| \delta_t^{(j)} \|^2,
\end{aligned}
\tag{64}
$$

where we use $(1 + \frac{\eta_{\mathbf{x}} \lambda_j}{2})(1 - \theta_{\mathbf{x}}) \geq (1 - \frac{\eta_{\mathbf{x}} \mathcal{L}}{2})(1 - \theta_{\mathbf{x}}) \geq 0$ and $\lambda_j < 0$ when $j \in \mathcal{S}_2$. Then,

$$
\begin{aligned}
\lambda_j \left\langle \tilde{\mathbf{x}}_{t+1}^{(j)} - \tilde{\mathbf{x}}_t^{(j)}, \tilde{\mathbf{x}}_t^{(j)} - \mathbf{v}_j \right\rangle &\leq (1 - \theta_{\mathbf{x}})^k \lambda_j \left\langle \tilde{\mathbf{x}}_{t_0+1}^{(j)} - \tilde{\mathbf{x}}_{t_0}^{(j)}, \tilde{\mathbf{x}}_{t_0}^{(j)} - \mathbf{v}_j \right\rangle + \frac{\eta_{\mathbf{x}}}{2} \sum_{t=1}^k (1 - \theta_{\mathbf{x}})^{k-t} \| \delta_t^{(j)} \|^2 \\
&\overset{b}{=} -(1 - \theta_{\mathbf{x}})^k \eta_{\mathbf{x}} \lambda_j^2 \| \tilde{\mathbf{x}}_{t_0}^{(j)} - \mathbf{v}_j \|^2 + \frac{\eta_{\mathbf{x}}}{2} \sum_{t=1}^k (1 - \theta_{\mathbf{x}})^{k-t} \| \delta_t^{(j)} \|^2 \\
&\leq \frac{\eta_{\mathbf{x}}}{2} \sum_{t=1}^k (1 - \theta_{\mathbf{x}})^{k-t} \| \delta_t^{(j)} \|^2,
\end{aligned}
\tag{65}
$$

where $\overset{b}{=}$ holds by using

$$
\begin{aligned}
\tilde{\mathbf{x}}_{t_0+1}^{(j)} - \tilde{\mathbf{x}}_{t_0}^{(j)} &= \tilde{\mathbf{x}}_{t_0+1}^{(j)} - \tilde{\mathbf{z}}_{t_0}^{(j)} = -\eta_{\mathbf{x}} \widetilde{\nabla}^{(j)} \Phi(\mathbf{z}_{t_0}) = -\eta_{\mathbf{x}} \widetilde{\nabla}^{(j)} \Phi(\mathbf{x}_{t_0}) \\
&= -\eta_{\mathbf{x}} \nabla^{(j)} g(\tilde{\mathbf{x}}_{t_0}^{(j)}) = -\eta_{\mathbf{x}} \lambda_j (\tilde{\mathbf{x}}_{t_0}^{(j)} - \mathbf{v}_j).
\end{aligned}
\tag{66}
$$

Plugging (65) into (62), we have

$$
g^{(j)}(\tilde{\mathbf{x}}_{t+1}^{(j)}) - g^{(j)}(\tilde{\mathbf{x}}_t^{(j)}) \leq -\frac{\theta_{\mathbf{x}}}{2\eta_{\mathbf{x}}} \| \tilde{\mathbf{x}}_{t+1}^{(j)} - \tilde{\mathbf{x}}_t^{(j)} \|^2 + \frac{\eta_{\mathbf{x}}}{2} \sum_{t=t_0+1}^k (1 - \theta_{\mathbf{x}})^{k-t} \| \delta_t^{(j)} \|^2
\tag{67}
$$

Summing over $t = t_0, \cdots, t_{\mathcal{K}}$ and $j \in \mathcal{S}_2$, we have

$$
\sum_{j \in \mathcal{S}_2} g^{(j)}(\tilde{\mathbf{x}}_{t_{\mathcal{K}}+1}^{(j)}) - \sum_{j \in \mathcal{S}_2} g^{(j)}(\tilde{\mathbf{x}}_{t_0}^{(j)}) \leq - \sum_{j \in \mathcal{S}_2} \frac{\theta_{\mathbf{x}}}{2\eta_{\mathbf{x}}} \sum_{k=t_0}^{t_{\mathcal{K}}} \|\tilde{\mathbf{x}}_{k+1}^{(j)} - \tilde{\mathbf{x}}_k^{(j)}\|^2 + \frac{\eta_{\mathbf{x}}}{2} \sum_{k=t_0}^{t_{\mathcal{K}}} \sum_{i=t_0+1}^{k} (1-\theta_{\mathbf{x}})^{k-i} \|\delta_k\|^2
$$

$$
\leq - \sum_{j \in \mathcal{S}_2} \frac{\theta_{\mathbf{x}}}{2\eta_{\mathbf{x}}} \sum_{k=t_0}^{t_{\mathcal{K}}} \|\tilde{\mathbf{x}}_{k+1}^{(j)} - \tilde{\mathbf{x}}_k^{(j)}\|^2 + \frac{\eta_{\mathbf{x}} \mathcal{K}}{2\theta_{\mathbf{x}}} \|\delta_t\|^2
$$

$$
\leq - \sum_{j \in \mathcal{S}_2} \frac{\theta_{\mathbf{x}}}{2\eta_{\mathbf{x}}} \sum_{k=t_0}^{t_{\mathcal{K}}} \|\tilde{\mathbf{x}}_{k+1}^{(j)} - \tilde{\mathbf{x}}_k^{(j)}\|^2 + \frac{9\eta_{\mathbf{x}} \rho^2 B^4 \mathcal{K}}{4\theta_{\mathbf{x}}}
$$

$$\tag{68}$$

Puts Lemma C.3 and C.4 together, we introduce the following lemma.

**Lemma C.5** *Running Algorithm 1 with parameters setting in Theorem 4.3. When the condition on Line 9 triggers, denote $t_{\mathcal{K}}$ to be the iteration number, $\mathcal{K}$ to be the value of $k$ on that iteration and $t_0 = t_{\mathcal{K}} - \mathcal{K} + 1$. If $\|\widehat{\nabla}\Phi(\mathbf{z}_{t_{\mathcal{K}}})\| \leq \frac{B}{\eta_{\mathbf{x}}}$, we have*

$$
\Phi(\mathbf{x}_{t_{\mathcal{K}}+1}) - \Phi(\mathbf{x}_{t_0}) \leq -\frac{3\theta_{\mathbf{x}} B^2}{8\eta_{\mathbf{x}} K} + \frac{9\rho B^3}{2} + \frac{45\eta_{\mathbf{x}} \rho^2 B^4 K}{4\theta_{\mathbf{x}}} \tag{69}
$$

Proof. Summing over (54) and (60), we have

$$
g(\tilde{\mathbf{x}}_{t_{\mathcal{K}}+1}) - g(\tilde{\mathbf{x}}_{t_0}) = \sum_{j \in \mathcal{S}_1 \cup \mathcal{S}_2} g_j(\tilde{\mathbf{x}}_{t_{\mathcal{K}}+1}^{(j)}) - g_j(\tilde{\mathbf{x}}_{t_0}^{(j)})
$$

$$
\leq -\frac{3\theta_{\mathbf{x}}}{8\eta_{\mathbf{x}}} \sum_{k=t_0}^{t_{\mathcal{K}}} \|\tilde{\mathbf{x}}_{k+1} - \tilde{\mathbf{x}}_k\|^2 + \frac{45\eta_{\mathbf{x}} \rho^2 B^4 \mathcal{K}}{4\theta_{\mathbf{x}}}
$$

$$
= -\frac{3\theta_{\mathbf{x}}}{8\eta_{\mathbf{x}}} \sum_{k=t_0}^{t_{\mathcal{K}}} \|\mathbf{x}_{k+1} - \mathbf{x}_k\|^2 + \frac{45\eta_{\mathbf{x}} \rho^2 B^4 \mathcal{K}}{4\theta_{\mathbf{x}}}
$$

$$
\leq -\frac{3\theta_{\mathbf{x}} B^2}{8\eta_{\mathbf{x}} \mathcal{K}} + \frac{45\eta_{\mathbf{x}} \rho^2 B^4 \mathcal{K}}{4\theta_{\mathbf{x}}},
$$

$$\tag{70}$$

where the second equility holds from the definition of $\tilde{\mathbf{x}}$. Pluging into (52) and using $\mathcal{K} \leq K$, we have

$$
\Phi(\mathbf{x}_{t_{\mathcal{K}}+1}) - \Phi(\mathbf{x}_{t_0}) \leq -\frac{3\theta_{\mathbf{x}} B^2}{8\eta_{\mathbf{x}} \mathcal{K}} + \frac{9\rho B^3}{2} + \frac{45\eta_{\mathbf{x}} \rho^2 B^4 \mathcal{K}}{4\theta_{\mathbf{x}}} \tag{71}
$$

Then we can establish the decrease of $\Phi(\mathbf{x})$ in epochs that Line 9 triggers.

## C.1 PROOF OF LEMMA 4.5

**Lemma C.6** *Running Algorithm 1 with parameters setting in Theorem 4.3. When the condition on Line 9 triggers, denote $t_{\mathcal{K}}$ to be the iteration number, $\mathcal{K}$ to be the value of $k$ on that iteration and $t_0 = t_{\mathcal{K}} - \mathcal{K} + 1$. In each epoch of Algorithm 1 where the Line 9 triggers, we have*

$$
\Phi(\mathbf{x}_{t_{\mathcal{K}}+1}) - \Phi(\mathbf{x}_{t_0}) \leq -\frac{51\epsilon^{3/2}}{64\sqrt{\rho}} \tag{72}
$$

Proof. Combing two lemmas togethers, we have

$$
\Phi(\mathbf{x}_{t_{\mathcal{K}}+1}) - \Phi(\mathbf{x}_{t_0}) \leq -\min\left\{ \frac{3\theta_{\mathbf{x}} B^2}{8\eta_{\mathbf{x}} K} - \frac{9\rho B^3}{2} - \frac{45\eta_{\mathbf{x}} \rho^2 B^4 K}{4\theta_{\mathbf{x}}}, \frac{5B^2}{128\eta_{\mathbf{x}}} \right\}
$$

$$
= -\min\left\{ \frac{51\epsilon^{3/2}}{64\sqrt{\rho}}, \frac{5\epsilon}{128\eta_{\mathbf{x}}\rho} \right\}
$$

$$\tag{73}$$

Taking $\theta_{\mathbf{x}} = 4\left(\epsilon \rho \eta_{\mathbf{x}}^2\right)^{1/4} \leq 1$ we have

$$
\Phi(\mathbf{x}_{t_{\mathcal{K}}+1}) - \Phi(\mathbf{x}_{t_0}) \leq -\frac{51\epsilon^{3/2}}{64\sqrt{\rho}} \tag{74}
$$

Then we finish the proof.

## C.2 PROOF OF LEMMA 4.6

**Lemma C.7** *Running Algorithm 1 with parameters setting in Theorem 4.3. In the epoch that the condition on Line 11 triggers, the point $\hat{\mathbf{z}}$ in Line 13 satisfies $\|\nabla\Phi(\hat{\mathbf{z}})\| \leq \mathcal{O}(\epsilon)$.*

*Proof.* Denote $\tilde{\mathbf{z}} = \mathbf{U}^T\hat{\mathbf{z}} = \frac{1}{K_0+1}\sum_{k=t_0}^{t_0+K_0}\mathbf{U}^T\mathbf{z}_t = \frac{1}{K_0+1}\sum_{k=t_0}^{t_0+K_0}\tilde{\mathbf{z}}_t$. Since $g$ is quadratic, we have

$$
\begin{aligned}
\|\nabla g(\tilde{\mathbf{z}})\| &= \left\|\frac{1}{K_0+1}\sum_{k=t_0}^{t_0+K_0}\nabla g(\tilde{\mathbf{z}}_k)\right\| \\
&= \frac{1}{\eta_{\mathbf{x}}(K_0+1)}\left\|\sum_{k=t_0}^{t_0+K_0}(\tilde{\mathbf{x}}_{k+1} - \tilde{\mathbf{z}}_k + \eta_{\mathbf{x}}\tilde{\delta}_k)\right\| \\
&\stackrel{a}{=} \frac{1}{\eta_{\mathbf{x}}(K_0+1)}\left\|\sum_{k=t_0}^{t_0+K_0}(\tilde{\mathbf{x}}_{k+1} - \tilde{\mathbf{x}}_k + \eta_{\mathbf{x}}\tilde{\delta}_k) - \sum_{k=t_0+1}^{t_0+K_0}(1-\theta_{\mathbf{x}})(\tilde{\mathbf{x}}_k - \tilde{\mathbf{x}}_{k-1})\right\| \\
&= \frac{1}{\eta_{\mathbf{x}}(K_0+1)}\left\|\tilde{\mathbf{x}}_{t_0+K_0+1} - \tilde{\mathbf{x}}_{t_0} - (1-\theta_{\mathbf{x}})(\tilde{\mathbf{x}}_{t_0+K_0} - \tilde{\mathbf{x}}_{t_0}) + \eta_{\mathbf{x}}\sum_{k=t_0}^{t_0+K_0}\tilde{\delta}_k\right\| \\
&= \frac{1}{\eta_{\mathbf{x}}(K_0+1)}\left\|\tilde{\mathbf{x}}_{t_0+K_0+1} - \tilde{\mathbf{x}}_{t_0+K_0} + \theta_{\mathbf{x}}(\tilde{\mathbf{x}}_{t_0+K_0} - \tilde{\mathbf{x}}_{t_0}) + \eta_{\mathbf{x}}\sum_{k=t_0}^{t_0+K_0}\tilde{\delta}_k\right\| \\
&\leq \frac{1}{\eta_{\mathbf{x}}(K_0+1)}(\|\tilde{\mathbf{x}}_{t_0+K_0+1} - \tilde{\mathbf{x}}_{t_0+K_0}\| + \theta_{\mathbf{x}}\|\tilde{\mathbf{x}}_{t_0+K_0} - \tilde{\mathbf{x}}_{t_0}\| + \eta_{\mathbf{x}}\sum_{k=t_0}^{t_0+K_0}\|\tilde{\delta}_k\|) \\
&\leq \frac{2}{\eta_{\mathbf{x}}K}\|\tilde{\mathbf{x}}_{t_0+K_0+1} - \tilde{\mathbf{x}}_{t_0+K_0}\| + \frac{2\theta_{\mathbf{x}}B}{\eta_{\mathbf{x}}K} + \frac{9\rho B^2}{4},
\end{aligned}
\tag{75}
$$

where we use $\mathbf{z}_{t_0} = \mathbf{x}_{t_0}$ in $\stackrel{a}{=}$. From $K_0 = \arg\min_{t_0+\lfloor\frac{K}{2}\rfloor\leq k\leq t_0+K-1}\|\mathbf{x}_{k+1} - \mathbf{x}_k\|$, we have

$$
\begin{aligned}
\|\mathbf{x}_{t_0+K_0+1} - \mathbf{x}_{t_0+K_0}\|^2 &\leq \frac{1}{K-\lfloor K/2\rfloor}\sum_{k=t_0+\lfloor K/2\rfloor}^{t_0+K-1}\|\mathbf{x}_{k+1} - \mathbf{x}_k\|^2 \\
&\leq \frac{1}{K-\lfloor K/2\rfloor}\sum_{k=t_0}^{t_0+K-1}\|\mathbf{x}_{k+1} - \mathbf{x}_k\|^2 \\
&\leq \frac{1}{K-\lfloor K/2\rfloor}\frac{B^2}{K} \leq \frac{2B^2}{K^2}
\end{aligned}
\tag{76}
$$

On the other hand, we also have

$$
\begin{aligned}
\|\nabla\Phi(\hat{\mathbf{z}})\| &= \|\tilde{\nabla}\Phi(\hat{\mathbf{z}})\| \\
&\leq \|\nabla g(\tilde{\mathbf{z}})\| + \|\hat{\tilde{\nabla}}\Phi(\hat{\mathbf{z}}) - \nabla g(\tilde{\mathbf{z}})\| \\
&= \|\nabla g(\tilde{\mathbf{z}})\| + \|\hat{\tilde{\nabla}}\Phi(\hat{\mathbf{z}}) - \tilde{\nabla}\Phi(\mathbf{x}_{t_0}) - \boldsymbol{\Lambda}(\tilde{\mathbf{z}} - \tilde{\mathbf{x}}_{t_0})\| \\
&\leq \|\nabla g(\tilde{\mathbf{z}})\| + \|\nabla\Phi(\hat{\mathbf{z}}) - \nabla\Phi(\mathbf{x}_{t_0}) - \mathbf{H}(\hat{\mathbf{z}} - \mathbf{x}_0)\| + \|\hat{\nabla}\Phi(\mathbf{z}_t) - \nabla\Phi(\mathbf{z}_t)\| \\
&\leq \|\nabla g(\tilde{\mathbf{z}})\| + \frac{\rho}{2}\|\hat{\mathbf{z}} - \mathbf{x}_{t_0}\|^2 + \|\hat{\nabla}\Phi(\mathbf{z}_t) - \nabla\Phi(\mathbf{z}_t)\| \\
&\leq \|\nabla g(\tilde{\mathbf{z}})\| + \frac{9\rho B^2}{4}
\end{aligned}
\tag{77}
$$

So we have

$$
\|\nabla\Phi(\hat{\mathbf{z}})\| \leq \frac{2\sqrt{2}B}{\eta_{\mathbf{x}}K^2} + \frac{2\theta_{\mathbf{x}}B}{\eta_{\mathbf{x}}K} + \frac{9\rho B^2}{2} \leq 82\epsilon
\tag{78}
$$

## D    PROOF OF SECTION 4.4.3

First, we set

$$r = \frac{\delta_0\epsilon}{64\mathcal{L}}\sqrt{\frac{\pi}{n}}, \quad \mathscr{T} = \frac{32\sqrt{\mathcal{L}}}{(\rho\epsilon)^{1/4}}\log\left(\frac{\mathcal{L}}{\delta_0}\sqrt{\frac{n}{\pi\rho\epsilon}}\right), \quad \delta_0 = \frac{\delta}{384\Delta_\Phi}\sqrt{\frac{\epsilon^3}{\rho}}. \tag{79}$$

Without loss of generality we assume $\hat{\mathbf{z}} = \mathbf{0}$ by shifting $\mathbb{R}^n$ such that $\hat{\mathbf{z}}$ is mapped to $\mathbf{0}$. Define a new $n$-dimensional function

$$h_\Phi(\mathbf{x}) = \Phi(\mathbf{x}) - \langle\nabla\Phi(\mathbf{0}), \mathbf{x}\rangle, \tag{80}$$

Since $\langle\nabla\Phi(\mathbf{0}), \mathbf{x}\rangle$ is a linear function with Hessian being 0, the Hessian of $h_\Phi$ equals to the Hessian of $\Phi$, and $h_\Phi(\mathbf{x})$ is also $(l_{\Phi,0}, l_{\Phi,1})$-smooth and $(\rho_{\Phi,0}, \rho_{\Phi,1})$-Hessian Lipschitz. In addition, note that $\nabla h_\Phi(\mathbf{0}) = \nabla\Phi(\mathbf{0}) - \nabla\Phi(\mathbf{0}) = 0$. Then for all $\mathbf{x} \in \mathbb{R}^n$ we have

$$\nabla h_\Phi(\mathbf{x}) = \int_{\xi=0}^1 \mathcal{H}(\xi\mathbf{x})\cdot\mathbf{x}\mathrm{d}\xi = \int_{\xi=0}^1 (\mathcal{H}(\xi\mathbf{x}) - \mathcal{H}(\mathbf{0}))\cdot\mathbf{x}\mathrm{d}\xi + \mathcal{H}(\mathbf{0})\mathbf{x} \tag{81}$$

Furthermore, due to the $(\rho_{\Phi,0}, \rho_{\Phi,1})$-Hessian Lipschitz condition of both $\Phi$ and $h_\Phi$, for any $\xi \in [0, G/\mathcal{L}]$ we have $\|\mathcal{H}(\xi\mathbf{x}) - \mathcal{H}(\mathbf{0})\| \le \rho\|\mathbf{x}\|$, where $\rho$ is the effective Hessian-smoothness constant, which leads to

$$\|\nabla h_\Phi(\mathbf{x}) - \mathcal{H}(\mathbf{0})\mathbf{x}\| \le \rho\|\mathbf{x}\|^2 \tag{82}$$

Use $\mathcal{H}(\hat{\mathbf{z}})$ to denote the Hessian matrix of $\Phi$ at $\hat{\mathbf{z}}$. Observe that $\mathcal{H}(\hat{\mathbf{z}})$ admits the following eigen-decomposition:

$$\mathcal{H}(\hat{\mathbf{z}}) = \sum_{i=1}^n \lambda_i\mathbf{u}_i\mathbf{u}_i^T, \tag{83}$$

where the set $\{\mathbf{u}_i\}_{i=1}^n$ forms an orthonormal basis of $\mathbb{R}^n$. Without loss of generality, we assume the eigenvalues $\lambda_1, \lambda_2, \ldots, \lambda_n$ corresponding to $\mathbf{u}_1, \mathbf{u}_2, \ldots, \mathbf{u}_n$ satisfy

$$\lambda_1 \le \lambda_2 \le \cdots \le \lambda_n \tag{84}$$

in which $\lambda_1 \le -\sqrt{\rho\epsilon}$. If $\lambda_n \le -\sqrt{\rho\epsilon}/2$, Lemma 4.3 holds directly, since no matter the value of $\hat{\mathbf{e}}$, we can have $\Phi(\mathbf{x}_{\mathscr{T}}) - \Phi(\hat{\mathbf{z}}) \le -\frac{1}{384}\sqrt{\frac{\epsilon^3}{\rho}}$. Hence, we only need to prove the case where $\lambda_n > -\sqrt{\rho\epsilon}$, in which there exists some $p$ with

$$\lambda_p \le -\sqrt{\rho\epsilon} < \lambda_{p+1} \tag{85}$$

We use $\mathfrak{S}_\parallel$ to denote the subspace of $\mathbb{R}^n$ spanned by $\{\mathbf{u}_1, \mathbf{u}_2, \ldots, \mathbf{u}_p\}$, and use $\mathfrak{S}_\perp$ to denote the subspace spanned by $\{\mathbf{u}_{p+1}, \mathbf{u}_{p+2}, \ldots, \mathbf{u}_n\}$. Then we can have the following lemma:

**Lemma D.1** *Running Algorithm 1 with parameters setting in Theorem 4.3. Denote $t_0$ to be the iteration number after the condition on Line 11 triggers. Define $\alpha'_t$ to be*

$$\alpha'_t = \frac{\|\mathbf{x}_{t,\parallel}\|}{\|\mathbf{x}_t\|}, \tag{86}$$

*in which $\mathbf{x}_{t,\parallel}$ is the component of $\mathbf{x}_t$ in the subspace $\mathfrak{S}_\parallel$. Define $\mathbf{v}_{t+1} := \mathbf{x}_{t+1} - \mathbf{x}_t$ for each iteration. Then, during all the $\mathscr{T}$ iterations after Line 11 triggers, we have $\alpha'_t \ge \alpha'_{\min}$ for*

$$\alpha'_{\min} = \frac{\delta_0}{8}\sqrt{\frac{\pi}{n}} \tag{87}$$

*given that $\alpha'_0 \ge \sqrt{\frac{\pi}{n}}\delta_0$.*

*Proof.* Without loss of generality, assume $\hat{\mathbf{z}} = \mathbf{0}$ and $\nabla\Phi(\hat{\mathbf{z}}) = \mathbf{0}$. If not, ... We consider the worst case, where the initial value $\alpha'_0 = \sqrt{\frac{\pi}{n}}\delta_0$ and the component $x_{0,1}$ along $\mathbf{u}_1$ equals 0. Also, the eigenvalues satisfy

$$\lambda_2 = \lambda_3 = \cdots = \lambda_p = -\sqrt{\rho\epsilon}, \quad \lambda_{p+1} = \lambda_{p+2} = \cdots = \lambda_{n-1} = -\sqrt{\rho\epsilon} + \nu, \tag{88}$$

for an arbitrarily small positive constant $\nu$, which can make components of $\mathbf{x}_t$ in $\mathfrak{S}_\perp$ as large as possible to make $\alpha'_t$ smaller. Out of the same reason, we assume that each time we make a gradient call at point $\mathbf{z}_t$, the derivation term $\Delta$ from pure quadratic approximation

$$
\begin{aligned}
\Delta &= \frac{\|\mathbf{z}_t\|}{r}\left(\widehat{\nabla}\Phi(r\frac{\mathbf{z}_t}{\|\mathbf{z}_t\|}) - \mathcal{H}(\mathbf{0})\cdot r\frac{\mathbf{z}_t}{\|\mathbf{z}_t\|}\right) \\
&= \frac{\|\mathbf{z}_t\|}{r}\cdot\left(\nabla\Phi(r\frac{\mathbf{z}_t}{\|\mathbf{z}_t\|}) - \mathcal{H}(\mathbf{0})\cdot r\frac{\mathbf{z}_t}{\|\mathbf{z}_t\|} + \widehat{\nabla}\Phi(r\frac{\mathbf{z}_t}{\|\mathbf{z}_t\|}) - \nabla\Phi(r\frac{\mathbf{z}_t}{\|\mathbf{z}_t\|})\right)
\end{aligned}
\tag{89}
$$

lies in the direction that can make $\alpha'_t$ as small as possible. Then, the component $\Delta_\parallel$ in $\mathfrak{S}_\parallel$ should be in the opposite direction to $\mathbf{z}_\parallel$, and the component $\Delta_\perp$ in $\mathfrak{S}_\perp$ should be in the direction of $\mathbf{z}_\perp$. Hence in this case, we have both $\|\mathbf{x}_{t,\perp}\|/\|\mathbf{x}_t\|$ and $\|\mathbf{z}_{t,\perp}\|/\|\mathbf{z}_t\|$ being non-decreasing, since $\nu$ can be arbitrarily small. Also, it admits the following recurrence formula:

$$
\begin{aligned}
\|\mathbf{x}_{t+2,\perp}\| &\le (1+\eta_\mathbf{x}(\sqrt{\rho\epsilon}-\nu))\left(\|\mathbf{x}_{t+1,\perp}\| + (1-\theta_\mathbf{x})\left(\|\mathbf{x}_{t+1,\perp}\|-\|\mathbf{x}_{t,\perp}\|\right)\right) + \eta_\mathbf{x}\|\Delta_\perp\| \\
&\le (1+\eta_\mathbf{x}\sqrt{\rho\epsilon})\left(\|\mathbf{x}_{t+1,\perp}\| + (1-\theta_\mathbf{x})\left(\|\mathbf{x}_{t+1,\perp}\|-\|\mathbf{x}_{t,\perp}\|\right)\right) + \eta_\mathbf{x}\|\Delta_\perp\|,
\end{aligned}
\tag{90}
$$

where the second inequality is due to the fact that $\nu$ can be an arbitrarily small positive number. Note that since $\|\mathbf{x}_{t,\perp}\|/\|\mathbf{x}_t\|$ is non-decreasing in this worst-case scenario, we have

$$
\frac{\|\Delta_\perp\|}{\|\mathbf{x}_{t+1,\perp}\|} \le \frac{\|\Delta\|}{\|\mathbf{x}_{t+1}\|}\cdot\frac{\|\mathbf{x}_{t_0}\|}{\|\mathbf{x}_{t_0,\perp}\|} \le \frac{2\|\Delta\|}{\|\mathbf{x}_{t+1}\|} \le 2\rho r + 2\|\delta_{\widehat{\Phi}}\| \le 4\rho r
\tag{91}
$$

which leads to

$$
\|\mathbf{x}_{t+2,\perp}\| \le (1+\eta\sqrt{\rho\epsilon}+4\eta\rho r)\left((2-\theta_\mathbf{x})\|\mathbf{x}_{t+1,\perp}\| - (1-\theta_\mathbf{x})\|\mathbf{x}_{t,\perp}\|\right).
\tag{92}
$$

On the other hand, suppose for some value $t$, we have $\alpha'_k \ge \alpha'_{\min}$ with any $t_0+1 \le k \le t+1$. Then,

$$
\begin{aligned}
\|\mathbf{x}_{t+2,\parallel}\| &\ge (1+\eta_\mathbf{x}(\sqrt{\rho\epsilon}-\nu))\left(\|\mathbf{x}_{t+1,\parallel}\| + (1-\theta_\mathbf{x})\left(\|\mathbf{x}_{t+1,\parallel}\|-\|\mathbf{x}_{t,\parallel}\|\right)\right) + \eta_\mathbf{x}\|\Delta_\parallel\| \\
&\ge (1+\eta_\mathbf{x}\sqrt{\rho\epsilon})\left(\|\mathbf{x}_{t+1,\parallel}\| + (1-\theta_\mathbf{x})\left(\|\mathbf{x}_{t+1,\parallel}\|-\|\mathbf{x}_{t,\parallel}\|\right)\right) - \eta_\mathbf{x}\|\Delta\|.
\end{aligned}
\tag{93}
$$

Note that since $\|\mathbf{x}_{t+1,\parallel}\|/\|\mathbf{x}_t\| \ge \alpha'_{\min}$, we have

$$
\frac{\|\Delta\|}{\|\mathbf{z}_{t+1,\parallel}\|} \le \frac{\|\Delta\|}{\alpha'_{\min}\|\mathbf{z}_{t+1}\|} \le \frac{\rho r + \|\delta_{\widehat{\Phi}}\|}{\alpha'_{\min}} \le \frac{2\rho r}{\alpha'_{\min}}
\tag{94}
$$

which leads to

$$
\|\mathbf{x}_{t+2,\parallel}\| \ge (1+\eta\sqrt{\rho\epsilon}-2\eta\rho r/\alpha'_{\min})\left((2-\theta_\mathbf{x})\|\mathbf{x}_{t+1,\parallel}\| - (1-\theta_\mathbf{x})\|\mathbf{x}_{t,\parallel}\|\right)
\tag{95}
$$

Consider the sequences with recurrence that can be written as

$$
\xi_{t+2} = (1+p)\left((2-\theta_\mathbf{x})\xi_{t+1} - (1-\theta_\mathbf{x})\xi_t\right)
\tag{96}
$$

for some $p > 0$. Its characteristic equation can be written as

$$
x^2 - (1+p)(2-\theta_\mathbf{x})x + (1+p)(1-\theta_\mathbf{x}) = 0,
\tag{97}
$$

whose roots satisfy

$$
x = \frac{1+p}{2}\left((2-\theta_\mathbf{x}) \pm \sqrt{(2-\theta_\mathbf{x})^2 - \frac{4(1-\theta_\mathbf{x})}{1+p}}\right),
\tag{98}
$$

indicating

$$
\xi_t = \left(\frac{1+p}{2}\right)^t\left(C_1(2-\theta_\mathbf{x}+q)^t + C_2(2-\theta_\mathbf{x}-q)^t\right),
\tag{99}
$$

where $q := \sqrt{(2-\theta_\mathbf{x})^2 - \frac{4(1-\theta_\mathbf{x})}{1+p}}$, for constants $C_1$ and $C_2$ being

$$
\begin{cases}
C_1 = -\frac{2-\theta_\mathbf{x}-q}{2q}\xi_0 + \frac{1}{(1+p)q}\xi_1 \\
C_2 = \frac{2-\theta_\mathbf{x}+q}{2q}\xi_0 - \frac{1}{(1+p)q}\xi_1
\end{cases}
\tag{100}
$$

Then by the inequalities 92 and 95, as long as $\alpha'_k \geq \alpha'_{\min}$ for any $t_0 + 1 \leq k \leq t - 1$, the values $\|\mathbf{x}_{t,\perp}\|$ and $\|\mathbf{x}_{t,\|}\|$ satisfy

$$
\begin{aligned}
\|\mathbf{x}_{t,\perp}\| \leq &\left(-\frac{2 - \theta_{\mathbf{x}} - \mu_{\perp}}{2\mu_{\perp}} \xi_{0,\perp} + \frac{1}{(1 + \kappa_{\perp})\mu_{\perp}} \xi_{1,\perp}\right) \cdot \left(\frac{1 + \kappa_{\perp}}{2}\right)^t \cdot (2 - \theta_{\mathbf{x}} + \mu_{\perp})^t \\
&+ \left(\frac{2 - \theta_{\mathbf{x}} + \mu_{\perp}}{2\mu_{\perp}} \xi_{0,\perp} - \frac{1}{(1 + \kappa_{\perp})\mu_{\perp}} \xi_{1,\perp}\right) \cdot \left(\frac{1 + \kappa_{\perp}}{2}\right)^t \cdot (2 - \theta_{\mathbf{x}} - \mu_{\perp})^t,
\end{aligned}
\tag{101}
$$

and

$$
\begin{aligned}
\|\mathbf{x}_{t,\|}\| \geq &\left(-\frac{2 - \theta_{\mathbf{x}} - \mu_{\|}}{2\mu_{\|}} \xi_{0,\|} + \frac{1}{(1 + \kappa_{\|})\mu_{\|}} \xi_{1,\|}\right) \cdot \left(\frac{1 + \kappa_{\|}}{2}\right)^t \cdot (2 - \theta_{\mathbf{x}} + \mu_{\|})^t \\
&+ \left(\frac{2 - \theta_{\mathbf{x}} + \mu_{\|}}{2\mu_{\|}} \xi_{0,\|} - \frac{1}{(1 + \kappa_{\|})\mu_{\|}} \xi_{1,\|}\right) \cdot \left(\frac{1 + \kappa_{\|}}{2}\right)^t \cdot (2 - \theta_{\mathbf{x}} - \mu_{\|})^t,
\end{aligned}
\tag{102}
$$

where

$$
\begin{aligned}
\kappa_{\perp} = \eta\sqrt{\rho\epsilon} + 4\eta\rho r, \quad \xi_{0,\perp} = \|\mathbf{x}_{t_0,\perp}\|, \quad \xi_{1,\perp} = (1 + \kappa_{\perp})\xi_{0,\perp} \\
\kappa_{\|} = \eta\sqrt{\rho\epsilon} - 2\eta\rho r/\alpha'_{\min}, \quad \xi_{0,\|} = \|\mathbf{x}_{t_0,\|}\|, \quad \xi_{1,\|} = (1 + \kappa_{\|})\xi_{0,\|}
\end{aligned}
\tag{103}
$$

Further we can derive that

$$
\|\mathbf{x}_{t,\perp}\| \leq \|\mathbf{x}_{t_0,\perp}\| \cdot \left(\frac{1 + \kappa_{\perp}}{2}\right)^t \cdot (2 - \theta_{\mathbf{x}} + \mu_{\perp})^t
\tag{104}
$$

and

$$
\|\mathbf{x}_{t,\|}\| \geq \frac{\|\mathbf{x}_{0,\|}\|}{2} \cdot \left(\frac{1 + \kappa_{\|}}{2}\right)^t \cdot (2 - \theta_{\mathbf{x}} + \mu_{\|})^t.
\tag{105}
$$

Then we can observe that

$$
\frac{\|\mathbf{x}_{t,\|}\|}{\|\mathbf{x}_{t,\perp}\|} \geq \frac{\|\mathbf{x}_{t_0,\|}\|}{2\|\mathbf{x}_{t_0,\perp}\|} \cdot \left(\frac{1 + \kappa_{\|}}{1 + \kappa_{\perp}}\right)^t \cdot \left(\frac{2 - \theta_{\mathbf{x}} + \mu_{\|}}{2 - \theta_{\mathbf{x}} + \mu_{\perp}}\right)^t,
\tag{106}
$$

where

$$
\begin{aligned}
\frac{1 + \kappa_{\|}}{1 + \kappa_{\perp}} &\geq (1 + \kappa_{\|})(1 - \kappa_{\perp}) \\
&\geq 1 - (4 + 2/\alpha'_{\min})\eta\rho r - \kappa_{\|}\kappa_{\perp} \\
&\geq 1 - 4\eta\rho r/\alpha'_{\min},
\end{aligned}
\tag{107}
$$

and

$$
\begin{aligned}
\frac{2 - \theta_{\mathbf{x}} + \mu_{\|}}{2 - \theta_{\mathbf{x}} + \mu_{\perp}} &= \frac{1 + \mu_{\|}/(2 - \theta_{\mathbf{x}})}{1 + \mu_{\perp}/(2 - \theta_{\mathbf{x}})} \\
&= \frac{1 + \sqrt{1 - \frac{4(1 - \theta_{\mathbf{x}})}{(1 + \kappa_{\|})(2 - \theta_{\mathbf{x}})^2}}}{1 + \sqrt{1 - \frac{4(1 - \theta_{\mathbf{x}})}{(1 + \kappa_{\perp})(2 - \theta_{\mathbf{x}})^2}}} \\
&\geq \left(1 + \frac{1}{2 - \theta_{\mathbf{x}}}\sqrt{\frac{\theta_{\mathbf{x}}^2 + \kappa_{\|}(2 - \theta_{\mathbf{x}})^2}{1 + \kappa_{\|}}}\right)\left(1 - \frac{1}{2 - \theta_{\mathbf{x}}}\sqrt{\frac{\theta_{\mathbf{x}}^2 + \kappa_{\perp}(2 - \theta_{\mathbf{x}})^2}{1 + \kappa_{\perp}}}\right) \\
&\geq 1 - \frac{2(\kappa_{\perp} - \kappa_{\|})}{\theta_{\mathbf{x}}} \geq 1 - \frac{6\eta\rho r}{\alpha'_{\min}\theta_{\mathbf{x}}}
\end{aligned}
\tag{108}
$$

by which we can derive that

$$
\begin{aligned}
\frac{\|\mathbf{x}_{t,\|\|}\|}{\|\mathbf{x}_{t,\perp}\|} &\geq \frac{\|\mathbf{x}_{t_0,\|}\|}{2\|\mathbf{x}_{t_0,\perp}\|} \cdot \left(1 - \frac{8\rho r}{\alpha'_{\min}\theta_{\mathbf{x}}}\right)^t \\
&\geq \frac{\|\mathbf{x}_{t_0,\|}\|}{2\|\mathbf{x}_{t_0,\perp}\|} \cdot (1 - 1/\mathscr{T})^t \\
&\geq \frac{\|\mathbf{x}_{t_0,\|}\|}{2\|\mathbf{x}_{t_0,\perp}\|} \cdot \exp\left(-\frac{t}{\mathscr{T} - 1}\right) \geq \frac{\|\mathbf{x}_{t_0,\|}\|}{4\|\mathbf{x}_{t_0,\perp}\|},
\end{aligned}
\tag{109}
$$

indicating

$$\alpha'_t = \frac{\|\mathbf{x}_{t,\|}\|}{\sqrt{\|\mathbf{x}_{t,\|}\|^2 + \|\mathbf{x}_{t,\perp}\|^2}} \geq \frac{\|\mathbf{x}_{t_0,\|}\|}{8\|\mathbf{x}_{t_0,\perp}\|} \geq \alpha'_{\min} \tag{110}$$

Hence, as long as $\alpha'_k \geq \alpha'_{\min}$ for any $t_0 + 1 \leq k \leq t - 1$, we can also have $\alpha'_t \geq \alpha'_{\min}$ if $t \leq \mathcal{T}$. Since we have $\alpha'_0 \geq \alpha'_{\min}$ and $\alpha'_1 \geq \alpha'_{\min}$, we can claim that $\alpha'_t \geq \alpha'_{\min}$ for any $t \leq \mathcal{T}$ using recurrence.

## D.1  PROOF OF LEMMA 4.7

**Lemma D.2** *Running Algorithm 1 with parameters setting in Theorem 4.3. Denote $t_0$ to be the iteration number after the condition on Line 11 triggers. Define $\mathbf{v}_{t+1} = \mathbf{x}_{t+1} - \mathbf{x}_t$ for each iteration. For the point $\hat{\mathbf{z}}$ satisfying $\lambda_{\min}\left(\nabla^2 \Phi(\hat{\mathbf{z}})\right) \leq -\sqrt{\rho\epsilon}$, adding an uniform perturbation in Line 16, the unit vector $\hat{\mathbf{e}}$ in Line 21 obtained after $\mathcal{T}$ iterations satisfies*

$$\mathbb{P}\left(\hat{\mathbf{e}}^T \mathcal{H}(\mathbf{x})\hat{\mathbf{e}} \leq -\sqrt{\rho\epsilon}/4\right) \geq 1 - \delta_0 \tag{111}$$

*Proof.* If $\lambda_n \leq -\sqrt{\rho\epsilon}/2$, Lemma D.2 holds directly. Hence, we only need to consider the case where $\lambda_n > -\sqrt{\rho\epsilon}/2$, in which there exists some $p'$ with

$$\lambda'_p \leq -\sqrt{\rho\epsilon}/2 < \lambda_{p+1} \tag{112}$$

We use $\mathfrak{S}'_{\|}, \mathfrak{S}'_{\perp}$ to denote the subspace of $\mathbb{R}^n$ spanned by

$$\{\mathbf{u}_1, \mathbf{u}_2, \ldots, \mathbf{u}_{p'}\}, \quad \{\mathbf{u}_{p'+1}, \mathbf{u}_{p+2}, \ldots, \mathbf{u}_n\}$$

Furthermore, we define

$$\mathbf{x}_{t,\|'} := \sum_{i=1}^{p'} \langle \mathbf{u}_i, \mathbf{x}_t \rangle \mathbf{u}_i, \quad \mathbf{x}_{t,\perp'} := \sum_{i=p'}^{n} \langle \mathbf{u}_i, \mathbf{x}_t \rangle \mathbf{u}_i,$$

$$\mathbf{v}_{t,\|'} := \sum_{i=1}^{p'} \langle \mathbf{u}_i, \mathbf{v}_t \rangle \mathbf{u}_i, \quad \mathbf{v}_{t,\perp'} := \sum_{i=p'}^{n} \langle \mathbf{u}_i, \mathbf{v}_t \rangle \mathbf{u}_i$$

respectively to denote the component of $\mathbf{x}'_t$ and $\mathbf{v}'_t$ in the subspaces $\mathfrak{S}'_{\|}$, $\mathfrak{S}'_{\perp}$, and let $\alpha'_t := \|\mathbf{x}_{t,\|}\|/\|\mathbf{x}_t\|$. Consider the case where $\alpha'_0 \geq \sqrt{\frac{\pi}{n}}\delta_0$, which can be achieved with probability

$$\Pr\left\{\alpha'_0 \geq \sqrt{\frac{\pi}{n}}\delta_0\right\} \geq 1 - \sqrt{\frac{\pi}{n}}\delta_0 \cdot \frac{\text{Vol}\left(\mathbb{B}_0^{n-1}(1)\right)}{\text{Vol}\left(\mathbb{B}_0^n(1)\right)} \geq 1 - \sqrt{\frac{\pi}{n}}\delta_0 \cdot \sqrt{\frac{n}{\pi}} = 1 - \delta_0 \tag{113}$$

we prove that there exists some $t'$ with $t_0 + 1 \leq t' \leq \mathcal{T}$ such that

$$\frac{\|\mathbf{x}_{t',\perp'}\|}{\|\mathbf{x}_{t'}\|} \leq \frac{\sqrt{\rho\epsilon}}{8\mathcal{L}} \tag{114}$$

Assume the contrary, for any $t$ with $1 \leq t \leq K'$, we all have $\frac{\|\mathbf{x}_{t,\perp'}\|}{\|\mathbf{x}_t\|} > \frac{\sqrt{\rho\epsilon}}{8\mathcal{L}}$ and $\frac{\|\mathbf{z}_{t,\perp'}\|}{\|\mathbf{z}_t\|} > \frac{\sqrt{\rho\epsilon}}{8\mathcal{L}}$. Focus on the case where $\|\mathbf{x}_{t,\perp'}\|$, the component of $\mathbf{x}_t$ in subspace $\mathfrak{S}'_{\perp}$, achieves the largest value possible. Then in this case, we have the following formula:

$$\|\mathbf{x}_{t+2,\perp'}\| \leq (1 + \eta_{\mathbf{x}}\sqrt{\rho\epsilon}/2)\left(\|\mathbf{x}_{t+1,\perp'}\| + (1 - \theta_{\mathbf{x}})\left(\|\mathbf{x}_{t+1,\perp'}\| - \|\mathbf{x}_{t,\perp'}\|\right)\right) + \eta_{\mathbf{x}}\|\Delta_{\perp'}\|. \tag{115}$$

Since $\frac{\|\mathbf{z}_{k,\perp'}\|}{\|\mathbf{z}_k\|} \geq \frac{\sqrt{\rho\epsilon}}{8\mathcal{L}}$ for any $t_0 + 1 \leq k \leq t + 1$, we can derive that

$$\frac{\|\Delta_{\perp}\|}{\|\mathbf{x}_{t+1,\perp}\| + (1 - \theta_{\mathbf{x}})\left(\|\mathbf{x}_{t+1,\perp}\| - \|\mathbf{x}_{t,\perp}\|\right)} \leq \frac{\|\Delta\|}{\|\mathbf{z}_{t,\perp'}\|} \leq \frac{2\rho r + 2\|\delta_{\widehat{\Phi}}\|}{\sqrt{\rho\epsilon}} \tag{116}$$

which leads to

$$\begin{aligned}\|\mathbf{x}_{t+2,\perp'}\| &\leq (1 + \eta_{\mathbf{x}}\sqrt{\rho\epsilon}/2)\left(\|\mathbf{x}_{t+1,\perp'}\| + (1 - \theta_{\mathbf{x}})\left(\|\mathbf{x}_{t+1,\perp'}\| - \|\mathbf{x}_{t,\perp'}\|\right)\right) + \eta_{\mathbf{x}}\|\Delta_{\perp'}\| \\ &\leq (1 + \eta_{\mathbf{x}}\sqrt{\rho\epsilon}/2 + 4\rho r/\sqrt{\rho\epsilon})\left((2 - \theta_{\mathbf{x}})\|\mathbf{x}_{t+1,\perp'}\| - (1 - \theta_{\mathbf{x}})\|\mathbf{x}_{t,\perp'}\|\right)\end{aligned} \tag{117}$$

Similar to the proof of Lemma D.1, it can be further derived that

$$\|\mathbf{x}_{t,\perp'}\| \leq \|\mathbf{x}_{t_0,\perp'}\| \cdot \left(\frac{1+\kappa_{\perp'}}{2}\right)^t \cdot (2 - \theta_{\mathbf{x}} + \mu_{\perp'})^t \tag{118}$$

for $\kappa_{\perp'} = \eta_{\mathbf{x}}\sqrt{\rho\epsilon}/2 + 4\rho r/\sqrt{\rho\epsilon}$ and $\mu_{\perp'} = \sqrt{(2-\theta_{\mathbf{x}})^2 - \frac{4(1-\theta_{\mathbf{x}})}{1+\kappa_{\perp'}}}$, given $\frac{\|\mathbf{x}_{k,\perp'}\|}{\|\mathbf{x}_k\|} \geq \frac{\sqrt{\rho\epsilon}}{8\mathcal{L}}$ and $\frac{\|\mathbf{z}_{k,\perp'}\|}{\|\mathbf{z}_k\|} \geq \frac{\sqrt{\rho\epsilon}}{8\mathcal{L}}$ for any $t_0 + 1 \leq k \leq t-1$. By Lemma D.1,

$$\alpha'_t \geq \alpha'_{\min} = \frac{\delta_0}{8}\sqrt{\frac{\pi}{n}}, \quad \forall t_0 + 1 \leq t \leq \mathscr{T} \tag{119}$$

and it is demonstrated in the proof of Lemma D.1 that,

$$\|\mathbf{x}_{t,\|}\| \geq \frac{\|\mathbf{x}_{t_0,\|}\|}{2} \cdot \left(\frac{1+\kappa_\|}{2}\right)^t \cdot \left(2 - \theta_{\mathbf{x}} + \mu_\|\right)^t, \quad \forall t_0 + 1 \leq t \leq \mathscr{T}, \tag{120}$$

for $\kappa_\| = \eta_{\mathbf{x}}\sqrt{\rho\epsilon} - 2\eta_{\mathbf{x}}\rho r/\alpha'_{\min}$ and $\mu_\| = \sqrt{(2-\theta_{\mathbf{x}})^2 - \frac{4(1-\theta_{\mathbf{x}})}{1+\kappa_\|}}$. Observe that

$$\begin{aligned}
\frac{\|\mathbf{x}_{\mathscr{T},\perp'}\|}{\|\mathbf{x}_{\mathscr{T},\|}\|} &\leq \frac{2\|\mathbf{x}_{t_0,\perp'}\|}{\|\mathbf{x}_{t_0,\|}\|} \cdot \left(\frac{1+\kappa_{\perp'}}{1+\kappa_\|}\right)^{\mathscr{T}} \cdot \left(\frac{2-\theta_{\mathbf{x}}+\mu_{\perp'}}{2-\theta_{\mathbf{x}}+\mu_\|}\right)^{\mathscr{T}} \\
&\leq \frac{2}{\delta_0}\sqrt{\frac{n}{\pi}} \left(\frac{1+\kappa_{\perp'}}{1+\kappa_\|}\right)^{\mathscr{T}} \cdot \left(\frac{2-\theta_{\mathbf{x}}+\mu_{\perp'}}{2-\theta_{\mathbf{x}}+\mu_\|}\right)^{\mathscr{T}}
\end{aligned} \tag{121}$$

where

$$\frac{1+\kappa_{\perp'}}{1+\kappa_\|} \leq \frac{1}{1+(\kappa_\| - \kappa_{\perp'})} = 1 - \frac{1}{\eta_{\mathbf{x}}\sqrt{\rho\epsilon}/2 + 2\rho r\left(\eta_{\mathbf{x}}/\alpha_{\min'} + 2/\sqrt{\rho\epsilon}\right)} \leq 1 - \frac{\eta_{\mathbf{x}}\sqrt{\rho\epsilon}}{4} \tag{122}$$

and

$$\begin{aligned}
\frac{2-\theta_{\mathbf{x}}+\mu_{\perp'}}{2-\theta_{\mathbf{x}}+\mu_\|} &= \frac{1+\sqrt{1-\frac{4(1-\theta_{\mathbf{x}})}{(1+\kappa_{\perp'})(2-\theta_{\mathbf{x}})^2}}}{1+\sqrt{1-\frac{4(1-\theta_{\mathbf{x}})}{(1+\kappa_\|)(2-\theta_{\mathbf{x}})^2}}} \\
&\leq \frac{1}{1+\left(\sqrt{1-\frac{4(1-\theta_{\mathbf{x}})}{(1+\kappa_{\perp'})(2-\theta_{\mathbf{x}})^2}} - \sqrt{1-\frac{4(1-\theta_{\mathbf{x}})}{(1+\kappa_\|)(2-\theta_{\mathbf{x}})^2}}\right)} \\
&\leq 1 - \frac{\kappa_\| - \kappa_{\perp'}}{\theta_{\mathbf{x}}} \\
&\leq 1 - \frac{\eta\sqrt{\rho\epsilon}}{4\theta_{\mathbf{x}}} = 1 - \frac{(\rho\epsilon)^{1/4}}{8\sqrt{\mathcal{L}}}.
\end{aligned} \tag{123}$$

Hence,

$$\frac{\|\mathbf{x}_{\mathscr{T},\perp'}\|}{\|\mathbf{x}_{\mathscr{T},\|}\|} \leq \frac{2}{\delta_0}\sqrt{\frac{n}{\pi}}\left(1 - \frac{(\rho\epsilon)^{1/4}}{8\sqrt{\mathcal{L}}}\right)^{\mathscr{T}} \leq \frac{\sqrt{\rho\epsilon}}{8\mathcal{L}} \tag{124}$$

Since $\|\mathbf{x}_{\mathscr{T},\|}\| \leq \|\mathbf{x}_{\mathscr{T}}\|$, we have $\frac{\|\mathbf{x}_{\mathscr{T},\perp'}\|}{\|\mathbf{x}_{\mathscr{T}}\|} \leq \frac{\sqrt{\rho\epsilon}}{8\mathcal{L}}$, contradiction. Hence, there here exists some $t_0$ with $t_0 + 1 \leq t' \leq \mathscr{T}$ such that $\frac{\|\mathbf{x}_{t',\perp'}\|}{\|\mathbf{x}_{t'}\|} \leq \frac{\sqrt{\rho\epsilon}}{8\mathcal{L}}$. Consider the normalized vector $\hat{\mathbf{e}} = \mathbf{x}_{t'}/r$, we use $\hat{\mathbf{e}}_{\perp'}$ and $\hat{\mathbf{e}}_{\|'}$ to separately denote the component of $\hat{\mathbf{e}}$ in $\mathfrak{S}'_\perp$ and $\mathfrak{S}'_\|$. Then, $\|\hat{\mathbf{e}}_{\perp'}\| \leq \sqrt{\rho\epsilon}/(8\mathcal{L})$ whereas $\|\hat{\mathbf{e}}_{\|'}\| \geq 1 - \rho\epsilon/(8\mathcal{L})^2$. Then,

$$\hat{\mathbf{e}}^T\mathcal{H}(\mathbf{0})\hat{\mathbf{e}} = \left(\hat{\mathbf{e}}_{\perp'} + \hat{\mathbf{e}}_{\|'}\right)^T \mathcal{H}(\mathbf{0})\left(\hat{\mathbf{e}}_{\perp'} + \hat{\mathbf{e}}_{\|'}\right), \tag{125}$$

since $\mathcal{H}(\mathbf{0})\hat{\mathbf{e}}_{\perp'} \in \mathfrak{S}'_\perp$ and $\mathcal{H}(\mathbf{0})\hat{\mathbf{e}}_{\|'} \in \mathfrak{S}'_\|$, it can be further simplified to

$$\hat{\mathbf{e}}^T\mathcal{H}(\mathbf{0})\hat{\mathbf{e}} = \hat{\mathbf{e}}_{\perp'}^T\mathcal{H}(\mathbf{0})\hat{\mathbf{e}}_{\perp'} + \hat{\mathbf{e}}_{\|'}^T\mathcal{H}(\mathbf{0})\hat{\mathbf{e}}_{\|'} \tag{126}$$

Due to the $\mathcal{L}$-smoothness of the function, all eigenvalue of the Hessian matrix has its absolute value upper bounded by $\mathcal{L}$. Hence,

$$\hat{\mathbf{e}}_{\perp'}^T \mathcal{H}(\mathbf{0})\hat{\mathbf{e}}_{\perp'} \leq \mathcal{L}\|\hat{\mathbf{e}}_{\perp'}^T\|^2 = \frac{\rho\epsilon}{64\mathcal{L}^2}. \tag{127}$$

Further according to the definition of $\mathfrak{S}_\parallel$, we have

$$\hat{\mathbf{e}}_{\parallel'}^T \mathcal{H}(\mathbf{0})\hat{\mathbf{e}}_{\parallel'} \leq -\frac{\sqrt{\rho\epsilon}}{2}\|\hat{\mathbf{e}}_{\parallel'}\|^2 \tag{128}$$

Combining these two inequalities together, we can obtain

$$\hat{\mathbf{e}}^T \mathcal{H}(\mathbf{0})\hat{\mathbf{e}} = \hat{\mathbf{e}}_{\perp}^T \mathcal{H}(\mathbf{0})\hat{\mathbf{e}}_{\perp'} + \hat{\mathbf{e}}_{\parallel'}^T \mathcal{H}(\mathbf{0})\hat{\mathbf{e}}_{\parallel'} \leq -\frac{\sqrt{\rho\epsilon}}{2}\left\|\hat{\mathbf{e}}_{\parallel'}\right\|^2 + \frac{\rho\epsilon}{64\mathcal{L}^2} \leq -\frac{\sqrt{\rho\epsilon}}{4}, \tag{129}$$

which finish the proof.

### D.2 Proof of Lemma 4.8

**Lemma D.3** *Running Algorithm 1 with parameters setting in Theorem 4.3. For each $\hat{\mathbf{z}}$ if there exists a unit vector $\hat{\mathbf{e}}$ satisfying $\hat{\mathbf{e}}^T \mathcal{H}(\hat{\mathbf{z}})\hat{\mathbf{e}} \leq -\frac{\sqrt{\rho\epsilon}}{4}$ where $\mathcal{H}$ stands for the Hessian matrix of function $\Phi$, the following inequality holds*

$$\Phi\left(\hat{\mathbf{z}} - \frac{1}{4}\sqrt{\frac{\epsilon}{\rho}} \cdot \hat{\mathbf{e}}\right) \leq \Phi(\hat{\mathbf{z}}) - \frac{1}{384}\sqrt{\frac{\epsilon^3}{\rho}}, \tag{130}$$

*where $\Phi'_{\hat{\mathbf{e}}}$ stands for the gradient component of $\Phi$ along the direction of $\hat{\mathbf{e}}$.*

*Proof.* Without loss of generality, we assume $\hat{\mathbf{z}} = \mathbf{0}$. We can also assume $\langle\nabla\Phi(\mathbf{0}), \hat{\mathbf{e}}\rangle \leq 0$; if this is not the case we can pick $-\hat{\mathbf{e}}$ instead, which still satisfies $(-\hat{\mathbf{e}})^T \mathcal{H}(\hat{\mathbf{z}})(-\hat{\mathbf{e}}) \leq -\frac{\sqrt{\rho\epsilon}}{4}$. Then, for any $\mathbf{x} = e\hat{\mathbf{e}}$ with some $e > 0$, we have $\frac{\partial^2\Phi}{\partial(e\hat{\mathbf{e}})^2}(\mathbf{x}) \leq -\frac{\sqrt{\rho\epsilon}}{4} + \rho e$ due to the $\rho$-Hessian Lipschitz condition of $\Phi$. Hence,

$$\frac{\partial\Phi}{\partial e\hat{\mathbf{e}}}(\mathbf{x}) \leq \Phi'_{\hat{\mathbf{e}}}(\mathbf{0}) - \frac{\sqrt{\rho\epsilon}}{4}e + \rho e^2 \tag{131}$$

by which we can further derive that

$$\Phi(e\hat{\mathbf{e}}) - \Phi(\mathbf{0}) \leq \Phi'_{\hat{\mathbf{e}}}(\mathbf{0})e - \frac{\sqrt{\rho\epsilon}}{8}e^2 + \frac{\rho}{3}e^3 \leq -\frac{\sqrt{\rho\epsilon}}{8}e^2 + \frac{\rho}{3}e^3. \tag{132}$$

Settings $e = \frac{1}{4}\sqrt{\frac{\epsilon}{\rho}}$ finishes the proof.

## E Proof of Theorem 4.3

*Proof.* Denote $\mathscr{F} = \frac{51}{64}\sqrt{\frac{\epsilon^3}{\rho}}$. Set the total step number $T$ to be

$$T = \max\left\{\frac{308\Delta_\Phi\left(K + \mathscr{T}\right)}{\mathscr{F}}, 768\Delta_\Phi\mathscr{T}\sqrt{\frac{\rho}{\epsilon^3}}\right\} = \mathcal{O}\left(\frac{\Delta_\Phi}{\epsilon^{1.75}} \cdot \log n\right) \tag{133}$$

We first assert that for each iteration $\mathbf{x}_{t+1}$ that a uniform perturbation is added, after $\mathscr{T}$ iterations we can successfully obtain a unit vector $\hat{\mathbf{e}}$ with $\hat{\mathbf{e}}^T \mathcal{H}\hat{\mathbf{e}} \leq -\sqrt{\rho\epsilon}/4$, as long as $\lambda_{\min}\left(\mathcal{H}(\mathbf{x}_{t+1})\right) \leq -\sqrt{\rho\epsilon}$. Under this assumption, the uniform perturbation can be called for at most $N_{\mathscr{T}} = 384\Delta_\Phi\sqrt{\frac{\rho}{\epsilon^3}}$ times, for otherwise the function value decrease will be greater than $\Phi\left(\mathbf{x}_0\right) - \Phi^*$, which is not possible. Then, the probability that at least one negative curvature finding subroutine after uniform perturbation fails is upper bounded by

$$384\Delta_\Phi\sqrt{\frac{\rho}{\epsilon^3}} \cdot \delta_0 = \delta \tag{134}$$

For the rest of steps which is not within $\mathscr{T}$ steps after uniform perturbation, they are either descent steps, $\|\nabla\Phi(\mathbf{x}_t)\| \geq 82\epsilon$, or $(\epsilon, \sqrt{\epsilon})$-second-order second-order stationary points. Next, we

demonstrate that at least one of these steps is an $(\epsilon, \sqrt{\epsilon})$-second-order stationary point. Assume the contrary. We use $N_K$ to denote the number of epochs where Line 9 triggers. Therefore, it satisfies

$$T \leq N_K \cdot K + N_{\mathscr{T}} \cdot (K + \mathscr{T}) \tag{135}$$

Then, we have

$$N_K \geq N_K \cdot \frac{K}{K + \mathscr{T}} \geq \frac{T}{K + \mathscr{T}} - N_{\mathscr{T}} \geq \frac{T}{K + \mathscr{T}} - 384\Delta_\Phi\sqrt{\frac{\rho}{\epsilon^3}} \geq \frac{308\Delta_\Phi}{\mathscr{F}} - 384\Delta_\Phi\sqrt{\frac{\rho}{\epsilon^3}} \geq \frac{\Delta_\Phi}{\mathscr{F}} \tag{136}$$

During these iterations the function value of $\Phi$ will decrease in total at least $N_K \cdot \mathscr{F} \geq \Delta_\Phi$, which is impossible due to Lemma 4.5, the function value of $\Phi$ decreases monotonically for every epoch except when the Line 11 triggers and the $\mathscr{T}$ steps after uniform perturbation, and the overall decrease cannot be greater than $\Delta_\Phi$. Therefore, we conclude that at least one of the iterations must be an $(\epsilon, \sqrt{\epsilon})$ second-order stationary point, with probability at least $1 - \delta$.

