# OpenReview forum: "Finding Second-order Stationary Points for Generalized-Smooth Nonconvex Minimax Optimization via Gradient-based Algorithm"
_ICLR.cc/2025/Conference — Submitted to ICLR 2025_

### Official Review · Reviewer_ZMsE · 2024-10-24

**Soundness:** 3
**Presentation:** 4
**Contribution:** 3
**Rating:** 6
**Confidence:** 4

**Summary:**

In this paper, the authors consider the nonconvex minimax problems. Specifically, the paper proposes a gradient-based accelerated method for finding a second-order stationary point  for the nonconvex-strongly-concave minimax optimization under the so-called generalized smoothness condition. This can be regarded as extension of the existing work Luo et al. (2022), Huang et al. (2022) and Yang et al. (2023) with the Lipschitz Hessian condition.  The theoretical comparison is listed clearly in Table 1.

**Strengths:**

The paper addresses an important theoretical questions on designing efficient optimization algorithms for finding second-order stationary points of nonconvex minimax problems under general smooth condition.  It extended the existing work to more general cases which can arise in various machine learning problem.

**Weaknesses:**

1. While the extension to the general smooth case is interesting and important, the design of optimization algorithms relies on  the combination of the Nesterov’s classical Accelerated Gradient Descent  and tricks from Li and Lin (2022). Could the authors comment what are the main novelty on the algorithmic design compared with the existing literature?

2. The proofs seem to heavily depend on the previous work Luo et al. (2022), Huang et al. (2022) and Yang et al. (2023) which assume Lipschitz Hessian condition.  The proof novelty can be incremental (see my questions below).

3.  The computational gain from the experiment compared with RAHGD and Clip RAHGD seems limited.

**Questions:**

1. The authors summarised the main challenges in Section 4.1. It would be helpful if  the authors describe in the proof sketch which lemmas are main novel ones.

2. The formulation of Domain-Adversarial Neural Network given by (9) seems not right.  There are no label for target domain. In the loss function $L_2$ contain $y$ for target domain. Do I miss something here?

---

> ### Author Response · Authors · 2024-11-22
> **Reply to Reviewer ZMsE (Part 1)**
>
> Thank you very much for your constructive comments and suggestions!
>
> ___
> **Q1**: While the extension to the general smooth case is interesting and important, the design of optimization algorithms relies on the combination of the Nesterov’s classical Accelerated Gradient Descent and tricks from Li and Lin (2022). Could the authors comment what are the main novelty on the algorithmic design compared with the existing literature?
>
> **A1**: Thank you for your valuable comment. After sufficient Nesterov’s Accelerated Gradient Descent steps are performed with restart technique to ensure the function value of $\\Phi$ decrease monotonically, **the negative curvature finding procedure that inspired by the idea of escaping saddle points is performed, which is the main novelty of the algorithmic design.**
>
> For the procedure of negative curvature finding procedure, we define a function $h\_{\\Phi}(\\mathbf{x})$ in Eq. (80) such that $h\_{\\Phi}(\\mathbf{x})=\\Phi(\\mathbf{x})-\\langle\\nabla \\Phi(\\hat{\\mathbf{z}}), \\mathbf{x}\\rangle$, as shown in Appendix D, and estimate the derivative of $h\_{\\Phi}(\\mathbf{x})$ with $\\zeta=\\widehat\\nabla \\Phi(\\hat{\\mathbf{z}})=\\nabla\_{\\mathbf{x}} f(\\hat{\\mathbf{z}}, \\hat{\\mathbf{y}})$, which is attained in Line 15, and $\\widehat\\nabla h\_{\\Phi}(\\mathbf{z}\_{t})=\\widehat\\nabla \\Phi(\\mathbf{z}\_{t})-\\widehat\\nabla \\Phi(\\hat{\\mathbf{z}})=\\nabla\_{\\mathbf{x}} f(\\mathbf{z}\_{t}, \\mathbf{y}\_{t})-\\zeta$ in the following epoch with $\\mathscr{T}$ iterations. Here, instead of the classical Nesterov’s Accelerated Gradient Descent steps in the epochs with $\\zeta=0$, the algorithm first obtains $\\hat{\\mathbf{x}}\_{t+1}$ (represented as $\\mathbf{x}\_{t+1}$ in Line 5) with $\\hat{\\mathbf{x}}\_{t+1}=\\mathbf{z}\_t-\\eta\_{\\mathbf{x}} \\cdot \\widehat\\nabla h\_{\\Phi}(\\mathbf{z}\_{t})$ and then updates $\\mathbf{x}\_{t+1}$ in Line 19 by $\\mathbf{x}\_{t+1}=\\hat{\\mathbf{z}}+r \\cdot \\frac{\\hat{\\mathbf{x}}\_{t+1}-\\hat{\\mathbf{z}}}{\|\mathbf{z}\_{t+1}-\\hat{\\mathbf{z}}\\|}$ for every iteration in this epoch, where the algorithm try to find a negative curvature direction around $\\hat{\\mathbf{z}}$ with a radius of $r$. Then after $\\mathscr{T}$ iterations, Lemma 4.7 implies that the unit vector $\\hat{\\mathbf{e}}$ is a negative curvature direction with high possibility and the value of function $\\Phi$ will decrease by the one-step descent along the direction in Line 22 according to Lemma 4.7. Further details of the proof can be found in Appendix D.
>
> ___
> **Q2**: The proofs seem to heavily depend on the previous work Luo et al. (2022), Huang et al. (2022) and Yang et al. (2023) which assume Lipschitz Hessian condition. The proof novelty can be incremental. The authors summarised the main challenges in Section 4.1. It would be helpful if the authors describe in the proof sketch which lemmas are main novel ones.
>
> **A2**: Thank you for your constructive comment. We will add the following discussion to the revised paper.
>
> Firstly, the main novelty compared with Luo et al. (2022) [1], Huang et al. (2022) [2] and Yang et al. (2023) [3] is that we proposed a generalized Lipschitz Hessian assumption for the objective function $f$ of minimax optimization in Definition 3.4 that is much weaker than the classical Hessian Lipschitz assumption in the previous works and under such assumptions **we further proposed Lemma 4.2**, which is a key technical lemma to prove that the primal function $\\Phi$ and $\\mathbf{y}^*$ is still well-defined under the relaxed smoothness condition on $f$ and analyse the structure and the generalized smoothness properties on $\\Phi$ and $\\mathbf{y}^{*}$, which depends on the generalized smoothness constants of $f$ and $\\mathbf{x}\_0$. It is quite significant because compared with Huang et al. (2022) [2] and Yang et al. (2023) [3], the analysis on the error term of the hypergradient estimator $\\|\\nabla \\Phi(\\mathbf{x}\_t)-\\widehat{\\nabla} \\Phi(\\mathbf{x}\_t)\\|$ become much difficult under generalized smoothness, whose upper bound depends on $l\_{\\mathbf{x}, 1} \\|\\mathbf{y}\_t-\\mathbf{y}\_{t}^{\*}\\| \\|\\nabla \\Phi(\\mathbf{x}\_t)\\|$. This quantity is difficult to handle because $\\|\\nabla \\Phi(\\mathbf{x}\_t)\\|$ can be large, and it is difficult to decouple the two measurable term $\\|\\mathbf{y}\_t-\\mathbf{y}\_{t}^{\*}\\|$ and $\\|\\nabla \\Phi(\\mathbf{x}\_t)\\|$.
>
> Also, in Lemma 4.7 and 4.8, we introduced two important lemmas for escaping saddle points by negative curvature direction finding technical, which guarantee the algorithm can find second-order stationary points with high possibility under the generalized hessian continuous assumption in $\\mathcal{O}(\\epsilon^{-1.75}\\log n)$. Compared with the results $\\mathcal{O}(\\epsilon^{-1.75}\\log^6 n)$ in Yang et al. (2023) [3], it improves the complexity with $\\mathcal{O}(\\log^5 n)$ even under the much weaker assumptions.

---

> ### Author Response · Authors · 2024-11-22
> **Reply to Reviewer ZMsE (Part 2)**
>
> **Q3**: The computational gain from the experiment compared with RAHGD and Clip RAHGD seems limited.
>
> **A3**: In the experiments, the test accuracy for the adaptation from both the SVHN and MNIST-M to MNIST show that our proposed ANCGDA algorithm converge faster and have better convergence result compared with RAHGD[3] and Clip RAHGD for these two 28x28 small size picture datasets. It shows that the experiments results are consistent with the theoretical complexity gap $\\mathcal{O}(\\log^5 n)$ between ANCGDA and RAHGD, which implies that the performance gap can be more obvious for the real-world datasets with large dimensions $n$. We will conduct more experiments on large-scale datasets in our future work.
>
> ___
> **Q4**: The formulation of Domain-Adversarial Neural Network given by (9) seems not right. There are no label for target domain. In the loss function $L\_{2}$ contain $y$ for target domain. Do I miss something here?
>
> **A4**: Sorry for omitting the details. We gives the missing details as follows, which will be added to our revised paper.
>
> The target domain dataset $\\mathcal{T}=\\{\\mathbf{a}\_i^{\\mathcal{T}}\\}\_{i=1}^{N\_{\\mathcal{T}}}$ only contains features.
> $\\Phi$ is a single-layer neural network as the feature extractor with the size of (28 $\\times$  28) $\\times$ 200 with
> parameter $\\mathbf{x}\_1$ and $\\ell$ is a two-layer neural network as the domain classifier with the size of 200 $\\times$ 20 $\\times$ 10 with parameter $\\mathbf{x}\_2$, followed by a cross entropy loss. For the logistic loss functions for $L\_2$, we let $h(\\mathbf{y} ; \\mathbf{z})=1 /(1+\\exp (-\\mathbf{y}^{\\top} \\mathbf{z})), D\_S(z)=1-\\log (z)$ and $D\_{\\mathcal{T}}(z)=\\log (1-z)$.
> ___
> **Reference**:
> [1] Luo L, Li Y, Chen C. Finding second-order stationary points in nonconvex-strongly-concave minimax optimization[J]. Advances in Neural Information Processing Systems, 2022, 35: 36667-36679.
> [2] Huang M, Chen X, Ji K, et al. Efficiently escaping saddle points in bilevel optimization[J]. arXiv preprint arXiv:2202.03684, 2022.
> [3] Yang H, Luo L, Li C J, et al. Accelerating inexact hypergradient descent for bilevel optimization[J]. arXiv preprint arXiv:2307.00126, 2023.

---

> > ### Comment · Reviewer_ZMsE · 2024-11-27
> > **thanks**
> >
> > Thank you for the detailed response.  I keep my score.

---

> ### Author Response · Authors · 2024-11-30
>
> We have conducted three numerical experiments to compare the performance of our algorithm with PRAHGD for the efficiency of escaping saddle points on $\\Phi(\\mathbf{x})$. In this table we gives the number of iterations for the two algorithms to make sure 90% of samplings have a descent value of $\\Delta\\Phi \\geq 0.9$, with $\\Phi_{quartic}(\\mathbf{x}\_1, \\mathbf{x}\_2)=\\frac{1}{16} \\mathbf{x}\_1^4-\\frac{1}{2} \\mathbf{x}\_1^2+\\frac{9}{8} \\mathbf{x}\_2^2$,  $\\Phi_{exp}(\\mathbf{x}\_1, \\mathbf{x}\_2)=\frac{1}{1+e^{\\mathbf{x}\_1^2}}+\frac{1}{2}(\\mathbf{x}\_2-\\mathbf{x}\_1^2 e^{-\\mathbf{x}\_1^2})^2-1$ and
> $\\Phi_{tri}(\\mathbf{x}\_1, \\mathbf{x}\_2)=\frac{1}{2} \cos (\pi \\mathbf{x}\_1)+\frac{1}{2}(\\mathbf{x}\_2+\frac{\cos (2 \pi \\mathbf{x}\_1)-1}{2})^2-\frac{1}{2}$, respectively.
> All the three functions have a saddle point at $(0, 0)$. Each experiment is conducted with same $\\eta$ and $r$ for 300 samples that generated in a Gaussian ball around $(0, 0)$. We will add these experiments in the final version of the paper.
> | Method | $\\Phi_{quartic}$ (Iterations)| $\\Phi_{exp}$ (Iterations)| $\\Phi_{tri}$ (Iterations)|
> | ---- |  :----: | :----: | :----: |
> | PRAHGD | 45 | 72 | 24 |
> | ANCGDA (Ours) | **19** |**22** | **6** |

---

### Official Review · Reviewer_M2QX · 2024-10-24

**Soundness:** 3
**Presentation:** 2
**Contribution:** 3
**Rating:** 6
**Confidence:** 2

**Summary:**

This paper provides the first first-order algorithm for non-stochastic generalized-smooth nonconvex-strongly-concave minimax optimization that achieves iteration complexity even better than SOTA for Lipschitz-smooth setting.

**Strengths:**

As shown in my summary, the claim of both weaker smoothness assumption and stronger result (lower iteration complexity) is significant.

**Weaknesses:**

Some points need to be clarified as shown below, especially the question 1 below about intuition of algorithm design. A possible proof error may invalidate the main convergence result as shown in question 14 below.

**Questions:**

(1) In line 5 of Algorithm 1, why can $\zeta=\nabla_x f(\hat{z},\hat{y})$ help get rid of saddle point? At the end of Algorithm 1, why is $\hat{e}$ the NC direction? In the revised paper, could you provide intuitive explanations of these questions or cite the works that contains such explanations?

(2) In line 88, by "Speically", do you mean "Specially" or "Specifically"?

(3) In line 103, do you mean to change $\mathcal{O}(\sqrt{\kappa}\epsilon^2)$ to $\mathcal{O}(\sqrt{\kappa}\epsilon^{-2})$?

(4) In line 216, "with Eq. (2)".

(5) In Eq. (5) of Lemma 4.2, can $\max\{u\ge 0|u^2\le 2\mathcal{L}[\Phi(x_0)-\Phi^*]\}$ be simplified to $\sqrt{2\mathcal{L}[\Phi(x_0)-\Phi^*]}$?

(6) In line 320, "reset to be an epoch" might be read together. Do you mean "we define an epoch to be a round from $k=0$ to the iteration that triggers one of these conditions and resets k to 0"?

(7) In line 325, does "hypergradient estimation of $\hat{z}$" mean "estimation of the hypergradient $\nabla\Phi(\hat{z})$"?

(8) In line 331: "an second-order" to "a second-order".

(9) You could define $\Phi^*$ around its first appearance.

(10) In Lemma 4.2, $G$ depends on $x_0$ but the conclusions do not depend on $x_0$, why? Can $x_0$ be any arbitrary point? Should we add $\|x'-x\|$ to the end of the equation of item 4, based on Eq. (36)? Adding this also will make the conclusion consistent with the inequality $\|\nabla^2 f(\mathbf{u})-\nabla^2 f(\mathbf{u}^{\prime})\| \leq(\rho_0+\rho_1\|\nabla f(x)\|)\|\mathbf{u}-\mathbf{u}^{\prime}\|$ in Definition 3.2 which contains $\|u'-u\|$.

(11) $t_0$ and $t_K$ are defined but not used in Lemmas 4.6 and 4.7.

(12) Right under Eq. (8), what are the roles of $\Phi$ and $\ell$ (for examples, are they predictor and loss function respectively)? In Eq. (9), what are $D_{\mathcal{S}}$, $D_{\mathcal{T}}$ and $h$?

(13) Some other hyperparameter choices like $B,r,K,\mathcal{J}$ could be revealed in your experiment for reproducibility.

(14) Possible error in Proof of Lemma 4.6: Should the second $\tilde{x} _ {t_0}$ be $\tilde{x} _ {t_0-1}$? Since their distance is not surely to be smaller than $\mathcal{O}(\epsilon)$, I am not sure if Lemma 4.6 holds.

---

> ### Author Response · Authors · 2024-11-22
> **Reply to Reviewer M2QX (Part 1)**
>
> Thanks for your constructive comments, and we have revised our paper accordingly! We have addressed the concerns about the questions as outlined below.
> ___
> **Q1**: In line 5 of Algorithm 1, why can $\\zeta=\\nabla\_x f(\\hat{z}, \\hat{y})$ help get rid of saddle point? At the end of Algorithm 1, why is $\\hat{e}$ the NC direction? In the revised paper, could you provide intuitive explanations of these questions or cite the works that contains such explanations?
>
> **A1**: Sorry for omitting the details. We provide a more detailed analysis as follows, which have been added to our revised paper.
>
> Firstly, to analysis the procedure of escaping saddle points after sufficient descent for finding first-order stationary points, we define a function $h\_{\\Phi}(\\mathbf{x})$ in Eq. (80) such that $h\_{\\Phi}(\\mathbf{x})=\\Phi(\\mathbf{x})-\\langle\\nabla \\Phi(\\hat{\\mathbf{z}}), \\mathbf{x}\\rangle$ (while assuming $\\hat{\\mathbf{z}}=\\mathbf{0}$ by shifting $\\mathbb{R}^n$ so that $\\hat{\\mathbf{z}}$ is mapped to $\\mathbf{0}$ for convenience without loss of generality). Then we have the derivative of the function $\\nabla h\_{\\Phi}(\\mathbf{x})=\\nabla \\Phi(\\mathbf{x})-\\nabla \\Phi(\\hat{\\mathbf{z}})$. So for the iteration $t$ that uniform perturbation is added, we can estimate the derivative of $h\_{\\Phi}(\\mathbf{x})$ with $\\zeta=\\widehat\\nabla \\Phi(\\hat{\\mathbf{z}})=\\nabla\_{\\mathbf{x}} f(\\hat{\\mathbf{z}}, \\hat{\\mathbf{y}})$ and $\\widehat\\nabla h\_{\\Phi}(\\mathbf{z}\_{t})=\\widehat\\nabla \\Phi(\\mathbf{z}\_{t})-\\widehat\\nabla \\Phi(\\hat{\\mathbf{z}})=\\nabla\_{\\mathbf{x}} f(\\mathbf{z}\_{t}, \\mathbf{y}\_{t})-\\zeta$ to update $\\mathbf{x}\_{t+1}$ in the escaping saddle point procedure and the estimation error can be bounded by $\\|\\delta\_{\widehat\\Phi}\\|=\mathcal{O}(\epsilon)$ according to Lemma 4.4. Further details of $h\_{\\Phi}(\\mathbf{x})$ can be founded in appendix D.
>
> Then for the unit vector $\\hat{\\mathbf{e}}$, Lemma 4.7 yeilds that for the point $\\hat{\\mathbf{z}}$ satisfying $\\lambda\_{\\min }\\left(\\nabla^2 \\Phi(\\hat{\\mathbf{z}})\\right) \\leq-\\sqrt{\\rho \\epsilon}$, after $\\mathscr{T}$ iterations of the negative curvature (NC) finding procedure with possibility of $1-\\delta\_{0}$, the obtained $\\hat{e}$ satisfies $\\hat{\\mathbf{e}}^T \\mathcal{H}(\\hat{\\mathbf{z}}) \\hat{\\mathbf{e}} \\leq -\\sqrt{\\rho \\epsilon} / 4$, where $\\mathcal{H}$ stands for the Hessian matrix of function $\\Phi$. Here, we take the definition of negative curvature direction from Xu et al. (2018) [1], which implies that for a non-degenerate saddle point $\\mathbf{x}$ of a function $f(\\mathbf{x})$ with $\\|\\nabla f(\\mathbf{x})\\| \\leq \\epsilon$ and $\\lambda\_{\\min }\\left(\\nabla^2 f(\\mathbf{x})\\right) \\leq-\\gamma$, the negative curvature direction $\\mathbf{v}$ satisfies $\\|\\mathbf{v}\\|=1$ and $\mathbf{v}^{\top} \nabla^2 f(\mathbf{x}) \mathbf{v} \leq-c \gamma$. Taking $c=\\frac{1}{4}$ and $\gamma=\\sqrt{\\rho \\epsilon}$ yields that the obtained $\\hat{\\mathbf{e}}$ is a NC direction. Further proof of Lemma 4.6 and 4.7 can be found in appendix D and the NC finding procedure is inspired by Xu et al. (2018) [1] and Zhang et al. (2021) [2].
> ___
> **Q2, 3, 4, 6, 7, 8**: There are some typos in the paper. Also, some expression is not proper and can be misread.
>
> **A**: Thank you for catching the typos! We have corrected them in the revised version.
>
> ___
> **Q5, 9**: In Eq. (5) of Lemma 4.2, can $\\max u \\geq 0 \\mid u^2 \\leq 2 \\mathcal{L}\\left[\\Phi\\left(x\_0\\right)-\\Phi^*\\right]$ be simplified to $\\sqrt{2\\mathcal{L}\\left[\\Phi\\left(x\_0\\right)-\Phi^*\\right]}$? Also, you could define $\\Phi^*$ around its first appearance.
>
> **A**: Thank you for your suggestion! We have simplified the definition of $G$ and add the definition of $\\Phi^*$ in Lemma 4.2.
> ___
> **Reference**:
> [1] Xu Y, Jin R, Yang T. First-order stochastic algorithms for escaping from saddle points in almost linear time[J]. Advances in neural information processing systems, 2018, 31.
> [2] Zhang C, Li T. Escape saddle points by a simple gradient-descent based algorithm[J]. Advances in Neural Information Processing Systems, 2021, 34: 8545-8556.

---

> ### Author Response · Authors · 2024-11-22
> **Reply to Reviewer M2QX (Part 2)**
>
> **Q10**: In Lemma 4.2, $G$ depends on $x\_{0}$ but the conclusions do not depend on $x\_{0}$, why? Can $x\_{0}$ be any arbitrary point? Should we add $\\left|x^{\\prime}-x\\right|$ to the end of the equation of item 4, based on Eq. (36)? Adding this also will make the conclusion consistent with the inequality $\\left|\\nabla^2 f(\\mathbf{u})-\\nabla^2 f\\left(\\mathbf{u}^{\\prime}\\right)\\right| \\leq\\left(\\rho_0+\\rho_1|\\nabla f(x)|\\right)\\left|\\mathbf{u}-\\mathbf{u}^{\\prime}\\right|$ in Definition 3.2 which contains $\\left|u^{\\prime}-u\\right|$.
>
> **A10**: Thanks for your constructive comments for Lemma 4.2. We apologize for the confusion. Here $\\mathbf{x}_{0}$ can be any arbitrary point as $\\mathbf{x}$ and $\\mathbf{x}^\\prime$ satisfy the generalized smoothness assumptions and Eq. (6) is presented improperly. We corrected it with $\\mathbf{x},\\ \\mathbf{x}^{\\prime} \\subseteq \\mathcal{B}(\\mathbf{x}\_0, \\frac{G\\mu}{\\mathcal{L}(\\mu+l\_{\\mathbf{y}, 0})})$, where $\\mathcal{B}(x, R)$ denotes the Euclidean ball with radius R centered at x, which is consistent with the further proof and conclusions. Also we added the missing term of $\\left|x^{\\prime}-x\\right|$ in item 4. Thanks again for pointing this mistake.
> ___
> **Q11**: $t\_{0}$ and $t\_{K}$ are defined but not used in Lemmas 4.6 and 4.7.
>
> **A11**:  Thank you for your suggestion. We removed the definition of $t\_{0}$ in Lemma 4.7 and added it to the proof in appendix D. Also, we rewrite the Lemma 4.6 as follows:
>
> *Lemma 4.6*. Running Algorithm 1 with parameters setting in Theorem 4.3. In the epoch that the condition on Line 11 triggers, the point $\\hat{\\mathbf{z}}$ in Line 13 satisfies $\\|\\nabla\\Phi(\\hat{\\mathbf{z}})\\| ≤ \mathcal{O}(\epsilon)$.
> ___
> **Q12, 13**: About the experiment. Right under Eq. (8), what are the roles of $\\Phi$ and $\\ell$ (for examples, are they predictor and loss function respectively)? In Eq. (9), what are $D\_{S}$, $D\_{\\mathcal{T}}$ and $h$? Also, some hyperparameter choices like $B$, $r$, $K$, $\\mathscr{T}$ could be revealed in your experiment for reproducibility.
>
> **A**:  Sorry for omitting the details. We gives the missing details as follows, which have been added to our revised paper.
> Here $\\Phi$ is a single-layer neural network as the feature extractor with the size of (28 $\\times$  28) $\\times$ 200 with
> parameter $\\mathbf{x}\_1$ and $\\ell$ is a two-layer neural network as the domain classifier with the size of 200 $\\times$ 20 $\\times$ 10 with parameter $\\mathbf{x}\_2$, followed by a cross entropy loss. For the logistic loss functions for $L\_2$, we let $h(\\mathbf{y} ; \\mathbf{z})=1 /(1+\\exp (-\\mathbf{y}^{\\top} \\mathbf{z})), D\_S(z)=1-\\log (z)$ and $D\_{\\mathcal{T}}(z)=\\log (1-z)$.
>
> About the other hyperparameters for ANCGDA, RAHGD and Clipping RAHGD, we choose $r=0.04$,  $K=30$ and $\\mathscr{T}=10$ for both the source domain dataset while setting $B=10$ for SVHN as source dataset and $B=7$ for MNIST-M.
> ___
> **Q14**: Possible error in Proof of Lemma 4.6: Should the second $\\tilde{x}\_{t\_0}$ be
> $\\tilde{x}\_{t\_0-1}$? Since their distance is not surely to be smaller than $\mathcal{O}(\epsilon)$, I am not sure if Lemma 4.6 holds.
>
> **A14**:  We apologize for the confusion. In the algorithm, we actually reset $\\mathbf{z}\_{t+1}=\\mathbf{x}\_{t+1}$ at the end of each epoch, then the beginning of the proof for Lemma 4.6 should be as follows:
> $$
> \\|\\nabla g(\\tilde{\\mathbf{z}})\\|
> =\\left\\|\\frac{1}{K\_{0}+1} \\sum\_{k=t\_{0}}^{t\_{0}+K\_{0}} \\nabla g(\\tilde{\\mathbf{z}}\_{k})\\right\\|
> = \\frac{1}{\\eta\_{\\mathbf{x}}(K\_{0}+1)}\\left\\|\\sum\_{k=t\_{0}}^{t\_{0}+K\_{0}}(\\tilde{\\mathbf{x}}\_{k+1}-\\tilde{\\mathbf{z}}\_{k}+\\eta\_{\\mathbf{x}} \\tilde{\\delta}\_{k})\\right\\|
> $$
> $$
> \\stackrel{a}{=} \\frac{1}{\\eta\_{\\mathbf{x}}(K\_{0}+1)}\\left\\|\\sum\_{k=t\_{0}}^{t\_{0}+K\_{0}}(\\tilde{\\mathbf{x}}\_{k+1}-\\tilde{\\mathbf{x}}\_{k}+\\eta\_{\\mathbf{x}} \\tilde{\\delta}\_{k})-\\sum\_{k=t\_{0}+1}^{t\_{0}+K\_{0}}(1-\\theta\_{\\mathbf{x}})(\\tilde{\\mathbf{x}}\_{k}-\\tilde{\\mathbf{x}}\_{k-1})\\right\\|
> $$
> $$
> = \\frac{1}{\\eta\_{\\mathbf{x}}(K\_{0}+1)}\\left\\|\\tilde{\\mathbf{x}}\_{t\_{0}+K\_{0}+1}-\\tilde{\\mathbf{x}}\_{t\_{0}}-(1-\\theta\_{\\mathbf{x}})(\\tilde{\\mathbf{x}}\_{t\_{0}+K\_{0}}-\\tilde{\\mathbf{x}}\_{t\_{0}})+\\eta\_{\\mathbf{x}} \\sum\_{k=t\_{0}}^{t\_{0}+K\_{0}} \\tilde{\\delta}\_{k}\\right\\|
> $$
> $$
> = \\frac{1}{\\eta\_{\\mathbf{x}}(K\_{0}+1)}\\left\\|\\tilde{\\mathbf{x}}\_{t\_{0}+K\_{0}+1}-\\tilde{\\mathbf{x}}\_{t\_{0}+K\_{0}}+\\theta\_{\\mathbf{x}}(\\tilde{\\mathbf{x}}\_{t\_{0}+K\_{0}}-\\tilde{\\mathbf{x}}\_{t\_{0}})+\\eta\_{\\mathbf{x}} \\sum\_{k=t\_{0}}^{t\_{0}+K\_{0}} \\tilde{\\delta}\_{k}\\right\\|
> $$
> $$
> ...,
> $$
> where $\\stackrel{a}{=}$ holds because $\\mathbf{z}\_{t\_0}=\\mathbf{x}\_{t\_0}$. So there is no $\\mathbf{x}\_{t\_0-1}$ term in Eq. (75) and the proof implies that **Lemma 4.6 holds properly**.
> ___

---

> > ### Comment · Reviewer_M2QX · 2024-11-22
> > **I'm satisfied with the response and increased my rating to 6.**
> >
> > I'm satisfied with the response and increased my rating to 6.
> >
> > By the way, Eq. (6) could be changed to $\mathbf{x},\ \mathbf{x}^{\prime} \in \mathcal{B}\Big(\mathbf{x} _ 0, \frac{G\mu}{\mathcal{L}(\mu+l _ {\mathbf{y}, 0})}\Big)$ with big parenthesis and $\in$.
> >
> > Thanks for your efforts for elaboration and improvements.

---

> > > ### Author Response · Authors · 2024-11-23
> > >
> > > Thank you very much for your kind reply! We have changed Eq. (6) in the revised paper according to your suggestions.

---

### Official Review · Reviewer_8wxo · 2024-10-30

**Soundness:** 2
**Presentation:** 2
**Contribution:** 2
**Rating:** 3
**Confidence:** 5

**Summary:**

This paper studied the nonconvex-strongly-convex minimax under the generalized-smooth setting, and proposed a ANCGDA method to find a second-order stationary point in nonconvex-strongly-concave minimax optimization with generalized smoothness. It provided the convergence analysis of the proposed method under some conditions. It also provided some numerical experimental results on the proposed method.

**Strengths:**

This paper proposed a ANCGDA method to find a second-order stationary point in nonconvex-strongly-concave minimax optimization with generalized smoothness. It provided the convergence analysis of the proposed method under some conditions. It also provided some numerical experimental results on the proposed method.

**Weaknesses:**

From algorithm 1 of this paper, the proposed algorithm basically extends the existing methods in [1] to minimax optimization. The novelty of the proposed algorithm is limited. Meanwhile, the convergence analysis mainly follows the exiting results.


[1] Huan Li and Zhouchen Lin. Restarted nonconvex accelerated gradient descent: No more polylogarithmic factor in the complexity. In International Conference on Machine Learning, pp. 12901–12916. PMLR, 2022.

The numerical experiments in the paper are limited. The authors should add some numerical experiments such as GAN and multi-agent reinforcement learning.

**Questions:**

From Figure 1, what does the horizontal axis represent?

Under the Definition 3.4, the generalized Hessian continuous condition may be generate the smoothness condition ? If not, please give some negative examples.

What is the full name of ANCGDA  in the paper?

---

> ### Author Response · Authors · 2024-11-22
> **Reply to Reviewer 8wxo (Part 1)**
>
> Thanks for your constructive comments. We have addressed the major concerns about the paper as outlined below.
> ___
> **Q1**: From algorithm 1 of this paper, the proposed algorithm basically extends the existing methods in [1] to minimax optimization. The novelty of the proposed algorithm is limited. Meanwhile, the convergence analysis mainly follows the exiting results.
>
> **A1**: Thank you for your valuable comment.
>
> **About the algorithmic design.**
>
> After sufficient Nesterov’s Accelerated Gradient Descent steps are performed with restart technique to ensure the function value of $\\Phi$ decrease monotonically, the negative curvature finding procedure that inspired by the idea of escaping saddle points is performed, which is the main novelty of the proposed algorithm.
> For the procedure of negative curvature finding, we define a function $h\_{\\Phi}(\\mathbf{x})$ in Eq. (80) such that $h\_{\\Phi}(\\mathbf{x})=\\Phi(\\mathbf{x})-\\langle\\nabla \\Phi(\\hat{\\mathbf{z}}), \\mathbf{x}\\rangle$, as shown in Appendix D, and estimate the derivative of $h\_{\\Phi}(\\mathbf{x})$ with $\\zeta=\\widehat\\nabla \\Phi(\\hat{\\mathbf{z}})=\\nabla\_{\\mathbf{x}} f(\\hat{\\mathbf{z}}, \\hat{\\mathbf{y}})$, which is attained in Line 15, and $\\widehat\\nabla h\_{\\Phi}(\\mathbf{z}\_{t})=\\widehat\\nabla \\Phi(\\mathbf{z}\_{t})-\\widehat\\nabla \\Phi(\\hat{\\mathbf{z}})=\\nabla\_{\\mathbf{x}} f(\\mathbf{z}\_{t}, \\mathbf{y}\_{t})-\\zeta$ in the following epoch with $\\mathscr{T}$ iterations. Here, instead of the classical Nesterov’s Accelerated Gradient Descent steps in the epochs with $\\zeta=0$, the algorithm first obtains $\\hat{\\mathbf{x}}\_{t+1}$ (represented as $\\mathbf{x}\_{t+1}$ in Line 5) with $\\hat{\\mathbf{x}}\_{t+1}=\\mathbf{z}\_t-\\eta\_{\\mathbf{x}} \\cdot \\widehat\\nabla h\_{\\Phi}(\\mathbf{z}\_{t})$ and then updates $\\mathbf{x}\_{t+1}$ in Line 19 by $\\mathbf{x}\_{t+1}=\\hat{\\mathbf{z}}+r \\cdot \\frac{\\hat{\\mathbf{x}}\_{t+1}-\\hat{\\mathbf{z}}}{\|\mathbf{z}\_{t+1}-\\hat{\\mathbf{z}}\\|}$ for every iteration in this epoch, where the algorithm try to find a negative curvature direction around $\\hat{\\mathbf{z}}$ with a radius of $r$. Then after $\\mathscr{T}$ iterations, Lemma 4.7 implies that the unit vector $\\hat{\\mathbf{e}}$ is a negative curvature direction with high possibility and the value of function $\\Phi$ will decrease by the one-step descent along the direction in Line 22 according to Lemma 4.7. Further details of the proof can be found in Appendix D.
>
> **About the convergence analysis.**
>
> Firstly, the main novelty compared with Luo et al. (2022) [2], Huang et al. (2022) [3] and Yang et al. (2023) [4] is that we proposed a generalized Lipschitz Hessian assumption for the objective function $f$ of minimax optimization in Definition 3.4 that is much weaker than the classical Hessian Lipschitz assumption in the previous works and under such assumptions we further proposed **Lemma 4.2**, which is a key technical lemma to prove that the primal function $\\Phi$ and $\\mathbf{y}^*$ is still well-defined under the relaxed smoothness condition on $f$ and analyse the structure and the generalized smoothness properties on $\\Phi$ and $\\mathbf{y}^{*}$, which depends on the generalized smoothness constants of $f$ and $\\mathbf{x}\_0$. It is quite significant because compared to Huang et al. (2022) [3] and Yang et al. (2023) [4], the analysis on the error term of the hypergradient estimator $\\|\\nabla \\Phi(\\mathbf{x}\_t)-\\widehat{\\nabla} \\Phi(\\mathbf{x}\_t)\\|$ become much difficult under generalized smoothness, whose upper bound depends on $l\_{\\mathbf{x}, 1} \\|\\mathbf{y}\_t-\\mathbf{y}\_{t}^{\*}\\| \\|\\nabla \\Phi(\\mathbf{x}\_t)\\|$. This quantity is difficult to handle because $\\|\\nabla \\Phi(\\mathbf{x}\_t)\\|$ can be large and it is difficult to decouple the two measurable term $\\|\\mathbf{y}\_t-\\mathbf{y}\_{t}^{\*}\\|$ and $\\|\\nabla \\Phi(\\mathbf{x}\_t)\\|$.  Leveraging by this important properties, we gives the analysis about the descent procedure shown in **Lemma 4.5 and 4.6**, which yield that algorithm can find first-order stationary points $\\hat{\\mathbf{z}}$ with suffcient small $\\|\\nabla \\Phi(\\hat{\\mathbf{z}})\\|$ as condition on Line 11 triggers.
>
> Then, in **Lemma 4.7 and 4.8**, we introduced two important lemmas for escaping saddle points by negative curvature direction finding technical, which guarantee the algorithm can find second-order stationary points with high possibility under the generalized hessian continuous assumption in $\\mathcal{O}(\\epsilon^{-1.75}\\log n)$. Compared with the results $\\mathcal{O}(\\epsilon^{-1.75}\\log^6 n)$ in Yang et al. (2023) [4], it improves the complexity with $\\mathcal{O}(\\log^5 n)$ even under the much weaker assumptions.

---

> ### Author Response · Authors · 2024-11-22
> **Reply to Reviewer 8wxo (Part 2)**
>
> **About the contribution.**
>
> We furtherly clarify the contributions of the proposed method as follows.
> ***Weaker assumptions.*** We first proposed a second-order theory of generalized smoothness condition for minimax optimization as shown in Definition 3.4, and conduct the new fundamental properties of the primal function $\\Phi$ and $\\mathbf{y}^{*}$ in Lemma 4.2 under the proposed second-order generalized smoothness condition, which is much weaker than the classical Lipschitz smoothness and Lipschitz Hessian continuous assumptions that are commonly used in previous works for both nonconvex optimization and minimax optimization like [1], [2], [3] and [4]. It extended the existing work to more general cases which can arise in various machine learning problem. To the best of our knowledge, this is the first work to show convergence for finding second-order stationary points on nonconvex minimax optimization with generalized smoothness.
>
> ***Lower complexity.***  We achieves the iteration complexity of $\\mathcal{O}(\\epsilon^{-1.75}\\log n)$ for finding a second-order stationary point in generalized smoothness nonconvex-strongly-concave minimax optimization, which is even better than SOTA for Lipschitz-smooth setting nonconvex-strongly-concave minimax optimization.
>
> ***Better experimental results.***  We conduct a numerical experiment on domain adaptation task to validate the practical performance of our method. We show that ANCGDA consistently outperforms other minimax optimization algorithms.
> ___
> **Q2**: What is the full name of ANCGDA in the paper? From Figure 1, what does the horizontal axis represent?
>
> **A2**: Sorry for omitting the details. We have added the full name of the algorithm (Accelerated Negative Curvature Gradient Descent Ascent) in the title of the Algorithm 1. The horizontal axis represent iterations and we have added it to Figure 1 in the revised paper.
> ___
> **Q3**: Under the Definition 3.4, the generalized Hessian continuous condition may be generate the smoothness condition ? If not, please give some negative examples.
>
> **A3**: According to Xie et al. (2024) [5], the second-order generalized smoothness of nonconvex optimization can be interpreted from the perspective of the boundness of higher-order derivatives, as the original definition of first-order generalized smoothness comes from the boundness of second-order derivatives in Zhang et al. (2019) [6]. We agree with the explanations in cited works.
> ___
> **Q4**: The numerical experiments in the paper are limited. The authors should add some numerical experiments such as GAN and multi-agent reinforcement learning.
>
> **A4**: Thank you for your suggestion! In the domain adaptation experiments, the test accuracy for the adaptation from both the SVHN and MNIST-M to MNIST show that our proposed ANCGDA algorithm converge faster and have better convergence result compared with RAHGD[3] and Clip RAHGD for these two 28x28 small size picture datasets. It shows that the experiments results are consistent with the theoretical complexity gap $\\mathcal{O}(\\log^5 n)$ between ANCGDA and RAHGD, which implies that the performance gap can be more obvious for the real-world datasets with large dimensions $n$. We will conduct more experiments on large-scale datasets in our future work.
> ___
> **Reference**:
> [1] Huan Li and Zhouchen Lin. Restarted nonconvex accelerated gradient descent: No more polylogarithmic factor in the complexity. In International Conference on Machine Learning, pp. 12901–12916. PMLR, 2022.
> [2] Luo L, Li Y, Chen C. Finding second-order stationary points in nonconvex-strongly-concave minimax optimization[J]. Advances in Neural Information Processing Systems, 2022, 35: 36667-36679.
> [3] Huang M, Chen X, Ji K, et al. Efficiently escaping saddle points in bilevel optimization[J]. arXiv preprint arXiv:2202.03684, 2022.
> [4] Yang H, Luo L, Li C J, et al. Accelerating inexact hypergradient descent for bilevel optimization[J]. arXiv preprint arXiv:2307.00126, 2023.
> [5] Xie C, Li C, Zhang C, et al. Trust Region Methods for Nonconvex Stochastic Optimization beyond Lipschitz Smoothness[C]//Proceedings of the AAAI Conference on Artificial Intelligence. 2024, 38(14): 16049-16057.
> [6] Zhang J, He T, Sra S, et al. Why Gradient Clipping Accelerates Training: A Theoretical Justification for Adaptivity[C]//International Conference on Learning Representations. 2019.

---

> > ### Comment · Reviewer_8wxo · 2024-11-28
> > **Reply to rebuttals**
> >
> > The authors basically did not deal with my concerns. I still concern the novelty of this paper. Meanwhile, the authors did not add any numerical experiments.
> >
> > Definition 3.4 in the paper is meaningless.
> > Following the generalized smoothness, this definition should be
> > $$||\nabla^2_{xx}f(u)-\nabla^2_{xx}f(u')||\leq (\rho_{x,0}+\rho_{x,1}\nabla^2_{xx}f(u))||u-u'||$$ and so on.
> > Clearly, Definition 3.4 in the paper is meaningless. I think that the authors use this meaningless definition to simply the convergence analysis in this paper.
> >
> > Thus, I keep my score.

---

> ### Author Response · Authors · 2024-11-30
>
> **About Definition 3.4.**
> In the Assumption 3 of Xie et al. (2024) [1], the authors gives a second-order theory of generalized smoothness for nonconvex optimization, that is,
> $$
> \\|\\nabla^2 F(x)-\\nabla^2 F(x^{\\prime})\\| \\leq(M\_0+M\_1\\|\\nabla F(x)\\|)\\|x-x^{\\prime}\\|,
> $$
> where F is twice-differentiable with some constants $M\_0$ and $M\_1$, interpreted from the perspective of the boundness of higher-order derivatives. So for minimax optimization we give a definition for generalized Hessian smoothness in definition 3.4. Also, it is reasonable because with the definition of $(L\_0, L\_1)$-smoothness in Zhang et al. (2019) [2], which is commonly accepted as the globle generalized smoothness assumption in nonconvex optimization, which reduces to the classical L-smoothness when $L\_0=L$ and $L\_1=0$, that is,
> $$
> \\|\\nabla^2 f(x)\\| \\leq L_0+L_1\\|\\nabla f(x)\\|.
> $$
> It shows  **the reasonableness** to present the generalized Hessian smoothness for minimax optimization with respect to $\\|\nabla\_\\mathbf{x}f(\\mathbf{u})\\|$ and $\\|\nabla\_\\mathbf{y}f(\\mathbf{u})\\|$ with constants $\\rho\_{\\mathbf{x}, 0}$, $\\rho\_{\\mathbf{x}, 1}$, $\\rho\_{\\mathbf{y}, 0}$, $\\rho\_{\\mathbf{y}, 1}$ and $\\rho\_{\\mathbf{xy}, 0}$, $\\rho\_{\\mathbf{xy}, 1}$, as  **the Definition 3.4 in our paper**, instead of $\\|\nabla^{2}\_\\mathbf{xx}f(\\mathbf{u})\\|$, $\\|\nabla^{2}\_\\mathbf{xy}f(\\mathbf{u})\\|$ and $\\|\nabla^{2}\_\\mathbf{yy}f(\\mathbf{u})\\|$ as you mentioned above.
> ___
> **About the experiment.**
> We have conducted three numerical experiments to compare the performance of our algorithm with PRAHGD for the efficiency of escaping saddle points on $\\Phi(\\mathbf{x})$. In this table we gives the number of iterations for the two algorithms to make sure 90% of samplings have a descent value of $\\Delta\\Phi \\geq 0.9$, with $\\Phi_{quartic}(\\mathbf{x}\_1, \\mathbf{x}\_2)=\\frac{1}{16} \\mathbf{x}\_1^4-\\frac{1}{2} \\mathbf{x}\_1^2+\\frac{9}{8} \\mathbf{x}\_2^2$,  $\\Phi_{exp}(\\mathbf{x}\_1, \\mathbf{x}\_2)=\frac{1}{1+e^{\\mathbf{x}\_1^2}}+\frac{1}{2}(\\mathbf{x}\_2-\\mathbf{x}\_1^2 e^{-\\mathbf{x}\_1^2})^2-1$ and
> $\\Phi_{tri}(\\mathbf{x}\_1, \\mathbf{x}\_2)=\frac{1}{2} \cos (\pi \\mathbf{x}\_1)+\frac{1}{2}(\\mathbf{x}\_2+\frac{\cos (2 \pi \\mathbf{x}\_1)-1}{2})^2-\frac{1}{2}$, respectively.
> All the three functions have a saddle point at $(0, 0)$. Each experiment is conducted with same $\\eta$ and $r$ for 300 samples that generated in a Gaussian ball around $(0, 0)$. We will add these experiments in the final version of the paper.
> | Method | $\\Phi_{quartic}$ (Iterations)| $\\Phi_{exp}$ (Iterations)| $\\Phi_{tri}$ (Iterations)|
> | ---- |  :----: | :----: | :----: |
> | PRAHGD | 45 | 72 | 24 |
> | ANCGDA (Ours) | **19** | **22** | **6** |
> ___
> Also, we think that **the main novelty** for our work is that we first proposed a **second-order theory of generalized smoothness for minimax optimization** while first guarantying the **convergence to second-order stationary points in nonconvex-strongly-concave settings** for our new algorithm in such **relaxed smoothness assumptions** while achieving the complexity that **even better than SOTA results in classical Lipschitz-smooth settings**. It is quite significant considering that all second-order stationary points of $\\Phi$ are (approximate) local minimax points of $f$ for nonconvex-strongly-concave minimax optimization. Relaxing the smoothness assumptions can extend the second-order convergence to more general cases of nonconvex minimax optimization and is more consistent with a class of well-studied nonconvex machine learning problem.
> ___
> **Reference**
> [1] Xie C, Li C, Zhang C, et al. Trust Region Methods for Nonconvex Stochastic Optimization beyond Lipschitz Smoothness[C]//Proceedings of the AAAI Conference on Artificial Intelligence. 2024, 38(14): 16049-16057.
> [2] Zhang J, He T, Sra S, et al. Why Gradient Clipping Accelerates Training: A Theoretical Justification for Adaptivity[C]//International Conference on Learning Representations. 2019.

---

### Meta-Review · Area_Chair_QqzV · 2024-12-19

**Metareview:**

Paper proposes a new class nonconvex-strongly-concave minimax problems with generalized first+second-order smoothness. Then it provides new algorithm and analysis which achieves new state-of-the-art iteration complexity under this assumption. Paper also provides neural network experiments however their rigor seems limited (e.g. no error bars). Authors are encouraged to improve clarity in writing so that novelty of their techniques and relevance of generalized second-order smoothness is better understood.

**Additional Comments On Reviewer Discussion:**

Authors were able to address most reviewers' concerns except the relevance of the second order generalized smoothness assumption. Authors argued that this assumption is justified by its use in a recent work (Xie et al., 2024) on minimization. However, unlike this prior work current manuscript fails to provide any problems where this assumption holds. Even for the additional synthetic experiments provided during discussion, authors did not show that the assumption holds without standard second-order smoothness assumption holding.

---

### Decision · Program_Chairs · 2025-01-22

Reject